# Neural Networks Efficiently Learn Low-Dimensional Representations with SGD

**Alireza Mousavi-Hosseini[1,2], Sejun Park[3], Manuela Girotti[4], Ioannis Mitliagkas[5,6], Murat A. Erdogdu[1,2]**
[1]University of Toronto, [2]Vector Insitute, [3]Korea University, [4]Saint Mary's University,
[5]Université de Montréal, [6]Mila Institute
`{mousavi, erdogdu}@cs.toronto,edu, sejun.park000@gmail.com`
`manuela.girotti@smu.ca, ioannis@iro.umontreal.ca`

## Abstract

We study the problem of training a two-layer neural network (NN) of arbitrary width using stochastic gradient descent (SGD) where the input $x \in \mathbb{R}^d$ is Gaussian and the target $y \in \mathbb{R}$ follows a multiple-index model, i.e., $y = g(\langle u_1, x \rangle, \ldots, \langle u_k, x \rangle)$ with a noisy link function $g$. We prove that the first-layer weights of the NN converge to the $k$-dimensional *principal subspace* spanned by the vectors $u_1, \ldots, u_k$ of the true model, when online SGD with weight decay is used for training. This phenomenon has several important consequences when $k \ll d$. First, by employing uniform convergence on this smaller subspace, we establish a generalization error bound of $\mathcal{O}(\sqrt{kd/T})$ after $T$ iterations of SGD, which is independent of the width of the NN. We further demonstrate that, SGD-trained ReLU NNs can learn a single-index target of the form $y = f(\langle u, x \rangle) + \epsilon$ by recovering the principal direction, with a sample complexity linear in $d$ (up to log factors), where $f$ is a monotonic function with at most polynomial growth, and $\epsilon$ is the noise. This is in contrast to the known $d^{\Omega(p)}$ sample requirement to learn any degree $p$ polynomial in the kernel regime, and it shows that NNs trained with SGD can outperform the neural tangent kernel at initialization. Finally, we also provide compressibility guarantees for NNs using the approximate low-rank structure produced by SGD.

## 1 Introduction

The task of learning an unknown statistical (teacher) model using data is fundamental in many areas of learning theory. There has been a considerable amount of research dedicated to this task, especially when the trained (student) model is a neural network (NN), providing precise and non-asymptotic guarantees in various settings (Zhong et al., 2017; Goldt et al., 2019; Ba et al., 2019; Sarao Mannelli et al., 2020; Zhou et al., 2021; Akiyama & Suzuki, 2021; Abbe et al., 2022; Ba et al., 2022; Damian et al., 2022; Veiga et al., 2022). As evident from these works, explaining the remarkable learning capabilities of NNs requires arguments beyond the classical learning theory (Zhang et al., 2021).

The connection between NNs and kernel methods has been particularly useful towards this expedition (Jacot et al., 2018; Chizat et al., 2019). In particular, a two-layer NN with randomly initialized and untrained weights is an example of a random features model (Rahimi & Recht, 2007), and regression on the second layer captures several interesting phenomena that NNs exhibit in practice (Louart et al., 2018; Mei & Montanari, 2022), e.g. *cusp* in the learning curve. However, NNs also inherit favorable characteristics from the optimization procedure (Ghorbani et al., 2019; Allen-Zhu & Li, 2019; Yehudai & Shamir, 2019; Li et al., 2020; Refinetti et al., 2021), which cannot be captured by associating NNs with regression on random features. Indeed, recent works have established a separation between NNs and kernel methods, relying on the emergence of representation learning as a consequence of gradient-based training (Abbe et al., 2022; Ba et al., 2022; Barak et al., 2022; Damian et al., 2022), which often exhibits a natural bias towards low-complexity models.

A theme that has emerged repeatedly in modern learning theory is the implicit regularization effect provided by the training dynamics (Neyshabur et al., 2014). The work by Soudry et al. (2018) has inspired an abundance of recent works focusing on the implicit bias of gradient descent favoring, in some sense, *low-complexity* models, e.g. by achieving min-norm and/or max-margin solutions

despite the lack of any explicit regularization (Gunasekar et al., 2018; Li et al., 2018; Ji & Telgarsky, 2019; Gidel et al., 2019; Chizat & Bach, 2020; Pesme et al., 2021). However, these works mainly consider linear models or unrealistically wide NNs, and the notion of reduced complexity as well as its implications on generalization varies. A concrete example in this domain is *compressiblity* and its connection to generalization (Arora et al., 2018; Suzuki et al., 2020). Indeed, when a trained NN can be compressed into a smaller NN with similar prediction behavior, the resulting models exhibit similar generalization performance. Thus, the model complexity of the original NN can be explained by the smaller complexity of the compressed one, which is classically linked to better generalization.

In this paper, we demonstrate the emergence of low-complexity structures during the training procedure. More specifically, we consider training a two-layer student NN with arbitrary width $m$ where the input $\boldsymbol{x} \in \mathbb{R}^d$ is Gaussian and the target $y \in \mathbb{R}$ follows a multiple-index teacher model, i.e. $y = g(\langle \boldsymbol{u_1}, \boldsymbol{x} \rangle, \ldots, \langle \boldsymbol{u_k}, \boldsymbol{x} \rangle; \epsilon)$ with a link function $g$ and a noise $\epsilon$ independent of the input. In this setting, we prove that the first-layer weights trained by online stochastic gradient descent (SGD) with weight decay converge to the $k$-dimensional subspace spanned by the weights of the teacher model, $\mathrm{span}(\boldsymbol{u_1}, ,\ldots, \boldsymbol{u_k})$, which we refer to as the *principal subspace*. Our primary focus is the case where the target values depend only on a few important directions along the input, i.e. $k \ll d$,

which induces a low-dimensional structure on the SGD-trained first-layer weights, whose impact on generalization is profound. First, convergence to the principal subspace leads to an improved bound on the generalization gap for SGD, independent of the width of the NN. In the specific case of learning a single-index target with a ReLU student network, we show that this convergence leads to useful features that improve upon the initial random features. Hence we prove that NNs can learn certain degree-$p$ polynomials with a number of samples (almost) linear in $d$ using online SGD, while learning a degree $p$ polynomial with any rotationally invariant kernel, including the neural tangent kernel (NTK) at initialization, requires $d^{\Omega(p)}$ samples (Donhauser et al., 2021). We summarize our contributions as follows.

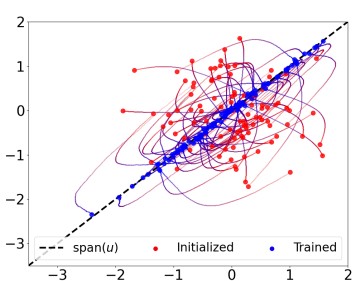

Figure 1: Two-layer ReLU network with $m = 1000$, $d = 2$ is trained to recover a $\tanh$ single-index model via SGD with weight decay. Initial neurons (red) converge to the principal subspace. 10% of student neurons are visualized.

- We show in Theorem 3 that NNs learn low-dimensional representations by proving that the iterates of online SGD on the first layer of a two-layer NN with width $m$ converge to $\sqrt{m}\varepsilon$ neighborhood of the principal subspace after $\mathcal{O}(d/\varepsilon^2)$ iterations, with high probability. The error tolerance of $\sqrt{m}\varepsilon$ is sufficient to guarantee that the risk of SGD iterates and that of its orthogonal projection to the principal subspace are within $\mathcal{O}(\varepsilon)$ distance.

- We demonstrate the impact of learning low-dimensional representations with three applications.

  - For a single-index target $y = f(\langle \boldsymbol{u}, \boldsymbol{x} \rangle) + \epsilon$ with a monotonic link function $f$ where $f''$ has at most polynomial growth, we prove in Theorem 4 that ReLU networks of width $m$ can learn this target after $T$ iterations of SGD with an excess risk estimate of $\tilde{\mathcal{O}}(\sqrt{d/T + 1/m})$, with high probability (see the illustration in Figure 1). In particular, the number of iterations is linear in the input dimension $d$, even when $f$ is a polynomial of any (fixed) degree $p$.
  - Based on a uniform convergence argument on the principal subspace, we prove in Theorem 5 that $T$ iterations of SGD will produce a model with generalization error of $\mathcal{O}(\sqrt{kd/T})$, with high probability. Remarkably, this rate is independent of the width $m$ of the NN, even in the case $k \asymp d$ where the target is any function of the input, and not necessarily low-dimensional.
  - Finally, we provide a compressiblity result directly following from the low-dimensionality of the principal subspace. We prove that $T$ iterations of SGD produce first-layer weights that are compressible to rank-$k$ with a risk deviation of $\mathcal{O}(\sqrt{d/T})$, with high probability.

The rest of the paper is organized as follows. We discuss the notation and the related work in the remainder of this section. We describe the problem formulation and preliminaries in Section 2, and provide an analysis for the warm-up case of population gradient descent in Section 2.1. Our main result on SGD is presented in Section 3. We discuss three implications of our main theorem in Section 4, where we provide results on learnability, generalization gap, and compressibility in Sections 4.1, 4.2, and 4.3, respectively. We finally conclude with a brief discussion in Section 5.

**Notation.** For a loss function $\ell : \mathbb{R}^2 \to \mathbb{R}$, let $\partial_i \ell$ and $\partial^2_{ij} \ell$ denote its partial derivatives with respect to $i$th and $j$th inputs for $i, j \in \{1, 2\}$. For quantities $a$ and $b$, $a \lesssim b$ implies $a \leq Cb$ for some absolute constant $C > 0$, and $a \asymp b$ implies both $a \gtrsim b$ and $a \lesssim b$. Finally, $\text{Unif}(A)$ denotes the uniform distribution over a set $A$ and $\mathcal{N}(0, \mathbf{I}_d)$ denotes the $d$-dimensional isotropic Gaussian distribution.

## 1.1 RELATED WORK

**Training dynamics of NNs.** Several works have demonstrated learnability in a special case of teacher-student setting where the teacher model is *similar* to the student NN being trained (Zhong et al., 2017; Brutzkus & Globerson, 2017; Li & Yuan, 2017; Zhang et al., 2019; Zhou et al., 2021). This setting has also been studied through the lens of loss landscape (Safran et al., 2021) and optimization over measures (Akiyama & Suzuki, 2021). We stress that our results work under misspecification and hold for generic teacher models that are not necessarily NNs with similar architecture to the student.

Two scaling regimes of analysis have seen a surge of recent interest. In the regime of lazy training (Chizat et al., 2019), the parameters hardly move from initialization and the NN does not learn useful features, behaving like a kernel method (Jacot et al., 2018; Du et al., 2019; Allen-Zhu et al., 2019; Arora et al., 2019; Oymak & Soltanolkotabi, 2020). However, many works have shown that deep learning is more powerful than kernel models (Yehudai & Shamir, 2019; Ghorbani et al., 2020; Geiger et al., 2020), establishing a clear separation between them; thus, several important characteristics of NNs cannot be captured with lazy training (Ghorbani et al., 2019), even though it might still perform better than feature learning in certain low-dimensional settings (Petrini et al., 2022). In the other scaling regime, gradient descent on infinitely wide NNs reduces to Wasserstein gradient flow, known as the mean-field regime where feature learning is possible (Chizat & Bach, 2018; Rotskoff & Vanden-Eijnden, 2018; Mei et al., 2019; Nitanda et al., 2022; Chizat, 2022). Closer to our results, the concurrent work of Hajjar & Chizat (2022) shows that low-dimensional targets induce low-dimensional dynamics on mean-field NNs. However, these results mostly hold for infinite or very wide NNs, and quantitative guarantees are difficult to obtain. Our setting is different from both of these regimes, as we allow for NNs of arbitrary width without excessive overparameterization.

**Feature learning with multiple-index teacher models.** The task of learning a target of an unknown low-dimensional function of the input is fundamental in statistics (Li & Duan, 1989). Several recent works in the learning theory literature have also focused on this problem, with an aim to demonstrate NNs can learn useful feature representations, outperforming kernel methods (Bauer & Kohler, 2019; Ghorbani et al., 2020). In particular, Abbe et al. (2022) studies the necessary and sufficient conditions for learning with sample complexity linear in $d$ with inputs on the hypercube, in the mean-field limit. Closer to our setting are the recent works Ba et al. (2022); Damian et al. (2022); Barak et al. (2022) which demonstrate a clear separation between NNs and kernel methods, leveraging the effect of representation learning. However, their analysis considers a single (full) gradient step on the first-layer weights followed by training the second-layer parameters. In contrast, in our learnability result, we consider training both layers with SGD, which induces essentially different learning dynamics.

**Generalization bounds for SGD.** A popular algorithm-dependent approach for studying generalization is through algorithmic stability (Bousquet & Elisseeff, 2002; Feldman & Vondrak, 2018; Bousquet et al., 2020), which has been used to study the generalization behavior of gradient-based methods in various settings (Hardt et al., 2016; Bassily et al., 2020; Farghly & Rebeschini, 2021; Kozachkov et al., 2022). Other approaches include studying the low-dimensional structure of the trajectory (Simsekli et al., 2020; Park et al., 2022) or the invariant measure of continuous-time approximations of SGD (Camuto et al., 2021), and employing information-theoretic tools (Neu et al., 2021). Among these works, Barsbey et al. (2021) show that SGD is able to learn compressible networks. However, they require large width for the mean-field approximation and assume that the SGD iterates converge to a heavy-tailed distribution, while we do not make either of the assumptions.

## 2 PRELIMINARIES: NEURAL NETWORKS AND THE PRINCIPAL SUBSPACE

For an input $\boldsymbol{x} \in \mathbb{R}^d$, we consider training a two-layer neural network (NN) with $m$ neurons

$$\hat{y}(\boldsymbol{x}; \boldsymbol{W}, \boldsymbol{a}, \boldsymbol{b}) = \sum_{i=1}^{m} a_i \sigma(\langle \boldsymbol{w}_i, \boldsymbol{x} \rangle + b_i), \tag{2.1}$$

where $\sigma$ is the activation function, $\{\boldsymbol{w}_i\}_{1\leq i\leq m}$ are the first-layer weights collected in the rows of the matrix $\boldsymbol{W} \in \mathbb{R}^{m\times d}$, $\boldsymbol{b} \in \mathbb{R}^m$ is the bias, and $\boldsymbol{a} \in \mathbb{R}^m$ is the second-layer weights. We assume $\boldsymbol{x} \sim \mathcal{N}(0, \mathbf{I}_d)$ and the target is generated from a multiple-index (teacher) model given by

$$y = g(\langle \boldsymbol{u}_1, \boldsymbol{x}\rangle, \ldots, \langle \boldsymbol{u}_k, \boldsymbol{x}\rangle; \epsilon), \tag{2.2}$$

for a weakly differentiable link function $g : \mathbb{R}^{k+1} \to \mathbb{R}$ and a noise $\epsilon$. Throughout the paper, the noise $\epsilon$ is assumed to be independent from the input $\boldsymbol{x}$, and our framework covers the special noiseless case where $\epsilon = 0$. While our results remain valid regardless of how $k$ and $d$ compare, they are most insightful when $k \ll d$; thus, we specifically consider this regime when interpreting the results. We also collect the teacher weights $\{\boldsymbol{u}_i\}_{1\leq i\leq k}$ in the rows of the matrix $\boldsymbol{U} \in \mathbb{R}^{k\times d}$ and use $y = g(\boldsymbol{U}\boldsymbol{x}; \epsilon)$ for simplicity.

For a given loss function $\ell(\hat{y}, y)$, we consider the population and the empirical risks

$$R(\boldsymbol{W}, \boldsymbol{a}, \boldsymbol{b}) := \mathbb{E}[\ell(\hat{y}(\boldsymbol{x}; \boldsymbol{W}, \boldsymbol{a}, \boldsymbol{b}), y)] \quad \text{and} \quad \hat{R}(\boldsymbol{W}, \boldsymbol{a}, \boldsymbol{b}) := \frac{1}{T}\sum_{t=0}^{T-1} \ell(\hat{y}(\boldsymbol{x}^{(t)}; \boldsymbol{W}, \boldsymbol{a}, \boldsymbol{b}), y^{(t)}),$$

where the expectation is over the data distribution. Similarly, for some $\tau \geq 1$, the truncated loss is defined as $\ell_\tau(\hat{y}, y) := \ell(\hat{y}, y) \wedge \tau$ with the corresponding risks $R_\tau$ and $\hat{R}_\tau$, both of which are used in Section 4 to obtain sharp high probability statements. In the warm-up case, we consider the $L_2$-regularized population risk with a penalty parameter $\lambda \geq 0$, defined as

$$\mathcal{R}_\lambda(\boldsymbol{W}, \boldsymbol{a}, \boldsymbol{b}) := R(\boldsymbol{W}, \boldsymbol{a}, \boldsymbol{b}) + \frac{\lambda}{2}\|\boldsymbol{W}\|_{\mathrm{F}}^2. \tag{2.3}$$

To minimize (2.3), we use stochastic gradient descent (SGD) over the first-layer weights, where we are interested in the convergence of iterates to the *principal subspace* defined by the teacher weights

$$\mathbb{S}(\boldsymbol{U}) := \mathrm{span}(\boldsymbol{u}_1, \ldots, \boldsymbol{u}_k)^m = \{\boldsymbol{C}\boldsymbol{U} : \boldsymbol{C} \in \mathbb{R}^{m\times k}\}.$$

Notice that the principal subspace satisfies $\mathbb{S}(\boldsymbol{U}) \subseteq \mathbb{R}^{m\times d}$, and its dimension is $mk$ as opposed to the ambient dimension of $md$, with any matrix in this subspace having rank at most $k$. For any vector $\boldsymbol{v} \in \mathbb{R}^d$, we let $\boldsymbol{v}_\parallel$ denote the orthogonal projection of $\boldsymbol{v}$ onto $\mathrm{span}(\boldsymbol{u}_1, \ldots, \boldsymbol{u}_k)$ and $\boldsymbol{v}_\perp := \boldsymbol{v} - \boldsymbol{v}_\parallel$. Similarly, for a matrix $\boldsymbol{W} \in \mathbb{R}^{m\times d}$, we define $\boldsymbol{W}_\parallel$ and $\boldsymbol{W}_\perp$ by applying the projection to each row.

We make the following assumption on the data generating process.

**Assumption 1** (Student-teacher setup). *The student model is a two-layer NN (2.1) trained over the data set $\{(\boldsymbol{x}^{(i)}, y^{(i)})\}_{i\geq 1}$, where the target values $y^{(i)}$ are generated according to the teacher model (2.2) and the inputs satisfy $\boldsymbol{x}^{(i)} \overset{\text{iid}}{\sim} \mathcal{N}(0, \mathbf{I}_d)$. The link function $g(\cdot, \ldots, \cdot; \epsilon)$ is weakly differentiable (see e.g. Evans (2010, Sec. 5.2.1) for definition) for any fixed $\epsilon$.*

The Gaussian input is a rather standard assumption in the literature, especially in recent works that consider the student-teacher setup; see e.g. Safran et al. (2021); Zhou et al. (2021); Damian et al. (2022). The multiple-index teacher model (2.2) can encode a broad class of input-output relations through the non-linear link function, including a multi-layer fully-connected NN with arbitrary depth and width and weakly differentiable activations. The smoothness properties of the activation $\sigma$ play an important role in our analysis. As such, we consider two scenarios, with different requirements on the loss function.

**Assumption 2.A** (Smooth activation). *The activation function $\sigma$ satisfies $|\sigma(z)|, |\sigma'(z)|, |\sigma''(z)| \leq 1$ for all $z \in \mathbb{R}$, the loss is $\ell(\hat{y}, y) = \frac{1}{2}(\hat{y} - y)^2$ for simplicity, and $y$ satisfies $|y| \leq K$ almost surely.*

**Assumption 2.B** (ReLU activation). *The activation function $\sigma$ is $\sigma(z) = \max(z, 0)$ for $z \in \mathbb{R}$. The loss satisfies $0 \leq \partial_1^2 \ell(\hat{y}, y) \leq 1$, $|\partial_1 \ell(\hat{y}, y)| \leq 1$, and $|\partial_{12}^2 \ell(\hat{y}, y)| \leq 1$.*

Commonly used activations such as sigmoid and tanh satisfy Assumption 2.A. For ReLU activation in Assumption 2.B, we choose $\sigma'(z) = \mathbf{1}(z \geq 0)$ as its weak derivative. We highlight that Assumption 2.B is satisfied by common Lipschitz and convex loss functions such as the Huber loss

$$\ell_{\mathrm{H}}(\hat{y} - y) := \begin{cases} \frac{1}{2}(\hat{y} - y)^2 & \text{if } |\hat{y} - y| \leq 1 \\ |\hat{y} - y| - \frac{1}{2} & \text{if } |\hat{y} - y| > 1, \end{cases} \tag{2.4}$$

as well as the logistic loss $\ell_{\mathrm{L}}(\hat{y}, y) := \log(1 + e^{-\hat{y}y})$, up to appropriate scaling constants.

## 2.1 WARM-UP: POPULATION GRADIENT DESCENT

In this section, we study the dynamics of population gradient descent (PGD) to motivate our investigation of the more practically relevant case of SGD. When initialized from $\boldsymbol{W}^0$, PGD with a fixed step size $\eta$ will update the current iterate $\boldsymbol{W}^t$ according to the update rule

$$\boldsymbol{W}^{t+1} = \boldsymbol{W}^t - \eta \nabla_{\boldsymbol{W}} \mathcal{R}_\lambda(\boldsymbol{W}^t), \tag{2.5}$$

We use the following initialization throughout the paper.

**Assumption 3** (Initialization). *For all $1 \le i \le m$, $1 \le j \le d$, we initialize the NN weights and biases with $\sqrt{d}W_{ij}^0 \overset{\text{iid}}{\sim} \mathcal{N}(0,1)$, $ma_i^0 \overset{\text{iid}}{\sim} \text{Unif}([-1,1])$, and $b_i^0 \overset{\text{iid}}{\sim} \text{Unif}(\{-1,1\})$.*

While this initialization is standard in the mean-field regime, we only use it to simplify the exposition. Indeed, we can initialize $\boldsymbol{W}$ and $\boldsymbol{a}$ with any scheme that guarantees $\|\boldsymbol{W}\|_F \lesssim \sqrt{m}$ and $\|\boldsymbol{a}\|_\infty \lesssim m^{-1}$ with high probability. Further, initialization of $\boldsymbol{b}$ mostly matters in the analysis of ReLU activation.

Next, we show that the population gradient admits a certain decomposition which plays a central role in our analysis. For smooth activations, the below result is a remarkable consequence of Stein's lemma, which provides a certain alignment between the true statistical model (teacher) and the model being trained (student), which has profound impact on the learning dynamics. We generalize this result for ReLU through a sequence of smooth approximations (see Appendix A.1 for details).

**Lemma 1.** *Under Assumptions 1&2.A or 1&2.B, the gradient of the population risk can be written as*

$$\nabla_{\boldsymbol{W}} \mathcal{R}_\lambda(\boldsymbol{W}) = (\mathcal{H}(\boldsymbol{W}) + \lambda \mathbf{I}_m)\boldsymbol{W} + \mathcal{D}(\boldsymbol{W})\boldsymbol{U}, \tag{2.6}$$

*for some $\mathcal{H}(\boldsymbol{W}) \in \mathbb{R}^{m \times m}$ and $\mathcal{D}(\boldsymbol{W}) \in \mathbb{R}^{k \times d}$ (with explicit forms provided in Appendix A.1).*

Notice that the subset of critical points $\{\boldsymbol{W}_* : \nabla \mathcal{R}_\lambda(\boldsymbol{W}_*) = 0\}$ for which $\mathcal{H}(\boldsymbol{W}_*) + \lambda \mathbf{I}_m$ is invertible belongs to the principle subspace, i.e. $\boldsymbol{W}_* \in \mathbb{S}(\boldsymbol{U})$. Further, if we initialize PGD (2.5) within the principal subspace, i.e. $\boldsymbol{W}^0 \in \mathbb{S}(\boldsymbol{U})$, the subsequent iterates for $t > 0$ remain in this subspace.

In statistics literature, the setting we consider is often termed as *model misspecification*, i.e. the teacher model generating the data and the student model being trained are different. Proposition 2 states a general result that, as long as the target depends on certain directions, PGD will produce weights in their span, despite the possible mismatch between the two models. We highlight that the classical results on dimension reduction, e.g. Li & Duan (1989); Li (1991), rely on a similar principle, which was also used for designing optimization algorithms, see e.g. Erdogdu et al. (2019). However, we are interested in the implications of this phenomenon in the setting of NNs trained with SGD.

The following result, proved in Appendix A.2, demonstrates the algorithmic implications of Lemma 1 for the simplistic case of PGD, and shows that the iterates converges to the principal subspace.

**Proposition 2.** *Consider running $T$ PGD iterations (2.5) with an initialization satisfying Assumption 3 and a step size $\eta > 0$. For any $\gamma > 0$, choose $\lambda = \frac{\tilde{\lambda}}{m}$ and $\eta = m\tilde{\eta}$ based on the following.*

1. ***Smooth activation.*** *Under Assumptions 1&2.A, let $\tilde{\lambda} \ge 1 + \gamma + \sqrt{1 + 2\gamma + 2R(\boldsymbol{W}^0)}$ and $\tilde{\eta} \lesssim (\tilde{\lambda} + \frac{\tilde{\lambda}_\varsigma}{\gamma} + \varsigma)^{-1}$ where $\varsigma := \mathbb{E}[\|\boldsymbol{U}^\top \nabla g(\boldsymbol{U}\boldsymbol{x}; \epsilon)\|_2]$.*

2. ***ReLU activation.*** *Under Assumptions 1&2.B, let $\tilde{\lambda} \ge \gamma + 2\sqrt{\frac{2}{e\pi}}$ and $\tilde{\eta} \lesssim \tilde{\lambda}^{-1}$.*

*Then, with probability at least $1 - e^{-Cmd}$ over the initialization, where $C$ is an absolute constant, the iterates of PGD satisfy*

$$\|\boldsymbol{W}_\perp^T\|_F \le (1 - \tilde{\eta}\gamma)^T \|\boldsymbol{W}_\perp^0\|_F. \tag{2.7}$$

A few remarks are in order. First and most importantly, PGD iterates converge to the principal subspace as $T \to \infty$ and this phenomenon is mainly due to the alignment provided by Lemma 1. Indeed in the limit, (2.7) provides sparsity in the basis of principal subspace, and it is widely known that $L_2$-regularization does not have a similar sparsity effect (unless $\lambda \to \infty$) in contrast to its $L_1$ counterpart. Thus, the choice of $\lambda$ in Proposition 2 will lead to non-trivial orthogonal projections $\boldsymbol{W}_\parallel^T$ in general, as we will demonstrate in Section 4.1 with a learnability result. However, without $L_2$-regularization, i.e. $\lambda = 0$, it is possible to converge to a critical point $\boldsymbol{W}^*$ for which $\mathcal{H}(\boldsymbol{W}^*) + \lambda \mathbf{I}_m$ is not invertible; hence, the weights are likely to be outside of the principal subspace in this case (cf. Figures 1&2). It is also worth emphasizing that the penalty level used in the above proposition still allows for non-convexity as we demonstrate with an example in Appendix D. We finally remark that, as evident from the proof, Proposition 2 remains valid even with unbounded smooth activations.

## 3 CONVERGENCE OF STOCHASTIC GRADIENT DESCENT

We now consider stochastic gradient descent (SGD) in the online setting where at each iteration $t$, we have access to a new data point $(\boldsymbol{x}^{(t)}, y^{(t)})$ drawn independently of the previous samples from the same distribution. We update the first-layer weights $\boldsymbol{W}^t$ with a (possibly) time varying step size $\eta_t$ and a weight decay, according to the update rule

$$\boldsymbol{W}^{t+1} = (1 - \eta_t \lambda) \boldsymbol{W}^t - \eta_t \nabla_{\boldsymbol{W}} \ell(\hat{y}(\boldsymbol{x}^{(t)}; \boldsymbol{W}^t, \boldsymbol{a}, \boldsymbol{b}), y^{(t)}). \tag{3.1}$$

The above algorithm can be used to minimize the population risk (2.3) in practice (Polyak & Juditsky, 1992), even in certain non-convex landscapes (Yu et al., 2021). As we demonstrate next, SGD still preserves several important characteristics of its population counterpart, PGD.

**Theorem 3.** *Consider running $T$ SGD iterations* (3.1) *over samples satisfying Assumption 1, with an initialization satisfying Assumption 3, and using the following step size schedules.*

1. ***Constant step size.*** *Under Assumption* 2.A, *choose the constant step size* $\eta_t = \frac{2m \log(T)}{\gamma T}$. *For any* $\gamma > 0$, *let* $\tilde{\lambda} \geq 1 + \gamma + \sqrt{1 + 2\gamma + 2R(\boldsymbol{W}^0) + \frac{C \log(T)^2 d}{\gamma^2 T}}$ *where $C$ is an absolute constant, and suppose* $T \gtrsim (1/\delta)^{C/d} \vee \frac{\tilde{\lambda}}{\gamma} \log \frac{\tilde{\lambda}}{\gamma}$. *Then, for a penalty* $\lambda = \frac{\tilde{\lambda}}{m}$, *with probability at least* $1 - \delta$,

$$\frac{\|\boldsymbol{W}_\perp^T\|_F}{\sqrt{m}} \lesssim \sqrt{\frac{\log(T)(d + \log(1/\delta))}{\gamma^2 T}}, \tag{3.2}$$

2. ***Decaying step size.*** *Let* $\zeta := \mathbb{E}[|y|] + 1$ *under Assumption* 2.A *and* $\zeta := 2\sqrt{2/e\pi}$ *under Assumption* 2.B. *Choose the decreasing step size* $\eta_t = m \frac{2(t+t^*)+1}{\gamma(t+t^*+1)^2}$, $\tilde{\lambda} \geq \gamma + \zeta$, *and* $t^* \asymp \frac{\tilde{\lambda}}{\gamma}$ *for any* $\gamma > 0$. *Then, for* $\lambda = \frac{\tilde{\lambda}}{m}$, *with probability at least* $1 - \delta$,

$$\frac{\|\boldsymbol{W}_\perp^T\|_F}{\sqrt{m}} \lesssim \sqrt{\frac{d + \log(1/\delta)}{\gamma^2 T}}, \tag{3.3}$$

*whenever* $m \gtrsim \log(1/\delta)$ *and* $T \gtrsim \frac{\tilde{\lambda}^2}{d + \log(1/\delta)}$.

**Remark.** Our results are most insightful when $\gamma \asymp 1$ with respective rates of $\widetilde{\mathcal{O}}(\sqrt{d/T})$ and $\mathcal{O}(\sqrt{d/T})$ in the constant and decreasing step size settings; this scaling allows efficient learning of certain targets (see Section 4.1). Indeed, choosing a large $\gamma$ may significantly restrict the learnability properties and will result in underfitting. However, if we ignore the underfitting issue, one can get the fastest convergence rate by choosing $\gamma \asymp \sqrt{T(d + \log(1/\delta))}$, from which we obtain $\|W_\perp^T\|_F / \sqrt{m} \lesssim 1/T$. We also note that the convergence rate is stated in the normalized distance to the principal subspace, i.e. $\|\boldsymbol{W}_\perp^T\|_F / \sqrt{m} \leq \varepsilon$, as this is sufficient to guarantee that the risk of $\boldsymbol{W}^T$ and its orthogonal projection $\boldsymbol{W}_\parallel^T$ are within $\mathcal{O}(\varepsilon)$ distance.

The above result states that, with a number of samples linear in $d$, SGD is able to produce iterates that are in close proximity to the principal subspace; thus, it efficiently learns (approximately) low-dimensional weights, exhibiting an implicit bias towards low-complexity models. While prior works have established that NNs adapt to low-dimensional manifold structures in the data in some contexts (Chen et al., 2019; Buchanan et al., 2021; Wang et al., 2021), our result has a different nature. More specifically, the interplay between two forces is in effect here. The most important one is the linear relationship between the first-layer weights and the input in both student and teacher models together with the input distribution. The alignment described in Lemma 1 yields *sparsified* weights in a basis defined by the teacher network, effectively reducing the dimension from $d$ to $k$. The second force is the explicit $L_2$-regularization. We emphasize that $L_2$-regularization does not play the main role in this sparsification; even though it may provide shrinkage to zero, $L_2$ penalty will in general produce non-sparse solutions. However, it is still required to ensure that SGD avoids critical points outside of the principal subspace.

Although Theorem 3 does not have any implications on the convergence behavior of $\boldsymbol{W}_\parallel$, in the next section we show that the implied low-dimensional structure is sufficient to provide guarantees on the generalization error, learnability, and compressibility of SGD. The proof of this Theorem is provided in Appendix B, and is based on a recursion on the moment generating function of $\|\boldsymbol{W}_\perp^t\|_F$. We note that, as in Proposition 2, regularization in Theorem 3 does not imply (strong) convexity in general, which we demonstrate in a non-convex example in Appendix D.

## 4 IMPLICATIONS OF LOW-DIMENSIONALITY

### 4.1 LEARNING SINGLE-INDEX TARGETS

An essential characteristic of NNs is their ability to learn useful representations, which allows them to adapt to the underlying misspecified statistical model. Although this fundamental property has been the guiding principle in all empirical studies, it was mathematically proven only recently for gradient-based training (Abbe et al., 2022; Ba et al., 2022; Barak et al., 2022; Damian et al., 2022; Frei et al., 2022); see also a survey of prior works in Malach et al. (2021). Our results in the previous section are in the same spirit, establishing the convergence of SGD to the principal subspace which is indeed a span of useful directions associated with the target function being learned. As such, we leverage the learned low-dimensional representations to demonstrate that SGD is capable of learning a target function of the form $y = f(\langle \boldsymbol{u}, \boldsymbol{x} \rangle) + \epsilon$ with a number of samples linear in $d$ (up to logarithmic factors). For simplicity, we work with the Huber loss below; however, we can accommodate any Lipschitz and convex loss at the expense of a more detailed analysis.

---

**Algorithm 1** Training a two-layer ReLU network with SGD.

---

**Input:** $\boldsymbol{a}^0, \boldsymbol{b}^0 \in \mathbb{R}^m$, $\boldsymbol{W}^0 \in \mathbb{R}^{m \times d}$, $\{(\boldsymbol{x}^{(t)}, y^{(t)})\}_{0 \leq t \leq T-1}$, $(\eta_t)_{t \geq 0}$, $(\eta'_t)_{t \geq 0}$, $\lambda$, $\lambda'$, $\Delta$.
  1: **for** $t = 0, ..., T - 1$ **do**
  2:      $\boldsymbol{W}^{t+1} = (1 - \eta_t \lambda)\boldsymbol{W}^t - \eta_t \nabla_{\boldsymbol{W}} \ell(\hat{y}(\boldsymbol{x}^{(t)}; \boldsymbol{W}^t, \boldsymbol{a}^0, \boldsymbol{b}^0), y^{(t)})$.
  3: **end for**
  4: Let $b_j \overset{iid}{\sim} \text{Unif}(-\Delta, \Delta)$ for $1 \leq j \leq m$.
  5: **for** $t = 0, ..., T' - 1$ **do**
  6:      Sample $i_t \sim \text{Unif}\{0, ..., T - 1\}$.
  7:      $\boldsymbol{a}^{t+1} = (1 - \eta'_t \lambda')\boldsymbol{a}^t - \eta'_t \nabla_{\boldsymbol{a}} \ell(\hat{y}(\boldsymbol{x}^{(i_t)}; \boldsymbol{W}^T, \boldsymbol{a}^t, \boldsymbol{b}), y^{(i_t)})$
  8: **end for**
  9: **return** $(\boldsymbol{W}^T, \boldsymbol{a}^{T'}, \boldsymbol{b})$.

---

In the sequel, we use Algorithm 1 and train the first layer of the NN with online SGD using $T$ data samples. Then, we randomly choose the biases and run $T'$ SGD iterations on the second layer using the same data samples used to train the first layer. Thus, the overall sample complexity is $T$ whereas the total number of SGD iterations performed is $T + T'$. We highlight that the recent works Ba et al. (2022); Barak et al. (2022); Damian et al. (2022) perform only one gradient step on the first layer weights, whereas in Algorithm 1, we train the entire NN with SGD.

**Theorem 4.** *Suppose that the data is from a single-index model $y = f(\langle \boldsymbol{u}, \boldsymbol{x} \rangle) + \epsilon$ with a monotone differentiable $f$ and $\nu$-sub-Gaussian noise $\epsilon$, and Assumptions 1&2.B hold. Further, let $\|\boldsymbol{u}\|_2 = 1$, $|f(0)| < 1$, and consider the Huber loss (2.4) for simplicity. Consider running Algorithm 1 with the initialization $0 < a_j^0 = a \lesssim 1/m$, $0 < b_j^0 = b \lesssim 1$, and $\boldsymbol{w}_j^0 = \boldsymbol{w}^0 \sim \mathcal{N}(0, \frac{1}{d}\mathbf{I}_d)$ for all $j$ with the hyper-parameters $\lambda = \frac{\tilde{\lambda}}{m} = \frac{\gamma}{m} + \frac{2a}{b}\sqrt{\frac{2}{e\pi}}$ for any $\gamma \gtrsim 1$, $\eta_t = m\frac{2(t^*+t)+1}{\gamma(t^*+t+1)^2}$ with $t^* \asymp \gamma^{-1}$, $\eta'_t = \frac{2t+1}{\lambda'(t+1)^2}$, and $\Delta \asymp \sqrt{\log(T/\delta)}$. Then, for $T \gtrsim (d + \log(\frac{1}{\delta})) \vee (\frac{\tilde{\lambda}}{\gamma d}\log(\frac{m}{\delta}))$, some $\lambda' > 0$ (see (C.8)), and sufficiently large $T'$ (see (C.9)), with probability at least $1 - \delta$,*

$$R_\tau(\boldsymbol{W}^T, \boldsymbol{a}^{T'}, \boldsymbol{b}) - \mathbb{E}[\ell_{\mathrm{H}}(\epsilon)] \lesssim \Delta_*^2 \left\{ \sqrt{\frac{\log(T/\delta)}{m}} + \sqrt{\frac{d + \log(1/\delta)}{T}} \right\} + \nu\sqrt{\frac{\log(1/\delta)}{T}}, \quad (4.1)$$

*where $\Delta_*$, defined in (C.7), is $\text{poly}(\log(\frac{T}{\delta}))$ when $f''$ has at most polynomial growth.*

This result implies that a ReLU NN trained with SGD can learn any monotone polynomial with a sample complexity linear in the input dimension $d$, up to logarithmic factors. Indeed, this is consistent with the work of Ben Arous et al. (2021); they establish a sharp sample complexity of $\tilde{\mathcal{O}}(d^{1 \vee (\mathcal{I}-2)})$ to learn a target with online SGD using the same activation $f$ in the student network, where $\mathcal{I}$ is the *information exponent* ($\mathcal{I} = 1$ in the above case due to the monotonocity of $f$). Despite assuming the link function $f$ is known, we highlight that their setting covers $\mathcal{I} \geq 1$, whereas Theorem 4 is a proof of concept to demonstrate the learnability implications of convergence to the principal subspace, even when $f$ is unknown. Building on their work, the concurrent work of Bietti et al. (2022) also proves learnability for unkown single-index targets with $\mathcal{I} \geq 1$, albeit with a sample complexity of $d^2$ for

$\mathcal{I} = 1$ when training ReLU students. Nonparametric regression with NNs has also been considered within the NTK framework (Hu et al., 2021; Kuzborskij & Szepesvári, 2021), but our result holds beyond this regime as $m$ grows with $\mathrm{poly}\log(T/\delta)$, in contrast to the $\mathrm{poly}(T)$ requirement of the NTK regime. Additionally, learning any degree $p$ polynomial using rotationally invariant kernels requires $d^{\Omega(p)}$ samples for a variety of input distributions including isotropic Gaussian (Donhauser et al., 2021); thus, our result shows that SGD is able to efficiently learn a target function where kernel methods cannot. For polynomial targets, Damian et al. (2022) consider training the first-layer weights with one gradient descent step with a carefully chosen weight decay, and obtain a sample complexity of $d^2$ to learn any degree $p$ polynomial depending on a few directions. Finally, Chen & Meka (2020) propose a method that can train NNs to learn such polynomials with sample complexity linear in $d$; yet, their algorithm is not a simple variant of SGD and requires a non-trivial *warm-start* initialization.

The proof of Theorem 4, detailed in Appendix C.2, relies on the fact that after training the first layer, the weights will align with the true direction $\boldsymbol{u}$. Then, similarly to Damian et al. (2022) we construct an optimal $\boldsymbol{a}^*$ with $|a_i^*| \asymp m^{-1}$ for every $i$ with a small empirical risk, and employ the generalization bound of Theorem 5 to achieve a rate estimate on the population risk.

## 4.2 GENERALIZATION GAP

For a given learning algorithm, the gap between its empirical and population risks is termed as the *generalization gap* (not to be confused with excess risk), and establishing convergence estimates for this quantity is a fundamental problem in learning theory. Classical results rely on uniform convergence over the feasible domain containing the weights; thus, they apply to any learning algorithm including SGD (Neyshabur et al., 2019). However, these bounds often diverge with the width of the NN, yielding vacuous estimates in the overparameterized regime (Zhang et al., 2021). To alleviate this, recent works considered establishing estimates for a specific learning algorithm; see e.g. Hardt et al. (2016); Soudry et al. (2018); Yun et al. (2021); Park et al. (2022).

Here, we are interested in deriving an estimate for the generalization gap over the SGD-trained first-layer weights, which holds uniformly over the second layer weights and biases. More specifically, we study, after $T$ iterations of SGD (3.1) initialized with $(\boldsymbol{W}^0, \boldsymbol{a}^0, \boldsymbol{b}^0)$, the following quantity

$$\mathcal{E}(\boldsymbol{W}^T) := \sup_S R_\tau(\boldsymbol{W}^T, \boldsymbol{a}, \boldsymbol{b}) - \hat{R}_\tau(\boldsymbol{W}^T, \boldsymbol{a}, \boldsymbol{b}) \quad \text{with} \quad S := \{\boldsymbol{a}, \boldsymbol{b} \in \mathbb{R}^m : \|\boldsymbol{a}\|_2 \le \tfrac{r_a}{\sqrt{m}}, \|\boldsymbol{b}\|_\infty \le r_b\},$$

where the scaling ensures $\hat{y} = \mathcal{O}(1)$ when $\|\boldsymbol{w}_j\|_2 \asymp 1$, which is the setting considered in Theorem 4. We state the following bound on $\mathcal{E}(\boldsymbol{W}^T)$; the proof is provided in Appendix C.1, and it is based on a covering argument over the smaller dimensional principal subspace implied by Theorem 3.

**Theorem 5.** *Consider the setting of Theorem 3 with either decreasing or constant step size. For any $\delta > 0$, if $T \gtrsim (d + \log(1/\delta)) \vee (\frac{\kappa\tilde{\lambda}}{\gamma d} \log(m/\delta))$, then with probability at least $1 - \delta$,*

$$\mathcal{E}(\boldsymbol{W}^T) \lesssim \tau r_a \left\{ \sqrt{\frac{\kappa(d + \log(1/\delta))}{\gamma^2 T}} + (r_b + \tilde{\lambda}^{-1})\sqrt{\frac{dk}{T}} \right\}, \tag{4.2}$$

*where we let $\kappa = 1$ for decreasing step size and $\kappa = \log(T)$ for constant step size.*

The above bound is independent of the width $m$ of the NN, and only grows with the dimension of the input space $d$ and that of the principal subspace $k$; thus, producing non-vacuous estimates in the overparametrized regime where $m$ is large. Further, the bound is stable in the number of SGD iterations $T$, that is, it converges to zero as $T \to \infty$. We remark that generalization bounds for SGD that rely on algorithmic stability are optimal for strongly convex objectives (Hardt et al., 2016); yet, they lead to unstable diverging bounds in non-convex settings as $T \to \infty$. As such, these techniques often require *early stopping*, which is clearly not needed in our result.

## 4.3 COMPRESSIBILITY

NNs exhibit compressiblity features in empirical studies, which is known to be associated with better generalization. Under the assumption that the trained network is compressible, several works established generalization bounds, see e.g. Arora et al. (2018); Suzuki et al. (2020). However, a theoretical justification of this assumption, specifically for a NN trained with SGD, was missing.

Indeed, Theorem 3 provides a concrete answer to this question; since the SGD iterate $\boldsymbol{W}^T$ converges to a low-rank matrix, the resulting weights are compressible. More precisely, let $\pi_k : \mathbb{R}^{m \times k} \to \mathbb{R}^{m \times k}$ be the low-rank approximation operator defined by $\pi_k(\boldsymbol{W}) := \arg\min_{\{\boldsymbol{W}':\mathrm{rank}(\boldsymbol{W}') \leq k\}} \|\boldsymbol{W} - \boldsymbol{W}'\|_{\mathrm{F}}$. As $\boldsymbol{W}_\|^T$ lies in the principal subspace, it has rank at most $k$. Thus, we can write

$$\|\boldsymbol{W}^T - \pi_k(\boldsymbol{W}^T)\|_{\mathrm{F}} \leq \|\boldsymbol{W}^T - \boldsymbol{W}_\|^T\|_{\mathrm{F}} = \|\boldsymbol{W}_\perp^T\|_{\mathrm{F}},$$

and the following is an immediate consequence of the bound in Theorem 3, that is $\|\boldsymbol{W}_\perp^T\|_{\mathrm{F}}/\sqrt{m} \lesssim \sqrt{d/T}$, combined with the Lipschitzness of $R_\tau(\boldsymbol{W})$, which we prove in Lemma 17.

**Proposition 6.** *Consider the setting of Theorem 3 with either decreasing or constant step size. Then, with probability at least $1 - \delta$,*

$$\left| R_\tau(\pi_k(\boldsymbol{W}^T), \boldsymbol{a}, \boldsymbol{b}) - R_\tau(\boldsymbol{W}^T, \boldsymbol{a}, \boldsymbol{b}) \right| \lesssim \frac{\tau\kappa}{\gamma}\sqrt{\frac{d + \log(1/\delta)}{T}}, \tag{4.3}$$

*where we let $\kappa = 1$ for decreasing step size and $\kappa = \sqrt{\log(T)}$ for constant step size.*

This result demonstrates that the low-dimensionality exhibited by the trained NN provides a rate of $\mathcal{O}(\sqrt{d/T})$ for the gap between its population risk and that of its compressed version. The bound is independent of both the width $m$ and the dimension of the principal subspace $k$. Finally, we highlight that Suzuki et al. (2020) provide generalization bounds by assuming a near low-rank structure for the weight matrix, namely that its $j$th singular value decays proportional to $j^{-\alpha}$ for some $\alpha > 1/2$. However, this condition imposes a structure quite different than what we proved in Theorem 3.

## 5 CONCLUSION

We studied the dynamics of SGD with weight decay on two-layer NNs, and proved that under a multiple-index teacher model, the first-layer weights converge to the principal subspace, i.e. the span of the weights of the teacher. This phenomenon is of particular interest when the target depends on the input along a few important directions. In this setting, we proved novel generalization bounds for SGD via uniform convergence on the low-dimensional principal subspace. Further, we proved that two-layer ReLU networks can learn a single-index target with a monotone link that has at most polynomial growth, using online SGD, with a number of samples almost linear in $d$. Thus, as an implication of low-dimensionality, we established a separation between kernel methods and trained NNs where the former suffers from the curse of dimensionality.

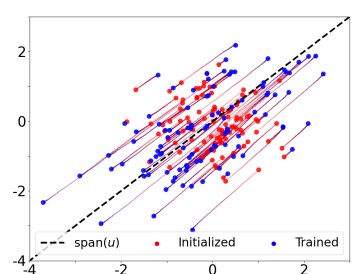

Figure 2: Neurons fail to converge to the principal subspace without weight decay, in the same experimental setup of Figure 1.

Two principal forces are responsible for the emergence of the low-dimensional structure. The main one is the linear interaction between the Gaussian input and the first-layer weights in both student and teacher models. The secondary one is the weight decay which allows SGD to avoid critical points outside of principal subspace. Figure 2 shows the convergence behavior in absence of weight decay. Understanding more precisely the range of $\lambda$ that implies convergence to the principal subspace, as well as investigating the possibility of learning multiple-index models using this convergence, are left as important directions for future studies.

### ACKNOWLEDGMENTS

The authors would like to thank Denny Wu for generating the figures, and both DW and Matthew S. Zhang for valuable feedback on the manuscript. This project was mainly funded by the CIFAR AI Catalyst grant. The authors also acknowledge the following funding sources: SP was supported by Institute of Information & communications Technology Planning & Evaluation (IITP) grant funded by the Korea government (MSIT) (No. 2019-0-00079, Artificial Intelligence Graduate School Program, Korea University). MG was supported by NSERC Grant [2022-04106], IM was supported by a Samsung grant and CIFAR AI Chairs program. Finally, MAE was supported by NSERC Grant [2019-06167] and CIFAR AI Chairs program.

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

# A  PROOFS OF SECTION 2.1

**Additional Notation.** We employ the following notation throughout the appendix. For vectors $\boldsymbol{v}$ and $\boldsymbol{u}$, we use $\langle \boldsymbol{v}, \boldsymbol{u} \rangle$ and $\boldsymbol{v} \circ \boldsymbol{u}$ to denote their Euclidean inner product and the element-wise product, and we use $\| \boldsymbol{v} \|_p$ and $\mathrm{diag}(\boldsymbol{v})$ to denote the $L_p$-norm and the diagonal matrix whose diagonal entries are $\boldsymbol{v}$. For matrices $\boldsymbol{V}$ and $\boldsymbol{W}$, we use $\langle \boldsymbol{V}, \boldsymbol{W} \rangle_{\mathrm{F}}$, $\|\boldsymbol{V}\|_{\mathrm{F}}$, and $\|\boldsymbol{V}\|_2$ to denote the Frobenius inner product, Frobenius norm, and the operator norm, respectively. For an activation function $\sigma : \mathbb{R} \to \mathbb{R}$, $\sigma'$ and $\sigma''$ denote its first and second (weak) derivatives, which are applied element-wise for vector inputs. We frequently use $\nabla \ell$ to denote $\nabla_{\boldsymbol{W}} \ell$ when it is clear from the context. We use the shorthand notation $\sigma_{\boldsymbol{a},\boldsymbol{b}}(\boldsymbol{W}\boldsymbol{x})$ to denote $\boldsymbol{a} \circ \sigma(\boldsymbol{W}\boldsymbol{x} + \boldsymbol{b})$, and similarly for $\sigma'_{\boldsymbol{a},\boldsymbol{b}}(\boldsymbol{W}\boldsymbol{x})$ and $\sigma''_{\boldsymbol{a},\boldsymbol{b}}(\boldsymbol{W}\boldsymbol{x})$. We use $\mathrm{vec}(\boldsymbol{A}) \in \mathbb{R}^{mn}$ to denote the vectorized representation of a matrix $\boldsymbol{A} \in \mathbb{R}^{m \times n}$, and $\boldsymbol{A} \otimes \boldsymbol{B}$ for the Kronecker product of two matrices $\boldsymbol{A} \in \mathbb{R}^{m \times n}$ and $\boldsymbol{B} \in \mathbb{R}^{p \times q}$; we recall that the Kronecker product is an $mp \times nq$ block matrix comprised of $m \times n$ blocks of shape $p \times q$, where block $(i,j)$ is given by $A_{ij}\boldsymbol{B}$.

In the appendix, we will prove the statements of the main text in a more general formulation. In particular, for smooth activations, we assume $\sup|\sigma'| \leq \beta_1$ and $\sup|\sigma''| \leq \beta_2$ for some $\beta_1, \beta_2 \in \mathbb{R}_+$, and we denote $\sup|\sigma| \leq \beta_0$, $\beta_0 \in (0, \infty]$. We will consider the following general case for the bias vector $\boldsymbol{b} \in \mathbb{R}^m$: $b_j \overset{\text{iid}}{\sim} \mathcal{D}_b$, such that $|b_j| \geq b^* > 0$, for some $b^* > 0$. This setting clearly covers the case of $b_j = \pm 1$ from the initialization of Assumption 3. Throughout the appendix, $C$ will denote a generic positive absolute constant (e.g. 10), whose value may change from line to line.

## A.1  PROOF OF LEMMA 1

In what follows, $\nabla^\top$ is the Jacobian matrix and $\nabla$ is the transpose of Jacobian for vector valued functions, which is the same as gradient for real-valued functions.

When $\sigma$ is twice differentiable (Assumption 2.A), standard matrix calculations yield

$$
\begin{aligned}
\nabla_{\boldsymbol{W}} \, \mathbb{E}[R(\boldsymbol{W})] &\overset{\text{(a)}}{=} \mathbb{E}[\nabla_{\boldsymbol{W}} \ell(\hat{y}(\boldsymbol{x}; \boldsymbol{W}), y)] \\
&= \mathbb{E}[\partial_1 \ell(\hat{y}(\boldsymbol{x}; \boldsymbol{W}), y) \nabla_{\boldsymbol{W}} \hat{y}(\boldsymbol{x}; \boldsymbol{W})] \\
&= \mathbb{E}\big[\partial_1 \ell(\hat{y}(\boldsymbol{x}; \boldsymbol{W}), y) \sigma'_{\boldsymbol{a},\boldsymbol{b}}(\boldsymbol{W}\boldsymbol{x}) \boldsymbol{x}^\top\big] \\
&= \mathbb{E}\big[\mathbb{E}\big[\partial_1 \ell(\hat{y}(\boldsymbol{x}; \boldsymbol{W}), y) \sigma'_{\boldsymbol{a},\boldsymbol{b}}(\boldsymbol{W}\boldsymbol{x}) \boldsymbol{x}^\top \mid \epsilon\big]\big] \\
&\overset{\text{(b)}}{=} \mathbb{E}\big[\mathbb{E}\big[\nabla_{\boldsymbol{x}}^\top \big\{\partial_1 \ell(\hat{y}(\boldsymbol{x}; \boldsymbol{W}), g_\epsilon(\boldsymbol{U}\boldsymbol{x})) \sigma'_{\boldsymbol{a},\boldsymbol{b}}(\boldsymbol{W}\boldsymbol{x})\big\} \mid \epsilon\big]\big] \\
&= \mathbb{E}\big[\partial_1^2 \ell \, \sigma'_{\boldsymbol{a},\boldsymbol{b}}(\boldsymbol{W}\boldsymbol{x}) \nabla_{\boldsymbol{x}}^\top \hat{y}(\boldsymbol{x}; \boldsymbol{W}) + \partial_1 \ell \, \nabla_{\boldsymbol{x}}^\top \sigma'_{\boldsymbol{a},\boldsymbol{b}}(\boldsymbol{W}\boldsymbol{x}) + \partial_{12}^2 \ell \, \sigma'_{\boldsymbol{a},\boldsymbol{b}}(\boldsymbol{W}\boldsymbol{x}) \nabla_{\boldsymbol{x}}^\top g_\epsilon(\boldsymbol{U}\boldsymbol{x})\big] \\
&= \mathbb{E}\big[\big\{\partial_1^2 \ell \, \sigma'_{\boldsymbol{a},\boldsymbol{b}}(\boldsymbol{W}\boldsymbol{x}) \sigma'_{\boldsymbol{a},\boldsymbol{b}}(\boldsymbol{W}\boldsymbol{x})^\top + \partial_1 \ell \, \mathrm{diag}(\sigma''_{\boldsymbol{a},\boldsymbol{b}}(\boldsymbol{W}\boldsymbol{x}))\big\} \boldsymbol{W}\big] + \\
&\qquad + \mathbb{E}\big[\partial_{12}^2 \ell \, \sigma'_{\boldsymbol{a},\boldsymbol{b}}(\boldsymbol{W}\boldsymbol{x}) \nabla g_\epsilon(\boldsymbol{U}\boldsymbol{x})^\top \boldsymbol{U}\big] \\
&= \boldsymbol{\mathcal{H}}(\boldsymbol{W})\boldsymbol{W} + \boldsymbol{\mathcal{D}}(\boldsymbol{W})\boldsymbol{U}, \tag{A.1}
\end{aligned}
$$

where (a) follows from the dominated convergence theorem and (b) follows from the Stein's lemma, and $\nabla g_\epsilon$ is the weak derivative of $g_\epsilon$ w.r.t. its inputs. Combining the above calculations with the gradient of the regularization term, with

$$
\boldsymbol{\mathcal{D}}(\boldsymbol{W}) = \mathbb{E}\big[\partial_{12}^2 \ell(\hat{y}, y)(\boldsymbol{a} \circ \sigma'(\boldsymbol{W}\boldsymbol{x} + \boldsymbol{b})) \nabla g_\epsilon^\top\big], \tag{A.2}
$$

where $\nabla g_\epsilon$ is the weak derivative of $g_\epsilon$ w.r.t. its inputs, and

$$
\boldsymbol{\mathcal{H}}(\boldsymbol{W}) = \mathbb{E}\Big[(\boldsymbol{a} \circ \sigma'(\boldsymbol{W}\boldsymbol{x} + \boldsymbol{b}))(\boldsymbol{a} \circ \sigma'(\boldsymbol{W}\boldsymbol{x} + \boldsymbol{b}))^\top\Big] + \mathbb{E}[(\hat{y} - y) \mathrm{diag}((\boldsymbol{a} \circ \sigma''(\boldsymbol{W}\boldsymbol{x} + \boldsymbol{b})))], \tag{A.3}
$$

the proof is complete for smooth activations.

For ReLU activations and $\ell$ satisfying Assumption 2.B, we introduce the following smooth approximation

$$
\sigma_\iota(z) = \frac{1}{\iota} \log(1 + e^{\iota z}), \quad \iota > 0 \,.
$$

Then we have

$$\boldsymbol{\mathcal{H}}_\iota(\boldsymbol{W}) = \mathbb{E}\big[\partial_1^2 \ell\,(\boldsymbol{a} \circ \sigma'_\iota(\boldsymbol{W}\boldsymbol{x} + \boldsymbol{b}))(\boldsymbol{a} \circ \sigma'_\iota(\boldsymbol{W}\boldsymbol{x} + \boldsymbol{b}))^\top\big] + \mathbb{E}[\partial_1 \ell\, \mathrm{diag}(\boldsymbol{a} \circ \sigma''_\iota(\boldsymbol{W}\boldsymbol{x} + \boldsymbol{b}))]$$
$$\succeq -\|\boldsymbol{a}\|_\infty \max_{1 \le j \le m} \mathbb{E}[|\sigma''_\iota(\langle \boldsymbol{w}_j, \boldsymbol{x}\rangle + b_j)|]\mathbf{I}_m.$$

As $\sigma''_\tau \ge 0$, the critical step is to show $\lim_{\iota \to \infty} \mathbb{E}[\sigma''_\iota(\langle \boldsymbol{w}, \boldsymbol{x}\rangle + b)] < \infty$, uniformly for all $\boldsymbol{w}$. Let $z = \langle \boldsymbol{w}, \boldsymbol{x}\rangle + b$. Then $z \sim \mathcal{N}(b, \|\boldsymbol{w}\|_2^2)$, and

$$\int_0^\infty \sigma''_\iota(z) \frac{e^{-\frac{(z-b)^2}{2\|\boldsymbol{w}\|_2^2}}}{\sqrt{2\pi}\|\boldsymbol{w}\|_2}\,\mathrm{d}z \le \iota \int_0^\infty \frac{e^{-\iota z - \frac{(z-b)^2}{2\|\boldsymbol{w}\|_2^2}}}{\sqrt{2\pi}\|\boldsymbol{w}\|_2}\,\mathrm{d}z$$

$$= \iota e^{-\frac{b^2}{2\|\boldsymbol{w}\|_2^2} + \frac{(\iota\|\boldsymbol{w}\|_2 - \frac{b}{\|\boldsymbol{w}\|_2})^2}{2}} \int_0^\infty \frac{e^{-\frac{1}{2}(\frac{z}{\|\boldsymbol{w}\|_2} + \iota\|\boldsymbol{w}\|_2 - \frac{b}{\|\boldsymbol{w}\|_2})^2}}{\sqrt{2\pi}\|\boldsymbol{w}\|_2}\,\mathrm{d}z.$$

$$= \iota e^{-\frac{b^2}{2\|\boldsymbol{w}\|_2^2} + \frac{(\iota\|\boldsymbol{w}\|_2 - \frac{b}{\|\boldsymbol{w}\|_2})^2}{2}} (1 - \Phi(\iota\|\boldsymbol{w}\|_2 - \frac{b}{\|\boldsymbol{w}\|_2})).$$

$$\overset{(a)}{\le} \frac{\iota e^{-\frac{b^2}{2\|\boldsymbol{w}\|_2^2}}}{\sqrt{2\pi}\|\boldsymbol{w}\|_2(\iota - \frac{b}{\|\boldsymbol{w}\|_2^2})}$$

$$\overset{(b)}{\le} \sqrt{\frac{2}{\pi}} \frac{e^{\frac{-b^2}{2\|\boldsymbol{w}\|_2^2}}}{\|\boldsymbol{w}\|_2}$$

$$\overset{(c)}{\le} \frac{1}{|b|}\sqrt{\frac{2}{e\pi}},$$

where (a) follows from the Gaussian tail bound $1 - \Phi(x) \le \frac{e^{-x^2/2}}{\sqrt{2\pi}x}$, where $\Phi$ is the standard Gaussian CDF; (b) holds for large enough $\iota$; and (c) holds by considering supremum over $\|\boldsymbol{w}\|_2$. Thus $\mathbb{E}[\sigma''_\iota(\langle \boldsymbol{w}_j, \boldsymbol{x}\rangle + b_j)] \le \frac{2}{|b_j|}\sqrt{\frac{2}{e\pi}}$ and consequently,

$$\frac{-2\|\boldsymbol{a}\|_\infty}{b^*}\sqrt{\frac{2}{e\pi}}\mathbf{I}_m \preceq \boldsymbol{\mathcal{H}}_\iota(\boldsymbol{W}) \preceq \left(\|\boldsymbol{a}\|_2^2 + \frac{2\|\boldsymbol{a}\|_\infty}{b^*}\sqrt{\frac{2}{e\pi}}\right)\mathbf{I}_m$$

where $b^* = \min_{1 \le j \le m}|b_j|$. Moreover, as $\sigma'_\iota(\boldsymbol{W}\boldsymbol{x} + \boldsymbol{b})$ converges a.s. (i.e. except when $\langle \boldsymbol{w}_j, \boldsymbol{x}\rangle + b_j = 0$ for some $j$) to $\sigma'(\boldsymbol{W}\boldsymbol{x} + \boldsymbol{b})$, by the dominated convergence theorem,

$$\nabla R(\boldsymbol{W}) = \lim_{\iota \to \infty} \boldsymbol{\mathcal{H}}_\iota(\boldsymbol{W})\boldsymbol{W} + \lim_{\iota \to \infty} \boldsymbol{\mathcal{D}}_\iota(\boldsymbol{W})\boldsymbol{U}$$

We can immediately observe from the dominated convergence theorem that $\boldsymbol{\mathcal{D}}_\iota(\boldsymbol{W}) \to \boldsymbol{\mathcal{D}}(\boldsymbol{W})$ as $\iota \to \infty$ with $\boldsymbol{\mathcal{D}}(\boldsymbol{W})$ given in (A.2). Moreover, we let $\boldsymbol{\mathcal{H}}(\boldsymbol{W}) = \lim_{\iota \to \infty} \boldsymbol{\mathcal{H}}_\iota(\boldsymbol{W})$, and observe that

$$\frac{-2\|\boldsymbol{a}\|_\infty}{b^*}\sqrt{\frac{2}{e\pi}}\mathbf{I}_m \preceq \boldsymbol{\mathcal{H}}(\boldsymbol{W}) \preceq \left(\|\boldsymbol{a}\|_2^2 + \frac{2\|\boldsymbol{a}\|_\infty}{b^*}\sqrt{\frac{2}{e\pi}}\right)\mathbf{I}_m. \tag{A.4}$$

This finishes the proof of Lemma 1. $\qquad\square$

In the case of smooth activations (Assumption 2.A), we have the following bounds.

**Lemma 7.** *Let* $R(\boldsymbol{W}) := \mathbb{E}[\ell(\hat{y}(\boldsymbol{x}; \boldsymbol{W}, \boldsymbol{a}, \boldsymbol{b}), y)]$ *be the unregularized population risk. Under Assumptions 1&2.A we have*

$$-\beta_2\|\boldsymbol{a}\|_\infty\sqrt{2R(\boldsymbol{W})}\mathbf{I}_m \preceq \boldsymbol{\mathcal{H}}(\boldsymbol{W}) \preceq \left\{\beta_1^2\|\boldsymbol{a}\|_2^2 + \beta_2\|\boldsymbol{a}\|_\infty\sqrt{2R(\boldsymbol{W})}\right\}\mathbf{I}_m. \tag{A.5}$$

**Proof.** Assumption 2.A requires $\ell(\hat{y}, y) = \frac{1}{2}(\hat{y} - y)^2$. Hence by definition of $\boldsymbol{\mathcal{H}}$,

$$\boldsymbol{\mathcal{H}}(\boldsymbol{W}) = \mathbb{E}\big[\sigma'_{\boldsymbol{a},\boldsymbol{b}}(\boldsymbol{W}\boldsymbol{x})\sigma'_{\boldsymbol{a},\boldsymbol{b}}(\boldsymbol{W}\boldsymbol{x})^\top\big] + \mathbb{E}\big[(\hat{y}(\boldsymbol{x}; \boldsymbol{W}) - y)\,\mathrm{diag}(\sigma''_{\boldsymbol{a},\boldsymbol{b}}(\boldsymbol{W}\boldsymbol{x}))\big].$$

The first term is positive semi-definite and it can be easily bounded:

$$0 \leq \boldsymbol{v}^\top \mathbb{E}\big[\sigma'_{\boldsymbol{a},\boldsymbol{b}}(\boldsymbol{W}\boldsymbol{x})\sigma'_{\boldsymbol{a},\boldsymbol{b}}(\boldsymbol{W}\boldsymbol{x})^\top\big]\boldsymbol{v} \leq \mathbb{E}\big[\|\sigma'_{\boldsymbol{a},\boldsymbol{b}}(\boldsymbol{W}\boldsymbol{x})^2\|_2\big]\|\boldsymbol{v}\|_2^2 \leq \beta_1^2\|\boldsymbol{a}\|_2^2\|\boldsymbol{v}\|_2^2$$

for an arbitrary vector $\boldsymbol{v} \in \mathbb{R}^m$. For the second term, we have

$$-\beta_2\|\boldsymbol{a}\|_\infty \mathbb{E}[|\hat{y} - y|]\mathbf{I}_m \preceq \mathbb{E}\big[(\hat{y}(\boldsymbol{x};\boldsymbol{W}) - y)\operatorname{diag}(\sigma''_{\boldsymbol{a},\boldsymbol{b}}(\boldsymbol{W}\boldsymbol{x}))\big] \preceq \beta_2\|\boldsymbol{a}\|_\infty \mathbb{E}[|\hat{y} - y|]\mathbf{I}_m$$

and $\mathbb{E}[|\hat{y}(\boldsymbol{x};\boldsymbol{W}) - y|] \leq \sqrt{2R(\boldsymbol{W})}$ by Jensen's inequality. $\qquad\square$

## A.2 PROOF OF PROPOSITION 2

In order to present the proof of Proposition 2, we need a uniform control over the eigenspectrum of $\mathcal{H}(\boldsymbol{W})$. In the case of ReLU (Assumption 2.B), this follows from (A.4). For smooth activations (Assumption 2.A), we need to establish the boundedness of $R(\boldsymbol{W}^t)$ along the trajectory. The first step towards achieving this goal is to obtain an estimate of $\mathcal{R}_\lambda(\boldsymbol{W}^{t+1}) - \mathcal{R}_\lambda(\boldsymbol{W}^t)$, which depends on the local smoothness of $\mathcal{R}_\lambda(\boldsymbol{W})$.

We denote by $\nabla^2 \mathcal{R}_\lambda(\boldsymbol{W})$ the full Hessian of the risk function, an $md \times md$ matrix comprised of $d \times d$ blocks $(\nabla^2_{\boldsymbol{w}_i,\boldsymbol{w}_j}\mathcal{R}_\lambda(\boldsymbol{W}))_{1 \leq i,j \leq m}$ where $(\nabla^2_{\boldsymbol{w}_i,\boldsymbol{w}_j}\mathcal{R}_\lambda(\boldsymbol{W}))_{pq} = \frac{\partial^2 \mathcal{R}_\lambda(\boldsymbol{W})}{\partial(\boldsymbol{w}_i)_p \partial(\boldsymbol{w}_j)_q}$.

**Lemma 8.** *Let* $R(\boldsymbol{W}) := \mathbb{E}[\ell(\hat{y}(\boldsymbol{x};\boldsymbol{W},\boldsymbol{a},\boldsymbol{b}),y)]$ *be the unregularized population risk. Under Assumptions 1&2.A, we have the following estimate for the eigenspectrum of the Hessian*

$$\Big(\lambda - \beta_2\|\boldsymbol{a}\|_\infty \sqrt{6R(\boldsymbol{W})}\Big)\mathbf{I}_{md} \preceq \nabla^2 \mathcal{R}_\lambda(\boldsymbol{W}) \preceq \Big(\lambda + \beta_1^2\|\boldsymbol{a}\|_2^2 + \beta_2\|\boldsymbol{a}\|_\infty \sqrt{6R(\boldsymbol{W})}\Big)\mathbf{I}_{md}. \tag{A.6}$$

**Proof.** By the chain rule for derivatives, we have

$$\nabla^2_{\boldsymbol{w}_i,\boldsymbol{w}_j} R(\boldsymbol{W}) = \mathbb{E}\big[\{a_i a_j \sigma'(\langle\boldsymbol{w}_i,\boldsymbol{x}\rangle + b_i)\sigma'(\langle\boldsymbol{w}_j,\boldsymbol{x}\rangle + b_j) + (\hat{y}(\boldsymbol{x};\boldsymbol{W}) - y)\delta_{ij}a_i\sigma''(\langle\boldsymbol{w}_i,\boldsymbol{x}\rangle + b_i)\}\boldsymbol{x}\boldsymbol{x}^\top\big],$$

where $\delta_{ij}$ is the Kronecker delta. As a result, in matrix form, the Hessian reads

$$\nabla^2 R(\boldsymbol{W}) = \mathbb{E}\big[\sigma'_{\boldsymbol{a},\boldsymbol{b}}(\boldsymbol{W}\boldsymbol{x})\sigma'_{\boldsymbol{a},\boldsymbol{b}}(\boldsymbol{W}\boldsymbol{x})^\top \otimes \boldsymbol{x}\boldsymbol{x}^\top\big] + \mathbb{E}\big[(\hat{y}(\boldsymbol{x};\boldsymbol{W}) - y)\operatorname{diag}(\sigma''_{\boldsymbol{a},\boldsymbol{b}}(\boldsymbol{W}\boldsymbol{x})) \otimes \boldsymbol{x}\boldsymbol{x}^\top\big].$$

The first term is a positive semi-definite matrix with bounded spectral norm; indeed, for any $\boldsymbol{V} \in \mathbb{R}^{m \times d}$

$$0 \leq \operatorname{vec}(\boldsymbol{V})^\top \mathbb{E}\big[\sigma'_{\boldsymbol{a},\boldsymbol{b}}(\boldsymbol{W}\boldsymbol{x})\sigma'_{\boldsymbol{a},\boldsymbol{b}}(\boldsymbol{W}\boldsymbol{x})^\top \otimes \boldsymbol{x}\boldsymbol{x}^\top\big]\operatorname{vec}(\boldsymbol{V}) = \mathbb{E}\big[\langle\sigma'_{\boldsymbol{a},\boldsymbol{b}}(\boldsymbol{W}\boldsymbol{x})\boldsymbol{x}^\top,\boldsymbol{V}\rangle_{\mathrm{F}}^2\big] =$$
$$= \mathbb{E}\big[\langle\sigma'_{\boldsymbol{a},\boldsymbol{b}}(\boldsymbol{W}\boldsymbol{x}),\boldsymbol{V}\boldsymbol{x}\rangle^2\big] \leq \mathbb{E}\big[\|\sigma'_{\boldsymbol{a},\boldsymbol{b}}(\boldsymbol{W}\boldsymbol{x})\|_2^2\|\boldsymbol{V}\boldsymbol{x}\|_2^2\big] \leq \beta_1^2\|\boldsymbol{a}\|_2^2\|\boldsymbol{V}\|_{\mathrm{F}}^2.$$

The second term is bounded by the following:

$$\big|\boldsymbol{v}^\top \mathbb{E}\big[(\hat{y}(\boldsymbol{x};\boldsymbol{W}) - y)a_j\sigma''(\langle\boldsymbol{w}_j,\boldsymbol{x}\rangle)\boldsymbol{x}\boldsymbol{x}^\top\big]\boldsymbol{v}\big| = \big|\mathbb{E}\big[(\hat{y}(\boldsymbol{x};\boldsymbol{W}) - y)a_j\sigma''(\langle\boldsymbol{w}_j,\boldsymbol{x}\rangle + b_j)(\boldsymbol{x}^\top\boldsymbol{v})^2\big]\big|$$
$$\leq \beta_2\|\boldsymbol{a}\|_\infty \mathbb{E}\big[(\hat{y}(\boldsymbol{x};\boldsymbol{W}) - y)^2\big]^{\frac{1}{2}}\mathbb{E}\big[(\boldsymbol{x}^\top\boldsymbol{v})^4\big]^{\frac{1}{2}} = \beta_2\|\boldsymbol{a}\|_\infty\sqrt{2R(\boldsymbol{W})}\sqrt{3\|\boldsymbol{v}\|^4},$$

for all $1 \leq j \leq m$ and for any $\boldsymbol{v} \in \mathbb{R}^d$, which completes the proof. $\qquad\square$

**Lemma 9.** *In the same setting as the previous Lemma, for any* $\boldsymbol{W},\boldsymbol{W}' \in \mathbb{R}^{m \times d}$ *we have*

$$\mathcal{R}_\lambda(\boldsymbol{W}') \leq \mathcal{R}_\lambda(\boldsymbol{W}) + \langle\nabla\mathcal{R}_\lambda(\boldsymbol{W}),\boldsymbol{W}' - \boldsymbol{W}\rangle_F + \frac{(\lambda + \beta_1^2\|\boldsymbol{a}\|_2^2 + 2\beta_2\|\boldsymbol{a}\|_\infty\sqrt{3R(\boldsymbol{W})})}{2}\|\boldsymbol{W}' - \boldsymbol{W}\|_F^2$$
$$+ \frac{\sqrt{6}\beta_2\beta_1\|\boldsymbol{a}\|_\infty\|\boldsymbol{a}\|_2}{2}\|\boldsymbol{W}' - \boldsymbol{W}\|_F^3.$$

**Proof.** By Taylor's theorem,

$$\mathcal{R}_\lambda(\boldsymbol{W}') = \mathcal{R}_\lambda(\boldsymbol{W}) + \langle\nabla\mathcal{R}_\lambda(\boldsymbol{W}),\boldsymbol{W}' - \boldsymbol{W}\rangle_{\mathrm{F}} + \frac{1}{2}\langle\operatorname{vec}(\boldsymbol{W}' - \boldsymbol{W}),\nabla^2\mathcal{R}_\lambda(\boldsymbol{W}_\alpha)\operatorname{vec}(\boldsymbol{W}' - \boldsymbol{W})\rangle \tag{A.7}$$

for some $\boldsymbol{W}_\alpha = \boldsymbol{W} + \alpha(\boldsymbol{W}' - \boldsymbol{W})$, $\alpha \in [0, 1]$. The last term can be estimated using Lemma 8:

$$\langle \mathrm{vec}(\boldsymbol{W}' - \boldsymbol{W}), \nabla^2 \mathcal{R}_\lambda(\boldsymbol{W}_\alpha) \, \mathrm{vec}(\boldsymbol{W}' - \boldsymbol{W}) \rangle \leq \|\nabla^2 \mathcal{R}_\lambda(\boldsymbol{W}_\alpha)\|_2 \|\boldsymbol{W}' - \boldsymbol{W}\|_F^2 \leq$$
$$\leq \left( \lambda + \beta_1^2 \|\boldsymbol{a}\|_2^2 + \beta_2 \|\boldsymbol{a}\|_\infty \sqrt{6R(\boldsymbol{W}_\alpha)} \right) \|\boldsymbol{W}' - \boldsymbol{W}\|_F^2, \qquad (A.8)$$

Next, we provide an upper bound for $R(\boldsymbol{W}_\alpha)$:

$$2(R(\boldsymbol{W}_\alpha) - R(\boldsymbol{W})) = \mathbb{E}\big[(\hat{y}(\boldsymbol{x}; \boldsymbol{W}_\alpha) - y)^2 - (\hat{y}(\boldsymbol{x}; \boldsymbol{W}) - y)^2\big]$$
$$= \mathbb{E}\Big[(\hat{y}(\boldsymbol{x}; \boldsymbol{W}_\alpha) - \hat{y}(\boldsymbol{x}; \boldsymbol{W}))^2\Big] + 2\,\mathbb{E}[(\hat{y}(\boldsymbol{x}; \boldsymbol{W}_\alpha) - \hat{y}(\boldsymbol{x}; \boldsymbol{W}))(\hat{y}(\boldsymbol{x}; \boldsymbol{W}) - y)]$$
$$\leq \mathbb{E}\Big[(\hat{y}(\boldsymbol{x}; \boldsymbol{W}_\alpha) - \hat{y}(\boldsymbol{x}; \boldsymbol{W}))^2\Big] + 2\,\mathbb{E}\Big[(\hat{y}(\boldsymbol{x}; \boldsymbol{W}_\alpha) - \hat{y}(\boldsymbol{x}; \boldsymbol{W}))^2\Big]^{\frac{1}{2}} \sqrt{2R(\boldsymbol{W})}$$
$$\overset{(*)}{\leq} \beta_1^2 \|\boldsymbol{a}\|_2^2 \|\boldsymbol{W}_\alpha - \boldsymbol{W}\|_F^2 + 2\beta_1 \|\boldsymbol{a}\|_2 \|\boldsymbol{W}_\alpha - \boldsymbol{W}\|_F \sqrt{2R(\boldsymbol{W})}$$
$$\leq \beta_1^2 \|\boldsymbol{a}\|_2^2 \|\boldsymbol{W}' - \boldsymbol{W}\|_F^2 + 2\beta_1 \|\boldsymbol{a}\|_2 \|\boldsymbol{W}' - \boldsymbol{W}\|_F \sqrt{2R(\boldsymbol{W})}$$
$$\leq 2\beta_1^2 \|\boldsymbol{a}\|_2^2 \|\boldsymbol{W}' - \boldsymbol{W}\|_F^2 + 2R(\boldsymbol{W}), \qquad (A.9)$$

where the last inequality follows from Young's inequality and $(*)$ is due to the estimate below:

$$\mathbb{E}\big[(\hat{y}(x; \boldsymbol{W}_\alpha)) - \hat{y}(\boldsymbol{x}; \boldsymbol{W}))^2\big] = \mathbb{E}\left[ \left( \sum_{j=1}^m a_j \{ \sigma(\langle (\boldsymbol{w}_\alpha)_j, \boldsymbol{x} \rangle + b_j) - \sigma(\langle \boldsymbol{w}_j, \boldsymbol{x} \rangle + b_j) \} \right)^2 \right]$$
$$\leq \sum_{j=1}^m a_j^2 \, \mathbb{E}\Big[ (\sigma(\langle (\boldsymbol{w}_\alpha)_j, \boldsymbol{x} \rangle + b_j) - \sigma(\langle \boldsymbol{w}_j, \boldsymbol{x} \rangle + b_j))^2 \Big]$$
$$\leq \sum_{j=1}^m a_j^2 \beta_1^2 \, \mathbb{E}\left[ \sum_{j=1}^m \langle (\boldsymbol{w}_\alpha)_j - \boldsymbol{w}_j, \boldsymbol{x} \rangle^2 \right]$$
$$= \beta_1^2 \|\boldsymbol{a}\|_2^2 \|\boldsymbol{W}_\alpha - \boldsymbol{W}\|_F^2., \qquad (A.10)$$

Plugging (A.9) into (A.8) completes the proof. $\qquad\square$

In order to prove Proposition 2 we will additionally need a bound on the norm of the iterates $\{\boldsymbol{W}^t\}_{t \geq 0}$ of the trajectory.

**Lemma 10.** *Let $\{\boldsymbol{W}^t\}_{t \geq 0}$ be the sequence of PGD iterates* (2.5)*. Suppose that there exists $T \geq 1$ such that $\mathcal{R}_\lambda(\boldsymbol{W}^t)$ is non-increasing in $t = 0, 1, \ldots, T$. Under Assumptions 1&2.A, for $\eta = m\tilde{\eta}$*

$$\lambda > \beta_2 \|\boldsymbol{a}\|_\infty \sqrt{2\mathcal{R}_\lambda(\boldsymbol{W}^0)} \quad \text{and} \quad \eta < \left( \lambda + \beta_1^2 \|\boldsymbol{a}\|_2^2 + \beta_2 \|\boldsymbol{a}\|_\infty \sqrt{2\mathcal{R}_\lambda(\boldsymbol{W}^0)} \right)^{-1}, \quad (A.11)$$

*we have*

$$\|\boldsymbol{W}^t\|_F \leq (1 - \tilde{\eta}\gamma)^t \|\boldsymbol{W}^0\|_F + \frac{\beta_1 m \|\boldsymbol{a}\|_2 \, \mathbb{E}\big[\|\nabla g_\epsilon^\top \boldsymbol{U}\|_2\big]}{\gamma} \qquad \forall\, t \leq T, \qquad (A.12)$$

*where $\gamma/m = \lambda - \beta_2 \|\boldsymbol{a}\|_\infty \sqrt{2\mathcal{R}_\lambda(\boldsymbol{W}^0)}$.*

**Proof.** The update rule of PGD reads

$$\boldsymbol{W}^{t+1} = (\mathbf{I}_m - \eta(\boldsymbol{\mathcal{H}}(\boldsymbol{W}^t) + \lambda \mathbf{I}_m))\boldsymbol{W}^t - \eta \boldsymbol{\mathcal{D}}(\boldsymbol{W}^t)\boldsymbol{U}. \qquad (A.13)$$

Since $R(\boldsymbol{W}^t) \leq \mathcal{R}_\lambda(\boldsymbol{W}^t) \leq \mathcal{R}_\lambda(\boldsymbol{W}^0)$, for all $t \leq T$, we obtain from Lemma 7

$$\left( \lambda - \beta_2 \|\boldsymbol{a}\|_\infty \sqrt{2\mathcal{R}_\lambda(\boldsymbol{W}^0)} \right) \mathbf{I}_m \preceq \boldsymbol{\mathcal{H}}(\boldsymbol{W}^t) + \lambda \mathbf{I}_m \preceq \left( \lambda + \beta_1^2 \|\boldsymbol{a}\|_2^2 + \beta_2 \|\boldsymbol{a}\|_\infty \sqrt{2\mathcal{R}_\lambda(\boldsymbol{W}^0)} \right) \mathbf{I}_m,$$
$$(A.14)$$

and for $\eta$ as in (A.11) we have

$$0 \preceq \mathbf{I}_m - \eta(\boldsymbol{\mathcal{H}}(\boldsymbol{W}^t) + \lambda \mathbf{I}_m) \preceq (1 - \tilde{\eta}\gamma)\mathbf{I}_m.$$

Therefore,

$$\|\boldsymbol{W}^{t+1}\|_{\mathrm{F}} \leq (1 - \tilde{\eta}\gamma)\|\boldsymbol{W}^t\|_{\mathrm{F}} + m\tilde{\eta}\|\boldsymbol{\mathcal{D}}(\boldsymbol{W}^t)\boldsymbol{U}\|_{\mathrm{F}}. \tag{A.15}$$

and we can easily bound the last term

$$\|\boldsymbol{\mathcal{D}}(\boldsymbol{W}^t)\boldsymbol{U}\|_{\mathrm{F}} = \|\mathbb{E}\big[\sigma'_{\boldsymbol{a},\boldsymbol{b}}(\boldsymbol{W}^t\boldsymbol{x})\nabla g_\epsilon^\top \boldsymbol{U}\big]\|_{\mathrm{F}} \leq \mathbb{E}\big[\|\sigma'_{\boldsymbol{a},\boldsymbol{b}}(\boldsymbol{W}^t\boldsymbol{x})\nabla g_\epsilon^\top \boldsymbol{U}\|_{\mathrm{F}}\big]$$
$$\leq \beta_1\|\boldsymbol{a}\|_2\,\mathbb{E}\big[\|\nabla g_\epsilon^\top \boldsymbol{U}\|_2\big]. \tag{A.16}$$

The statement of the lemma then follows by plugging the above bound back into (A.15) and expanding the recursion. $\square$

We are now ready to prove Proposition 2.

**Proof. [Proposition 2]** Along the proof, we set $\varrho := \lambda + \beta_1^2\|\boldsymbol{a}\|_2^2 + \beta_2\|\boldsymbol{a}\|_\infty\sqrt{2\mathcal{R}_\lambda(\boldsymbol{W}^0)}$. Notice that in the setting of Proposition 2, we have $\varrho \asymp \frac{\tilde{\lambda}}{m} = \lambda$. We will consider the event where $\|\boldsymbol{W}^0\|_{\mathrm{F}} \leq \sqrt{2m}$, which happens with probability at least $1 - \exp(-Cmd)$.

*Smooth activations.* We begin by considering the following condition

$$\lambda \geq \frac{\gamma}{m} + \beta_2\|\boldsymbol{a}\|_\infty\sqrt{2\mathcal{R}_\lambda(\boldsymbol{W}^0)}. \tag{A.17}$$

Solving the quadratic equation to find the range of $\lambda$ where the above condition is satisfied yields

$$\lambda \geq \frac{\gamma}{m} + \frac{\beta_2^2\|\boldsymbol{a}\|_\infty^2\|\boldsymbol{W}^0\|_{\mathrm{F}}^2}{2} + \sqrt{\frac{\beta_2^4\|\boldsymbol{a}\|_\infty^4\|\boldsymbol{W}^0\|_{\mathrm{F}}^4}{4} + \frac{\gamma}{m}\beta_2^2\|\boldsymbol{a}\|_\infty^2\|\boldsymbol{W}^0\|_{\mathrm{F}}^2 + 2\beta_2^2\|\boldsymbol{a}\|_\infty^2 R(\boldsymbol{W}^0)}$$

In the setting of Proposition 2 with $\|\boldsymbol{a}\|_\infty \lesssim m^{-1}$, the above simplifies to

$$\lambda \geq \frac{1 + \gamma + \sqrt{1 + 2\gamma + 2R(\boldsymbol{W}^0)}}{m},$$

which is satisfied in Proposition 2. Thus we will assume (A.17) holds in the rest of the proof for smooth activations.

We will use induction on $t$ to show that $\mathcal{R}_\lambda(\boldsymbol{W}^t)$ is non-increasing. The base case is trivial, and assuming the claim holds up to time $t$, Lemma 9 implies

$$\mathcal{R}_\lambda(\boldsymbol{W}^{t+1}) \leq \mathcal{R}_\lambda(\boldsymbol{W}^t) - \eta\|\nabla\mathcal{R}_\lambda(\boldsymbol{W}^t)\|_{\mathrm{F}}^2 + C\eta^2\varrho\|\nabla\mathcal{R}_\lambda(\boldsymbol{W}^t)\|_{\mathrm{F}}^2$$
$$+ C\beta_1\beta_2\|\boldsymbol{a}\|_\infty\|\boldsymbol{a}\|_2\eta^3\|\nabla\mathcal{R}_\lambda(\boldsymbol{W}^t)\|_{\mathrm{F}}^3. \tag{A.18}$$

Moreover, we have the following upper bound on gradient norm

$$\|\nabla\mathcal{R}_\lambda(\boldsymbol{W}^t)\|_{\mathrm{F}} \overset{(a)}{\leq} \|\boldsymbol{\mathcal{H}}(\boldsymbol{W}^t) + \lambda\mathbf{I}_m\|_2\|\boldsymbol{W}^t\|_{\mathrm{F}} + \|\boldsymbol{\mathcal{D}}(\boldsymbol{W}^t)\boldsymbol{U}\|_{\mathrm{F}}$$
$$\overset{(b)}{\lesssim} \varrho\|\boldsymbol{W}^0\|_{\mathrm{F}} + \beta_1\varsigma\|\boldsymbol{a}\|_2\left(\frac{m\varrho}{\gamma} + 1\right) \tag{A.19}$$

where (a) follows from the closed form of the gradient (2.6) and (b) follows from (A.12), (A.14) and (A.16). Thus with a choice of $\eta \lesssim \left(\varrho\|\boldsymbol{W}^0\|_{\mathrm{F}}\|\boldsymbol{a}\|_2 + \beta_1\varsigma\|\boldsymbol{a}\|_2^2(\frac{m\varrho}{\gamma} + 1)\right)^{-1}$ we have $\eta\|\boldsymbol{a}\|_2\|\nabla\mathcal{R}_\lambda(\boldsymbol{W})\|_{\mathrm{F}} \leq 1$. Consequently, with $\eta\|\boldsymbol{a}\|_2\|\nabla\mathcal{R}_\lambda(\boldsymbol{W})\|_{\mathrm{F}} \leq 1$,

$$\mathcal{R}_\lambda(\boldsymbol{W}^{t+1}) \leq \mathcal{R}_\lambda(\boldsymbol{W}^t) - \eta\|\nabla\mathcal{R}_\lambda(\boldsymbol{W}^t)\|_{\mathrm{F}}^2 + C\eta^2(\varrho + \beta_1\beta_2\|\boldsymbol{a}\|_\infty)\|\nabla\mathcal{R}_\lambda(\boldsymbol{W}^t)\|_{\mathrm{F}}^2.$$

Therefore, with a choice of $\eta \lesssim (\varrho + \beta_1\beta_2\|\boldsymbol{a}\|_\infty)^{-1}$, we will have

$$\mathcal{R}_\lambda(\boldsymbol{W}^{t+1}) \leq \mathcal{R}_\lambda(\boldsymbol{W}^t) - C\eta\|\nabla\mathcal{R}_\lambda(\boldsymbol{W}^t)\|_{\mathrm{F}}^2.$$

As $\|\boldsymbol{a}\|_2 \lesssim m^{-1/2}$, $\|\boldsymbol{W}^0\|_{\mathrm{F}} \lesssim m^{1/2}$, and $\varrho \asymp \frac{\lambda}{m}$, we can simplify the two conditions to $\eta \lesssim m(\tilde{\lambda} + \frac{\tilde{\lambda}\varsigma}{\gamma} + \varsigma)^{-1}$ and $\eta \lesssim m\tilde{\lambda}^{-1}$ respectively, hence proof of the induction is complete.

Finally, recall the update rule of PGD (A.13):

$$\boldsymbol{W}^{t+1} = (\mathbf{I}_m - \eta(\boldsymbol{\mathcal{H}}(\boldsymbol{W}^t) + \lambda\mathbf{I}_m))\boldsymbol{W}^t - \eta\boldsymbol{\mathcal{D}}(\boldsymbol{W}^t)\boldsymbol{U}.$$

By projecting each row of this recursion onto the orthogonal complement of the principal subspace, we have

$$\boldsymbol{W}_{\perp}^{t+1} = (\mathbf{I}_m - \eta(\boldsymbol{\mathcal{H}}(\boldsymbol{W}^t) + \lambda\mathbf{I}_m))\boldsymbol{W}_{\perp}^{t} \ .$$

Again from (A.14), we have that for $\eta < \varrho^{-1}$,

$$0 \preceq \mathbf{I}_m - \eta(\boldsymbol{\mathcal{H}}(\boldsymbol{W}) + \lambda\mathbf{I}_m) \preceq (1 - \tilde{\eta}\gamma)\mathbf{I}_m \ ,$$

therefore

$$\|\boldsymbol{W}_{\perp}^{t+1}\|_{\mathrm{F}} \leq (1 - \eta\gamma)\|\boldsymbol{W}_{\perp}^{t}\|_{\mathrm{F}}.$$

We have shown (2.7) and the proof for smooth activations is complete.

*ReLU activation.* By (A.4), for $\lambda \geq \gamma + \frac{2\|\boldsymbol{a}\|_{\infty}}{b^*}\sqrt{\frac{2}{e\pi}}$ and $\eta < (\lambda + \|\boldsymbol{a}\|_2^2 + \frac{2\|\boldsymbol{a}\|_{\infty}}{b^*}\sqrt{\frac{2}{e\pi}})^{-1}$ we have

$$0 \preceq \mathbf{I}_m - \eta(\boldsymbol{\mathcal{H}}(\boldsymbol{W}) + \lambda\mathbf{I}_m) \preceq (1 - \eta\gamma)\mathbf{I}_m.$$

Notice that in the setting of Proposition 2, $\|\boldsymbol{a}\|_2^2 + \|\boldsymbol{a}\|_{\infty} \lesssim \lambda$, thus $\eta \lesssim \lambda^{-1}$ suffices for the above inequality to hold. The rest of the proof follows similarly to the smooth case. $\qquad\square$

# B    PROOFS OF SECTION 3

We begin by characterizing the tail behavior of the stochastic gradient noise in the SGD updates (3.1) through the following lemma.

**Lemma 11.** *For any fixed $\boldsymbol{W} \in \mathbb{R}^{m \times k}$, let*

$$\boldsymbol{\Gamma} := \nabla\ell(\hat{y}(\boldsymbol{x}; \boldsymbol{W}), y) - \mathbb{E}[\nabla\ell(\hat{y}(\boldsymbol{x}; \boldsymbol{W}), y)]$$

*denote the zero-mean stochastic noise in the gradient of the loss function $\ell$ when $(\boldsymbol{x}, y)$ are generated according to Assumption 1, and recall that*

$$\nabla\ell(\hat{y}(\boldsymbol{x}; \boldsymbol{W}), y) = \partial_1\ell(\hat{y}(\boldsymbol{x}; \boldsymbol{W}), y)\sigma'_{\boldsymbol{a},\boldsymbol{b}}(\boldsymbol{W}\boldsymbol{x})\boldsymbol{x}^{\top}.$$

*Suppose $\sup_{\hat{y},y}|\partial_1\ell(\hat{y}, y)| \leq \varkappa$. Then for any $\boldsymbol{V} \in \mathbb{R}^{m \times d}$, the zero-mean random variable $\langle\boldsymbol{V}, \boldsymbol{\Gamma}\rangle_F$ is $C\beta_1\varkappa\|\boldsymbol{a}\|_2\|\boldsymbol{V}\|_F$-sub-Gaussian.*

**Proof.** We use the shorthand notation $\nabla\ell := \nabla_{\boldsymbol{W}}\ell(\boldsymbol{W}\boldsymbol{x}, y)$ and $\nabla R := \nabla_{\boldsymbol{W}}R(\boldsymbol{W})$. We compute the following

$$\mathbb{E}[|\langle\boldsymbol{V}, \nabla\ell - \nabla R\rangle_{\mathrm{F}}|^p]^{\frac{1}{p}} \overset{(a)}{\leq} \mathbb{E}[|\langle\boldsymbol{V}, \nabla\ell\rangle_{\mathrm{F}}|^p]^{\frac{1}{p}} + \mathbb{E}[|\langle\boldsymbol{V}, \nabla R\rangle_{\mathrm{F}}|^p]^{\frac{1}{p}}$$

$$\overset{(b)}{\leq} 2\,\mathbb{E}[|\langle\boldsymbol{V}, \nabla\ell\rangle_{\mathrm{F}}|^p]^{\frac{1}{p}}$$

$$\leq 2\varkappa\,\mathbb{E}\Big[|\langle\boldsymbol{V}, \sigma'_{\boldsymbol{a},\boldsymbol{b}}(\boldsymbol{W}\boldsymbol{x})\boldsymbol{x}^{\top}\rangle_{\mathrm{F}}|^{2p}\Big]^{\frac{1}{2p}}.$$

where (a) and (b) follow from the Minkowski and Jensen inequalities respectively. Furthermore, we have

$$\mathbb{E}\Big[|\langle\boldsymbol{V}, \sigma'_{\boldsymbol{a},\boldsymbol{b}}(\boldsymbol{W}\boldsymbol{x})\boldsymbol{x}^{\top}\rangle_{\mathrm{F}}|^{2p}\Big]^{\frac{1}{2p}} = \mathbb{E}\big[|\langle\boldsymbol{V}\boldsymbol{x}, \sigma'_{\boldsymbol{a},\boldsymbol{b}}(\boldsymbol{W}\boldsymbol{x})\rangle|^{2p}\big]^{\frac{1}{2p}}$$

$$\leq \beta_1\|\boldsymbol{a}\|_2\,\mathbb{E}\Big[\|\boldsymbol{V}\boldsymbol{x}\|_2^{2p}\Big]^{\frac{1}{2p}}$$

$$\leq \beta_1\|\boldsymbol{a}\|_2(\|\boldsymbol{V}\|_{\mathrm{F}} + C\|\boldsymbol{V}\|_2\sqrt{p}),$$

where the last inequality follows from Gaussianity of $\boldsymbol{V}\boldsymbol{x}$ and Lemma 32. Hence

$$\mathbb{E}[|\langle\boldsymbol{V}, \nabla\ell - \nabla R\rangle_{\mathrm{F}}|^p]^{\frac{1}{p}} \leq C\beta_1\varkappa\|\boldsymbol{a}\|_2\|\boldsymbol{V}\|_{\mathrm{F}}\sqrt{p}.$$

Invoking Lemma 28 implies sub-Gaussianity of $\langle\boldsymbol{V}, \nabla\ell - \nabla R\rangle_{\mathrm{F}}$ and completes the proof. $\qquad\square$

We proceed by presenting a lemma which constitutes the main part of the proof of Theorem 3 via establishing a recursive bound on the moment generating function (MGF) of $\|\boldsymbol{W}_{\perp}^t\|_{\mathrm{F}}^2$, which will in turn be used to prove high probability statements for $\|\boldsymbol{W}_{\perp}^t\|_{\mathrm{F}}^2$.

**Lemma 12.** *Consider running the iterates of SGD* (3.1), *under either Assumptions* 1&2.A *or* 1&2.B, *with stepsize sequence* $\{\eta_t\}_{t\geq 0}$ *that is either constant* $\eta_t = \eta$ *or decreasing* $\eta_t = m\frac{2(t^*+t)+1}{\gamma(t^*+t+1)^2}$ *(cf.* (Gower et al., 2019, Theorem 3.2)). *Let* $\varkappa := \sup|\partial_1 \ell(\hat{y}, y)|$, $\kappa := \beta_1\|\boldsymbol{a}\|_2 \varkappa$, *and* $\tilde{\varrho} := \lambda + \beta_1^2\|\boldsymbol{a}\|_2^2 + \beta_2 \varkappa \|\boldsymbol{a}\|_\infty$. *Suppose* $\eta_0 \lesssim \tilde{\varrho}^{-1}$. *Let* $\mathcal{F}_t$ *denote the sigma algebra generated by* $\{\boldsymbol{W}^j\}_{j=0}^t$, *and let* $\{A_t\}_{t\geq 0}$ *be a sequence of decreasing events (i.e.* $A_{t+1} \subseteq A_t$*), such that* $A_t \in \mathcal{F}_t$ *and on* $A_t$ *we have* $\mathcal{H}(\boldsymbol{W}^t) + \lambda \mathbf{I}_m \succeq \frac{\gamma}{m}\mathbf{I}_m$. *Then, for every* $t \geq 0$, *with probability at least* $\mathbb{P}(A_t) - \delta$,

$$\|\boldsymbol{W}_\perp^t\|_F^2 \lesssim \prod_{j=0}^{t-1}(1 - \tfrac{\eta_j \gamma}{m})\|\boldsymbol{W}_\perp^0\|_F^2 + \frac{m\eta_t \kappa^2(d + \log(1/\delta))}{\gamma}. \tag{B.1}$$

**Proof.** Let $\mathcal{F}_t$ denote the sigma algebra generated by $\{\boldsymbol{W}^j\}_{j=0}^t$. Recall from Lemma 11 that we define

$$\boldsymbol{\Gamma}^t = \nabla \ell(\hat{y}(\boldsymbol{x}^{(t)}; \boldsymbol{W}^t), y^{(t)}) - \mathbb{E}\Big[\nabla \ell(\hat{y}(\boldsymbol{x}^{(t)}; \boldsymbol{W}^t), y^{(t)})\Big]$$

with

$$\nabla \ell(\hat{y}(\boldsymbol{x}^{(t)}; \boldsymbol{W}^t), y^{(t)}) = \partial_1 \ell(\hat{y}(\boldsymbol{x}^{(t)}; \boldsymbol{W}^t), y^{(t)})\sigma'_{a,b}(\boldsymbol{W}^t\boldsymbol{x}^{(t)})(\boldsymbol{x}^{(t)})^\top.$$

Then for the SGD updates we have

$$\boldsymbol{W}^{t+1} = \boldsymbol{W}^t - \eta_t \nabla \mathcal{R}_\lambda(\boldsymbol{W}^t) - \eta_t \boldsymbol{\Gamma}_t.$$

By projecting the iterates onto the orthogonal complement of the principal subspace,

$$\boldsymbol{W}_\perp^{t+1} = \big(\mathbf{I}_m - \eta_t(\mathcal{H}(\boldsymbol{W}^t) + \lambda\mathbf{I}_m)\big)\boldsymbol{W}_\perp^t - \eta_t \boldsymbol{\Gamma}_\perp^t.$$

Let $\boldsymbol{M}_t := \mathbf{I}_m - \eta_t(\mathcal{H}(\boldsymbol{W}^t) + \lambda\mathbf{I}_m)$. Then, by observing that $\mathbf{1}_{A_{t+1}} \leq \mathbf{1}_{A_t}$, for any $0 \leq s \lesssim \frac{\gamma}{m\eta_t\kappa^2}$ we have

$$\mathbb{E}\Big[\mathbf{1}_{A_{t+1}}e^{s\|\boldsymbol{W}_\perp^{t+1}\|_F^2} \,\big|\, \mathcal{F}_0\Big] \leq \mathbb{E}\Big[\mathbf{1}_{A_t}e^{s\|\boldsymbol{M}_t\boldsymbol{W}_\perp^t\|_F^2 + s\eta_t^2\|\boldsymbol{\Gamma}_\perp^t\|_F^2 + \langle -2s\eta_t\boldsymbol{M}_t\boldsymbol{W}_\perp^t, \boldsymbol{\Gamma}_\perp^t\rangle_F} \,\big|\, \mathcal{F}_0\Big]$$

$$= \mathbb{E}\Big[\mathbf{1}_{A_t}e^{s\|\boldsymbol{M}_t\boldsymbol{W}_\perp^t\|_F^2}\, \mathbb{E}\Big[e^{s\eta_t^2\|\boldsymbol{\Gamma}_\perp^t\|_F^2}e^{\langle -2s\eta_t\boldsymbol{M}_t\boldsymbol{W}_\perp^t, \boldsymbol{\Gamma}_\perp^t\rangle_F} \,\big|\, \mathcal{F}_t\Big] \,\big|\, \mathcal{F}_0\Big]$$

$$\leq \mathbb{E}\Big[\mathbf{1}_{A_t}e^{s\|\boldsymbol{M}_t\boldsymbol{W}_\perp^t\|_F^2}\, \mathbb{E}\Big[e^{2s\eta_t^2\|\boldsymbol{\Gamma}_\perp^t\|_F^2} \,\big|\, \mathcal{F}_t\Big]^{\frac{1}{2}}\, \mathbb{E}\Big[e^{\langle -4s\eta_t\boldsymbol{M}_t\boldsymbol{W}_\perp^t, \boldsymbol{\Gamma}_\perp^t\rangle_F} \,\big|\, \mathcal{F}_t\Big]^{\frac{1}{2}} \,\big|\, \mathcal{F}_0\Big],$$

$$\tag{B.2}$$

where the last inequality follows from the Cauchy-Schwartz inequality for conditional expectation.

Moreover, it is straightforward to observe that $\|\nabla \ell(\hat{y}(\boldsymbol{x}^{(t)}; \boldsymbol{W}^t), y^{(t)})\|_F^2 \leq \kappa^2\|\boldsymbol{x}\|_2^2$, hence

$$\mathbb{E}\Big[\|\nabla \ell(\hat{y}(\boldsymbol{x}^{(t)}; \boldsymbol{W}^t), y^{(t)})\|_F^2\Big] \leq \kappa^2 d.$$

Note that by Jensen's inequality

$$\|\boldsymbol{\Gamma}_\perp^t\|_F^2 \leq 2\|\nabla \ell(\boldsymbol{W}^t\boldsymbol{x}^{(t)}, y^{(t)})\|_F^2 + 2\,\mathbb{E}\Big[\|\nabla \ell(\boldsymbol{W}^t\boldsymbol{x}^{(t)}, y^{(t)})\|_F^2\Big].$$

Consequently

$$\mathbb{E}\big[\exp\big(2s\eta_t^2\|\boldsymbol{\Gamma}_\perp^t\|_F^2\big) \,\big|\, \mathcal{F}_t\big] \leq \exp\big(4s\eta_t^2\kappa^2 d\big)\, \mathbb{E}\big[\exp\big(4s\eta_t^2\kappa^2\|\boldsymbol{x}\|_2^2\big) \,\big|\, \mathcal{F}_t\big]$$
$$\leq \exp\big(4s\eta_t^2\kappa^2 d\big)\exp\big(8s\eta_t^2\kappa^2 d\big),$$

where the second inequality follows from Lemma 33 for $4s\eta_t^2\kappa^2 \leq 1/4$. Since $s \lesssim \frac{\gamma}{m\eta_t\kappa^2}$, in order to satisfy the condition of Lemma 33 we need to ensure $\eta_t\gamma/m \lesssim 1$, which is guaranteed by our $\eta_t\tilde{\varrho} \lesssim 1$ assumption for a suitably small absolute constant, as $\gamma/m \leq \lambda \leq \tilde{\varrho}$.

Next, we bound the last term in (B.2). Let $\boldsymbol{V} := -4s\eta_t\boldsymbol{M}_t\boldsymbol{W}_\perp^t$. Then by Lemma 11 we have

$$\mathbb{E}\big[\exp\big(\langle \boldsymbol{V}, \boldsymbol{\Gamma}_\perp^t\rangle_F\big) \,|\, \mathcal{F}_t\big] \leq \exp\big(Cs^2\eta_t^2\kappa^2\|\boldsymbol{M}_t\boldsymbol{W}_\perp^t\|_F^2\big)$$

Putting things back together in (B.2) and using the tower property of expectation, we have

$$\mathbb{E}\Big[\mathbf{1}_{A_{t+1}}e^{s\|\boldsymbol{W}_\perp^{t+1}\|_F^2} \,\big|\, \mathcal{F}_0\Big] \leq \mathbb{E}\Big[\mathbf{1}_{A_t}e^{s(1+Cs\eta_t^2\kappa^2)\|\boldsymbol{M}_t\boldsymbol{W}_\perp^t\|_F^2 + Cs\eta_t^2\kappa^2 d} \,\big|\, \mathcal{F}_0\Big]. \tag{B.3}$$

Next, we bound $\|M_t\|_2$. By definition of $A_t$, we can already ensure $\mathcal{H}(W^t) \succeq \frac{\gamma}{m}\mathbf{I}_m$ in (B.3). Recall the definition of $\mathcal{H}(W^t)$

$$\mathcal{H}(W^t) = \mathbb{E}\big[\partial_1^2 \ell(\hat{y}(x; W^t), y)\sigma'_{a,b}(W^t x)\sigma'_{a,b}(W^t x)^\top + \partial_1 \ell(\hat{y}(x; W^t), y)\,\mathrm{diag}(\sigma''_{a,b}(W^t x))\big].$$

Notice that $0 \leq \partial_1^2 \ell(\hat{y}(x; W), y) \leq 1$ under either Assumption 2.A or Assumption 2.B. Moreover we have, $|\partial_1 \ell(\hat{y}, y)| \leq \varkappa$. Thus,

$$\mathcal{H}(W^t) + \lambda\mathbf{I}_m \preceq \big(\lambda + \beta_1^2\|a\|_2^2 + \beta_2\varkappa\|a\|_\infty\big)\mathbf{I}_m = \tilde{\varrho}\mathbf{I}_m.$$

Therefore,

$$0 \preceq \mathbf{I}_m - \eta_t(\mathcal{H}(W^t) + \lambda\mathbf{I}_m) \preceq (1 - \tfrac{\eta_t\gamma}{m})\mathbf{I}_m.$$

As a result $\|M_t\|_2 \leq 1 - \frac{\eta_t\gamma}{m}$. Combined with (B.3) we have

$$\mathbb{E}\big[\mathbf{1}_{A_{t+1}} \exp\big(s\|W_\perp^{t+1}\|_F^2\big) \mid \mathcal{F}_0\big] \leq \mathbb{E}\big[\mathbf{1}_{A_t} \exp\big(s(1 + Cs\eta_t^2\kappa^2)(1 - \tfrac{\eta_t\gamma}{m})^2\|W_\perp^t\|_F^2 + Cs\eta_t^2 d\kappa^2\big) \mid \mathcal{F}_0\big]$$
$$\leq \exp\big(Cs\eta_t^2\kappa^2 d\big)\,\mathbb{E}\big[\mathbf{1}_{A_t} \exp\big(s(1 - \tfrac{\eta_t\gamma}{m})\|W_\perp^t\|_F^2\big) \mid \mathcal{F}_0\big] \quad \text{(B.4)}$$

where the second inequality holds by the fact that $Cs\eta_t^2\kappa^2 \leq \eta_t\gamma/m$, which in turn holds when a small enough absolute constant is chosen in $0 \leq s \lesssim \frac{\gamma}{m\eta_t\kappa^2}$. Also notice that for decreasing stepsize,

$$1 - \tfrac{\gamma\eta_t}{m} = \frac{(t + t^*)^2}{(t + t^* + 1)^2} \leq \frac{1 - \frac{(t+t^*)^2}{(t+t^*+1)^2}}{1 - \frac{(t+t^*-1)^2}{(t+t^*)^2}} = \frac{\eta_t}{\eta_{t-1}}, \quad \text{(B.5)}$$

(and the above holds trivially for constant step size), thus when $s \leq \frac{C_1\gamma}{\eta_t\kappa^2}$ for some absolute constant $C_1$, we have $s(1 - \eta_t\gamma) \leq \frac{C_1\gamma}{\eta_{t-1}\kappa^2}$ with the same absolute constant. Hence we are allowed to expand the recursion (B.4), which implies

$$\mathbb{E}\big[\mathbf{1}_{A_t} \exp\big(s\|W_\perp^t\|_F^2\big) \mid \mathcal{F}_0\big] \leq \exp\left(s\prod_{j=0}^{t-1}(1 - \tfrac{\eta_j\gamma}{m})\|W_\perp^0\|_F^2 + Cs\kappa^2 d\sum_{i=0}^{t-1}\eta_i^2 \prod_{j=i+1}^{t-1}(1 - \tfrac{\eta_j\gamma}{m})\right)$$

for all $0 \leq s \lesssim \frac{\gamma}{m\eta_{t-1}\kappa^2}$. Moreover, direct calculation implies that with both constant and decreasing step sizes of Lemma 12, we have $\sum_{i=0}^{t-1}\eta_i^2 \prod_{j=i+1}^{t-1}(1 - \tfrac{\eta_j\gamma}{m}) \leq \frac{Cm\eta_t}{\gamma}$ (with $C = 1$ for constant stepsize). Thus, for all $0 \leq s \lesssim \frac{\gamma}{m\eta_{t-1}\kappa^2}$

$$\mathbb{E}\big[\mathbf{1}_{A_t} \exp\big(s\|W_\perp^t\|_F^2\big) \mid \mathcal{F}_0\big] \leq \exp\left(s\prod_{j=0}^{t-1}(1 - \tfrac{\eta_j\gamma}{m})\|W_\perp^0\|_F^2 + \frac{Csm\eta_t\kappa^2 d}{\gamma}\right).$$

Finally, we can apply a Chernoff bound to obtain

$$\mathbb{P}\big(A_t \cap \{\|W_\perp^t\|_F^2 \geq \varepsilon\} \mid \mathcal{F}_0\big) \leq \exp\left(s\left\{\prod_{j=0}^{t-1}(1 - \eta_j\gamma)\|W_\perp^0\|_F^2 + \frac{Cm\eta_t\kappa^2 d}{\gamma} - \varepsilon\right\}\right)$$

By choosing

$$\varepsilon = \prod_{j=0}^{t-1}(1 - \tfrac{\eta_j\gamma}{m})\|W_\perp^0\|_F^2 + \frac{Cm\eta_t\kappa^2(d + \log(1/\delta))}{\gamma}.$$

and the largest possible $s \lesssim \frac{\gamma}{m\eta_t\kappa^2}$, we obtain

$$\mathbb{P}\big(\|W_\perp^t\|_F^2 \geq \varepsilon \mid \mathcal{F}_0\big) \leq \mathbb{P}\big(A_t \cap \{\|W_\perp^t\|_F \geq \varepsilon\}\big) + \mathbb{P}\big(A_t^C\big) \leq \delta + \mathbb{P}\big(A_t^C\big).$$

Taking another expectation to remove conditioning on initialization completes the proof. $\qquad\square$

The proof of Theorem 3 for decreasing stepsize follows by a direct computation of the quantities in Lemma 12 and is presented below. On the other hand, in order to get a better dependence on $\lambda$, choosing the events $A_t$ for constat stepsize is more subtle and is presented in Section B.2.

### B.1 PROOF OF THEOREM 3 FOR DECREASING STEPSIZE

This part is directly implied by Lemma 12. The following argument holds on an event where $\|\boldsymbol{W}\|_{\mathrm{F}} \lesssim \sqrt{m}$, which happens with probability at least $1 - \mathcal{O}(\delta)$. In order to see this connection, we will first present an improved statement over Lemma 7 for the case of smooth activations. Recall the definition of $\mathcal{H}(\boldsymbol{W})$ for the squared error loss $\ell(\hat{y}, y) = \frac{(\hat{y}-y)^2}{2}$,

$$\mathcal{H}(\boldsymbol{W}) = \mathbb{E}\big[\sigma'_{\boldsymbol{a},\boldsymbol{b}}(\boldsymbol{W}\boldsymbol{x})\sigma'_{\boldsymbol{a},\boldsymbol{b}}(\boldsymbol{W}\boldsymbol{x})^\top\big] + \mathbb{E}\big[(\hat{y}(\boldsymbol{x}; \boldsymbol{W}, \boldsymbol{a}, \boldsymbol{b}) - y)\operatorname{diag}(\sigma''_{\boldsymbol{a},\boldsymbol{b}}(\boldsymbol{W}\boldsymbol{x}))\big].$$

Notice that under Assumption 2.A we have $|\hat{y}| \leq \beta_0 \|\boldsymbol{a}\|_1$. Then basic matrix algebra similar to that of Lemma 7 along with the triangle inequality shows

$$-\beta_2 \|\boldsymbol{a}\|_\infty (\beta_0 \|\boldsymbol{a}\|_1 + \mathbb{E}[|y|])\mathbf{I}_m \prec \mathcal{H}(\boldsymbol{W}) \preceq \big(\beta_1^2 \|\boldsymbol{a}\|_2^2 + \beta_2 \|\boldsymbol{a}\|_\infty (\beta_0 \|\boldsymbol{a}\|_1 + \mathbb{E}[|y|])\big)\mathbf{I}_m.$$

Therefore, with $\lambda \geq \gamma/m + \beta_2 \|\boldsymbol{a}\|_\infty (\|\boldsymbol{a}\|_1 \beta_0 + \mathbb{E}[|y|])$, we have $\mathcal{H}(\boldsymbol{W}) + \lambda \mathbf{I}_m \succeq \gamma/m \mathbf{I}_m$ for all $\boldsymbol{W}$. In addition, $|\partial_1 \ell(\hat{y}, y)| \leq \beta_0 \|\boldsymbol{a}\|_1 + K$ by the triangle inequality. Thus we can invoke Lemma 12 with $\eta_t = \frac{m}{\gamma}\left(1 - \frac{(t^*+t)^2}{(t^*+t+1)^2}\right)$, $\varkappa = \beta_0 \|\boldsymbol{a}\|_1 + K$, and $\mathbf{1}_{A_t} = 1$. Recall that in the statement of the theorem, $\beta_0 = \beta_1 = \beta_2 = 1$, $K \lesssim 1$, $\|\boldsymbol{a}\|_\infty \leq 1/m$, $\|\boldsymbol{a}\|_2 \leq 1/\sqrt{m}$, and $\|\boldsymbol{a}\|_1 \leq 1$, hence $\tilde{\varrho} \asymp \lambda$, and with $t^* \asymp \frac{\lambda}{\gamma}$ we can guarantee $\eta_t \lambda \lesssim 1$. As the step size condition of Lemma 12 is satisfied, the desired result follows.

Similarly, for ReLU we have $|\partial_1 \ell(\hat{y}, y)| \leq 1$ by Assumption 2.B, and for $\lambda \geq \gamma/m + \frac{2\|\boldsymbol{a}\|_\infty}{b^*}\sqrt{\frac{2}{e\pi}}$ (recall $b^* = 1$ in the statement of the theorem), we have $\mathcal{H}(\boldsymbol{W}) + \lambda \mathbf{I}_m \succeq \gamma/m \mathbf{I}_m$. Hence this time, we can invoke Lemma 12 with the same decreasing $\eta_t$, $\mathbf{1}_{A_t} = 1$, and $\varkappa = 1$. $\qquad\square$

### B.2 PROOF OF THEOREM 3 FOR CONSTANT STEPSIZE

In order to improve the condition on $\lambda$, we will specifically look at the events $A_t$ on which $\max_{0 \leq j \leq t} \mathcal{R}_\lambda(\boldsymbol{W}^j)$ is bounded. The following lemma indicates that these events occur with high probability.

**Lemma 13.** *Under Assumptions 1&2.A, consider the setting of Lemma 12 with constant stepsize $\eta \lesssim \tilde{\varrho}^{-1}$. Then we have with probability at least $1 - T\exp(-CT\eta\tilde{\varrho}d)$,*

$$\max_{0 \leq t \leq T} \mathcal{R}_\lambda(\boldsymbol{W}^t) \leq \mathcal{R}_\lambda(\boldsymbol{W}^0) + CT\eta^2\kappa^2\tilde{\varrho}d. \tag{B.6}$$

**Proof.** First, recall from Lemma 8 together with $\sqrt{\mathcal{R}_\lambda(\boldsymbol{W})} \lesssim \beta_0 \|\boldsymbol{a}\|_1 + K = \varkappa$, that

$$\|\nabla^2 \mathcal{R}_\lambda(\boldsymbol{W})\|_2 \lesssim \lambda + \beta_1^2 \|\boldsymbol{a}\|_2^2 + \beta_2 \varkappa \|\boldsymbol{a}\|_\infty = \tilde{\varrho}.$$

We will first prove that for any $t \leq T$ and any $s \lesssim (\eta\kappa^2)^{-1}$, we have

$$\mathbb{E}\Big[e^{s\mathcal{R}_\lambda(\boldsymbol{W}^t)} \,\big|\, \boldsymbol{W}^0\Big] \leq \mathbb{E}\Big[e^{s\mathcal{R}_\lambda(\boldsymbol{W}^0) + Cst\eta^2\kappa^2\tilde{\varrho}d}\Big].$$

Recall that $\mathcal{F}_t$ is the sigma algebra generated by $\{\boldsymbol{W}^j\}_{j=0}^t$, and $\Gamma^t := \nabla\ell(\boldsymbol{W}^t) - \nabla R(\boldsymbol{W}^t)$. By Taylor's theorem and Young's inequality

$$\begin{aligned}
\mathcal{R}_\lambda(\boldsymbol{W}^{t+1}) &\leq \mathcal{R}_\lambda(\boldsymbol{W}^t) - \eta\big\langle\nabla\mathcal{R}_\lambda(\boldsymbol{W}^t), \nabla\mathcal{R}_\lambda(\boldsymbol{W}^t) + \Gamma^t\big\rangle_{\mathrm{F}} + \frac{\tilde{\varrho}\eta^2}{2}\|\nabla\mathcal{R}_\lambda(\boldsymbol{W}^t) + \Gamma^t\|_{\mathrm{F}}^2 \\
&\leq \mathcal{R}_\lambda(\boldsymbol{W}^t) - \eta(1 - \eta\tilde{\varrho})\|\nabla\mathcal{R}_\lambda(\boldsymbol{W}^t)\|_{\mathrm{F}}^2 - \eta\big\langle\nabla\mathcal{R}_\lambda(\boldsymbol{W}^t), \Gamma^t\big\rangle_{\mathrm{F}} + \tilde{\varrho}\eta^2\|\Gamma^t\|_{\mathrm{F}}^2,
\end{aligned}$$

and

$$\begin{aligned}
\mathbb{E}\Big[e^{s\mathcal{R}_\lambda(\boldsymbol{W}^{t+1})} \,\big|\, \mathcal{F}_0\Big] &\leq \mathbb{E}\Big[e^{s\mathcal{R}_\lambda(\boldsymbol{W}^t) - s\eta(1-\eta\tilde{\varrho})\|\nabla\mathcal{R}_\lambda(\boldsymbol{W}^t)\|_{\mathrm{F}}^2 - s\eta\langle\nabla\mathcal{R}_\lambda(\boldsymbol{W}^t),\Gamma^t\rangle_{\mathrm{F}} + s\tilde{\varrho}\eta^2\|\Gamma^t\|_{\mathrm{F}}^2} \,\big|\, \mathcal{F}_0\Big] \\
&= \mathbb{E}\Big[e^{s\mathcal{R}_\lambda(\boldsymbol{W}^t) - s\eta(1-\eta\tilde{\varrho})\|\nabla\mathcal{R}_\lambda(\boldsymbol{W}^t)\|_{\mathrm{F}}^2}\,\mathbb{E}\Big[e^{-s\eta\langle\nabla\mathcal{R}_\lambda(\boldsymbol{W}^t),\Gamma^t\rangle_{\mathrm{F}} + s\tilde{\varrho}\eta^2\|\Gamma^t\|_{\mathrm{F}}^2} \,\big|\, \mathcal{F}_t\Big] \,\big|\, \mathcal{F}_0\Big] \\
&\overset{\text{(a)}}{\leq} \mathbb{E}\Big[e^{s\mathcal{R}_\lambda(\boldsymbol{W}^t) - s\eta(1-\eta\tilde{\varrho})\|\nabla\mathcal{R}_\lambda(\boldsymbol{W}^t)\|_{\mathrm{F}}^2}\,\mathbb{E}\Big[e^{-2s\eta\langle\nabla\mathcal{R}_\lambda(\boldsymbol{W}^t),\Gamma^t\rangle_{\mathrm{F}}} \,\big|\, \mathcal{F}_t\Big]^{\frac{1}{2}}\,\mathbb{E}\Big[e^{2s\tilde{\varrho}\eta^2\|\Gamma^t\|_{\mathrm{F}}^2} \,\big|\, \mathcal{F}_t\Big]^{\frac{1}{2}} \,\big|\, \mathcal{F}_0\Big],
\end{aligned}$$

where (a) follows from the Cauchy-Schwartz inequality for conditional expectation. Moreover, in this setting we have $|\partial_1 \ell(\hat{y}, y)| = |\hat{y} - y| \leq \beta_0 \|\boldsymbol{a}\|_1 + K$, thus letting $\varkappa = \beta_0 \|\boldsymbol{a}\|_1 + K$ in Lemma 11, by the sub-Gaussianity of $\Gamma^t$ we have

$$\mathbb{E}\left[e^{\langle -2s\eta \nabla \mathcal{R}_\lambda(\boldsymbol{W}^t), \Gamma^t \rangle_{\mathrm{F}}} \mid \mathcal{F}_t\right] \leq e^{Cs^2 \eta^2 \kappa^2 \|\nabla \mathcal{R}_\lambda(\boldsymbol{W}^t)\|_{\mathrm{F}}^2}.$$

Furthermore, we have the following upper bound

$$\mathbb{E}\left[e^{Cs\tilde{\varrho}\eta^2 \|\nabla \ell(\boldsymbol{W}^t)\|_{\mathrm{F}}^2} \mid \mathcal{F}_t\right] \leq \mathbb{E}\left[e^{Cs\tilde{\varrho}\eta^2 \kappa^2 \|\boldsymbol{x}\|_2^2} \mid \mathcal{F}_t\right]$$
$$\leq e^{Cs\tilde{\varrho}\eta^2 \kappa^2 d}$$

where the second inequality holds for $s \lesssim \frac{1}{\tilde{\varrho}\eta^2 \kappa^2}$ with a sufficiently small absolute constant by Lemma 33. Similar to the argument in Lemma 12, as we choose $s \lesssim (\eta\kappa^2)^{-1}$, in order to satisfy the condition of Lemma 12 it suffices to have $\eta\tilde{\varrho} \lesssim 1$ for a sufficiently small absolute constant. Putting the above bounds back together, we have

$$\mathbb{E}\left[e^{s\mathcal{R}_\lambda(\boldsymbol{W}^{t+1})} \mid \mathcal{F}_0\right] \leq \mathbb{E}\left[e^{s\mathcal{R}_\lambda(\boldsymbol{W}^t) - s\eta(1 - \eta\tilde{\varrho} - Cs\eta\kappa^2)\|\nabla \mathcal{R}_\lambda(\boldsymbol{W}^t)\|_{\mathrm{F}}^2 + Cs\eta^2 \kappa^2 \tilde{\varrho}d} \mid \mathcal{F}_0\right].$$

Expanding the recursion yields, for any $s \lesssim (\eta\kappa^2)^{-1}$ (with a sufficiently small absolute constant chosen)

$$\mathbb{E}\left[e^{s\mathcal{R}_\lambda(\boldsymbol{W}^t)} \mid \mathcal{F}_0\right] \leq e^{s\mathcal{R}_\lambda(\boldsymbol{W}^0) + Cst\eta^2 \kappa^2 \tilde{\varrho}d}.$$

As a result, by applying Markov's inequality at time $t \leq T$, we have

$$\mathbb{P}\left(\mathcal{R}_\lambda(W^t) \geq \mathcal{R}_\lambda(\boldsymbol{W}^0) + CT\eta^2 \kappa^2 \tilde{\varrho}d\right) \leq e^{-CsT\eta^2 \kappa^2 \tilde{\varrho}d} \leq e^{-CT\eta\tilde{\varrho}d}.$$

Consequently, with a union bound we have

$$\mathbb{P}\left(\max_{0 \leq t \leq T} \mathcal{R}_\lambda(\boldsymbol{W}^t) \geq \mathcal{R}_\lambda(\boldsymbol{W}^0) + CT\eta^2 \kappa^2 \tilde{\varrho}d\right) \leq Te^{-CT\eta\tilde{\varrho}d},$$

which completes the proof. $\qquad\square$

As depicted by the following proposition, the rest of the proof is analogous to the decreasing stepsize case.

**Proposition 14.** *Consider the setting of Lemma 13 with constant stepsize $\eta \lesssim \tilde{\varrho}^{-1}$ and $\lambda$ sufficiently large such that $\lambda \geq \gamma/m + \beta_2 \|\boldsymbol{a}\|_\infty \sqrt{2(\mathcal{R}_\lambda(\boldsymbol{W}^0) + CT\eta^2 \kappa^2 \tilde{\varrho}d)}$. Then with probability at least $1 - Te^{-CT\eta\tilde{\varrho}d} - \delta$ we have*

$$\|\boldsymbol{W}_\perp^T\|_F^2 \leq (1 - \tfrac{\eta\gamma}{m})^T \|\boldsymbol{W}^0\|_F^2 + \frac{Cm\eta\kappa^2(d + \log(1/\delta))}{\gamma} \tag{B.7}$$

**Proof.** Let $A_t = \{\max_{0 \leq i \leq t} \mathcal{R}_\lambda(\boldsymbol{W}^i) \leq \mathcal{R}_\lambda(\boldsymbol{W}^0) + CT\eta^2 \kappa^2 \tilde{\varrho}d\}$. Notice that $A_t$ is $\{\boldsymbol{W}^j\}_{j=0}^t$ measurable and $A_{t+1} \subseteq A_t$. By the bound established on $\mathcal{H}(\boldsymbol{W})$ in Lemma 7, on $A_t$ we have $\mathcal{H}(\boldsymbol{W}^i) + \lambda \mathbf{I}_m \succeq \gamma/m \mathbf{I}_m$ for all $0 \leq i \leq T$. Moreover, from Lemma 13, we have $\mathbb{P}(A_T^C) \leq Te^{-CT\eta\tilde{\varrho}d}$. Invoking Lemma 12 finishes the proof. $\qquad\square$

The above proposition immediately implies the statement of Theorem 3 for constant stepsize, which we repeat here as a corollary of Proposition 14.

**Corollary 15** (Proof of Theorem 3 for constant stepsize). *Consider the setting of Lemma 13 with $\lambda$ given in Proposition 14, for constant stepsize $\eta = \frac{2m\log(T)}{\gamma T}$ with $T \geq (1/\delta)^{C/d}$. Then with probability at least $1 - \delta$ we have*

$$\|\boldsymbol{W}_\perp^T\|_F \lesssim \frac{\|\boldsymbol{W}_\perp^0\|_F}{T} + \frac{m\kappa}{\gamma}\sqrt{\frac{\log(T)(d + \log(1/\delta))}{T}}. \tag{B.8}$$

## C    PROOFS OF SECTION 4

### C.1    PROOF OF THEOREM 5

As our arguments are based on the Rademacher complexity of a two-layer neural network, we require the knowledge of the norm of $\boldsymbol{W}^t$. We prove a high probability bound for this norm in the following lemma.

**Lemma 16.** *Under Assumptions 1&2.A or 1&2.B with either decreasing or constant stepsize as in Theorem 3, let $\varkappa = \sup_{\hat{y},y}|\partial_1\ell(\hat{y},y)| < \infty$ and $\kappa_\infty := \beta_1\|\boldsymbol{a}\|_\infty\varkappa$. Then for any $t \geq 1$, with probability at least $1 - m\exp\left(\frac{-\gamma td}{m\varphi\lambda}\right)$ we have for all $1 \leq j \leq m$*

$$\|\boldsymbol{w}_j^t\|_2 \leq \prod_{i=0}^{t-1}(1 - \eta_i\lambda)\|\boldsymbol{w}_j^0\|_2 + \frac{3\kappa_\infty\sqrt{d}}{\lambda}, \tag{C.1}$$

*where $\varphi = 1$ for decreasing step size and $\varphi = \log(T)$ for constant step size.*

**Proof.** First, we prove that for any $t > 0$ and $0 \leq s \leq \frac{2\sqrt{d}}{\kappa_\infty\eta_{t-1}}$, we have

$$\mathbb{E}\big[\exp(s\|\boldsymbol{w}_j^t\|_2)\,|\,\boldsymbol{W}^0\big] \leq \exp\left(s\prod_{i=0}^{t-1}(1 - \eta_i\lambda)\|\boldsymbol{w}_j^0\|_2 + \frac{2s\kappa_\infty\sqrt{d}}{\lambda}\right), \tag{C.2}$$

The base case of $t = 0$ is trivial, and for the induction step, for any $0 \leq s \leq \frac{2\sqrt{d}}{\kappa_\infty\eta_t}$ we have

$$
\begin{aligned}
\mathbb{E}\big[\exp\big(s\|\boldsymbol{w}_j^{t+1}\|_2\big)\,|\,\boldsymbol{W}^0\big] &= \mathbb{E}\big[\exp\big(s\|(1 - \eta_t\lambda)\boldsymbol{w}_j^t - \eta_t\nabla_{\boldsymbol{w}_j}\ell(\hat{y}(\boldsymbol{x};\boldsymbol{W}^t),y)\|_2\big)\,|\,\boldsymbol{W}^0\big] \\
&\leq \mathbb{E}\big[\exp\big(s(1 - \eta_t\lambda)\|\boldsymbol{w}_j^t\|_2 + s\eta_t\|\nabla_{\boldsymbol{w}_j}\ell(\hat{y}(\boldsymbol{x};\boldsymbol{W}^t),y)\|_2\big)\,|\,\boldsymbol{W}^0\big] \\
&= \mathbb{E}\big[\exp\big(s(1 - \eta_t\lambda)\|\boldsymbol{w}_j^t\|_2 + s\eta_t\kappa_\infty\|\boldsymbol{x}\|_2\big)\,|\,\boldsymbol{W}^0\big] \\
&= \mathbb{E}\big[\exp\big(s(1 - \eta_t\lambda)\|\boldsymbol{w}_j^t\|_2\big)\,\mathbb{E}\big[\exp(s\eta_t\kappa_\infty\|\boldsymbol{x}\|_2)\,|\,\boldsymbol{W}^t,\boldsymbol{W}^0\big]\,|\,\boldsymbol{W}^0\big] \\
&\overset{(a)}{\leq} \mathbb{E}\left[\exp\big(s(1 - \eta_t\lambda)\|\boldsymbol{w}_j^t\|_2\big)\exp\left(s\eta_t\kappa_\infty\sqrt{d} + \frac{s^2\kappa_\infty^2\eta_t^2}{2}\right)\,|\,\boldsymbol{W}^0\right] \\
&\overset{(b)}{\leq} \exp\left(s\prod_{i=0}^{t}(1 - \eta_i\lambda)\|\boldsymbol{w}_j^0\|_2 + \frac{2s\kappa_\infty\sqrt{d}}{\lambda}\right)
\end{aligned}
$$

where (a) holds since $\|\boldsymbol{x}\|_2$ is a 1-Lipschitz function of a standard Gaussian random vector, thus it is sub-Gaussian with parameter 1 (Lemma 29) and additionally $\mathbb{E}[\|\boldsymbol{x}\|_2] \leq \sqrt{d}$, and (b) holds by the induction hypothesis (notice that for decreasing stepsize $s(1 - \eta_t\lambda) \leq \frac{2\sqrt{d}}{\kappa_\infty\eta_{t-1}}$ by (B.5)). Next, we apply the following Chernoff bound,

$$\mathbb{P}\left(\|\boldsymbol{w}_j^t\|_2 > \prod_{i=0}^{t-1}(1 - \eta_i\lambda)\|\boldsymbol{w}_j^0\|_2 + \frac{3\kappa_\infty\sqrt{d}}{\lambda}\,\Big|\,\boldsymbol{W}^0\right) \leq \exp\left(-\frac{s\kappa_\infty\sqrt{d}}{\lambda}\right),$$

which holds for any $0 \leq s \leq \frac{2\sqrt{d}}{\kappa_\infty\eta_{t-1}}$. Choosing the largest $s$ possible and noting that $\eta_{t-1} \leq \frac{2m\varphi}{\gamma t}$ yields an $\exp\left(\frac{-\gamma td}{m\varphi\lambda}\right)$ upper bound on the conditional probability, which followed by taking expectation removes the randomness of conditioning on $\boldsymbol{w}_j^0$. Finally applying a union bound gives us the desired bound. $\qquad\square$

In addition, we would like to approximate $R_\tau(\boldsymbol{W}^T)$ and $\hat{R}_\tau(\boldsymbol{W}^T)$ with $R_\tau(\boldsymbol{W}_\parallel^T)$ and $\hat{R}_\tau(\boldsymbol{W}_\parallel^T)$ respectively. As a result, we will investigate the Lipschitzness of the population and empirical risk in the next lemma.

**Lemma 17.** *Under either Assumptions 1&2.A or 1&2.B, the truncated risk $\boldsymbol{W} \mapsto R_\tau(\boldsymbol{W})$ is $\sqrt{2}\tau\beta_1\|\boldsymbol{a}\|_2$-Lipschitz. Moreover, for $T \geq d + \log(1/\delta)$ with probability at least $1 - \delta$ over the stochasticity of $\{\boldsymbol{x}^{(t)}\}_{0 \leq t \leq T-1}$, the truncated empirical risk $\boldsymbol{W} \mapsto \hat{R}_\tau(\boldsymbol{W})$ is $C\tau\beta_1\|\boldsymbol{a}\|_2$-Lipschitz for some absolute constant $C$.*

**Proof.** We begin by the simple observation that $\hat{y} \mapsto \ell(\hat{y}, y) \wedge \tau$ is $\sqrt{2\tau}$-Lipschitz when $\ell(\hat{y}, y) = 1/2(\hat{y} - y)^2$ and 1-Lipschitz when $|\partial_1 \ell(\hat{y}, y)| \leq 1$. As $\tau \geq 1$, we can consider both of them as $\sqrt{2\tau}$ Lipschitz. Thus by Jensen's inequality

$$|R_\tau(\boldsymbol{W}) - R_\tau(\boldsymbol{W}')| \leq \sqrt{2}\tau \, \mathbb{E}\big[|\hat{y}(\boldsymbol{x}; \boldsymbol{W}) - \hat{y}(\boldsymbol{x}; \boldsymbol{W}')|\big]$$

$$\leq \sqrt{2}\tau \, \mathbb{E}\left[\left(\sum_{j=1}^m a_j \sigma(\langle \boldsymbol{w}_j, \boldsymbol{x}\rangle + b_j) - \sum_{j=1}^m a_j \sigma(\langle \boldsymbol{w}'_j, \boldsymbol{x}\rangle + b_j)\right)^2\right]^{\frac{1}{2}}$$

$$\overset{(a)}{\leq} \sqrt{2}\tau \|\boldsymbol{a}\|_2 \sqrt{\sum_{j=1}^m \mathbb{E}\Big[\big(\sigma(\langle \boldsymbol{w}_j, \boldsymbol{x}\rangle + b_j) - \sigma(\langle \boldsymbol{w}'_j, \boldsymbol{x}\rangle + b_j)\big)^2\Big]}$$

$$\leq \sqrt{2}\tau\beta_1 \|\boldsymbol{a}\|_2 \sqrt{\sum_{j=1}^m \mathbb{E}\Big[\langle \boldsymbol{w}_j - \boldsymbol{w}'_j, \boldsymbol{x}\rangle^2\Big]} \qquad (\text{C.3})$$

$$\leq \sqrt{2}\tau\beta_1 \|\boldsymbol{a}\|_2 \|\boldsymbol{W} - \boldsymbol{W}'\|_{\mathrm{F}}$$

where (a) follows from the Cauchy-Schwartz inequality. Note that Equation (C.3) also holds for $|\hat{R}_\tau(\boldsymbol{W}) - \hat{R}_\tau(\boldsymbol{W}')|$ when expectation is over the empirical distribution given by the training samples, meaning

$$|\hat{R}_\tau(\boldsymbol{W}) - \hat{R}_\tau(\boldsymbol{W}')| \leq \sqrt{2}\tau\beta_1 \|\boldsymbol{a}\|_2 \sqrt{\sum_{j=1}^m (\boldsymbol{w}_j - \boldsymbol{w}'_j)^\top \left(\frac{1}{T}\sum_{t=0}^{T-1} \boldsymbol{x}^{(t)}\boldsymbol{x}^{(t)\top}\right)(\boldsymbol{w}_j - \boldsymbol{w}'_j)}. \quad (\text{C.4})$$

By Lemma 30, with probability at least $1 - \delta$, we have

$$\left\|\frac{1}{T}\sum_{t=0}^{T-1} \boldsymbol{x}^{(t)}\boldsymbol{x}^{(t)\top} - \mathbf{I}_d\right\|_2 \lesssim 1,$$

which completes the proof. $\qquad\qquad\qquad\qquad\qquad\qquad\qquad\qquad\qquad\qquad\qquad\qquad\qquad\square$

**Lemma 18.** *Suppose either Assumptions* (1,2.A) *or* (1,2.B) *hold. Denote the loss with* $\ell(\hat{y}, y) = \ell(\hat{y} - y)$,

$$\tilde{S} = \left\{\tilde{\boldsymbol{W}} \in \mathbb{R}^{m \times k}, \boldsymbol{a}, \boldsymbol{b} \in \mathbb{R}^m \; : \; \|\boldsymbol{a}\|_2 \leq \frac{r_a}{\sqrt{m}}, \quad \|\boldsymbol{b}\|_\infty \leq r_b, \quad \|\tilde{\boldsymbol{w}}_j\|_2 \leq r_{\tilde{\boldsymbol{w}}}, \; \forall 1 \leq j \leq m\right\}$$

*and*

$$\mathcal{G} = \left\{(\tilde{\boldsymbol{x}}, y) \mapsto \ell(\hat{y}(\tilde{\boldsymbol{x}}; \tilde{\boldsymbol{W}}, \boldsymbol{a}, \boldsymbol{b}), y) \wedge \tau \; : \; (\tilde{\boldsymbol{W}}, \boldsymbol{a}, \boldsymbol{b}) \in \tilde{S}\right\}$$

*for* $\tilde{\boldsymbol{x}} \in \mathbb{R}^k$ *and* $y \in \mathbb{R}$. *Let* $\mathfrak{R}(G)$ *denote the Rademacher complexity of the function class* $\mathcal{G}$ *(see Lemma 18 for definition). Then with* $\tilde{\boldsymbol{x}} \sim \mathcal{N}(0, \boldsymbol{U}\boldsymbol{U}^\top)$ *for some* $\boldsymbol{U} \in \mathbb{R}^{k \times d}$ *we have*

$$\mathfrak{R}(\mathcal{G}) \leq 2\tau\beta_1(r_{\tilde{\boldsymbol{w}}}\|\boldsymbol{U}\|_F + r_b)r_a\sqrt{\frac{2}{T}},$$

*where* $T$ *is the number of samples.*

**Proof.** Let $\mathcal{F} = \{(\tilde{\boldsymbol{x}}, y) \mapsto f_{\boldsymbol{a}, \tilde{\boldsymbol{W}}}(\tilde{\boldsymbol{x}}, y) \; : \; (\tilde{\boldsymbol{W}}, \boldsymbol{a}, \boldsymbol{b}) \in \tilde{S}\}$ for $f_{\boldsymbol{a}, \tilde{\boldsymbol{W}}}(\tilde{\boldsymbol{x}}, y) = \hat{y}(\tilde{\boldsymbol{x}}; \tilde{\boldsymbol{W}}, \boldsymbol{a}, \boldsymbol{b}) - y$. Define $g(z) := \ell(z) \wedge \tau$, and notice $\mathcal{G} = \{(\tilde{\boldsymbol{x}}, y) \mapsto g(f_{\boldsymbol{a}, \tilde{\boldsymbol{W}}}(\tilde{\boldsymbol{x}}, y)) \; : \; f_{\boldsymbol{a}, \tilde{\boldsymbol{W}}} \in \mathcal{F}\}$, and that $g$ is a $\sqrt{2\tau}$-Lipschitz (thus $\sqrt{2}\tau$-Lipschitz as well, for $\tau > 1$) function. Then by Talagrand's contraction principle we have $\mathfrak{R}(\mathcal{G}) \leq \sqrt{2}\tau\mathfrak{R}(\mathcal{F})$. Moreover, let $\{\xi_t\}_{0 \leq t \leq T-1}$ be a sequence of i.i.d.

Rademacher random variables. Then similar to the Rademacher bound of Damian et al. (2022)

$$\mathfrak{R}(\mathcal{F}) = \mathbb{E}\left[\sup_{(\tilde{\boldsymbol{W}},\boldsymbol{a},\boldsymbol{b})\in\tilde{S}} \frac{1}{T}\sum_{t=0}^{T-1}\xi_t\Big(\boldsymbol{a}^\top\sigma(\tilde{\boldsymbol{W}}\tilde{\boldsymbol{x}}^{(t)}+\boldsymbol{b})-y^{(i)}\Big)\right]$$

$$= \mathbb{E}\left[\sup_{(\tilde{\boldsymbol{W}},\boldsymbol{a},\boldsymbol{b})\in\tilde{S}} \frac{1}{T}\sum_{t=0}^{T-1}\xi_t\boldsymbol{a}^\top\sigma(\tilde{\boldsymbol{W}}\tilde{\boldsymbol{x}}^{(t)}+\boldsymbol{b})\right]$$

$$\overset{(a)}{\leq} \frac{r_a}{T}\mathbb{E}\left[\sup_{(\tilde{\boldsymbol{W}},\boldsymbol{b})\in\tilde{S}} \|\sum_{t=0}^{T-1}\xi_t\sigma(\tilde{\boldsymbol{W}}\tilde{\boldsymbol{x}}^{(t)}+\boldsymbol{b})\|_\infty\right]$$

$$\leq \frac{r_a}{T}\mathbb{E}\left[\sup_{\|\tilde{\boldsymbol{w}}\|_2\leq r_{\tilde{w}},|\tilde{b}|\leq r_b} |\sum_{t=0}^{T-1}\xi_t\sigma\Big(\big\langle\tilde{\boldsymbol{w}},\tilde{\boldsymbol{x}}^{(t)}\big\rangle+\tilde{b}\Big)|\right]$$

$$\overset{(b)}{\leq} \frac{2\beta_1 r_a}{n}\mathbb{E}\left[\sup_{\|\tilde{\boldsymbol{w}}\|_2\leq r_{\tilde{w}},|b|\leq r_b} |\sum_{t=0}^{T-1}\xi_t\Big(\big\langle\tilde{\boldsymbol{w}},\tilde{\boldsymbol{x}}^{(t)}\big\rangle+b\Big)|\right]$$

$$\leq \frac{2\beta_1 r_a}{T}\mathbb{E}\left[\sup_{\|\tilde{\boldsymbol{w}}\|_2\leq r_{\tilde{w}}} |\sum_{t=0}^{T-1}\xi_t\langle\tilde{\boldsymbol{w}},\tilde{\boldsymbol{x}}\rangle| + \sup_{|\tilde{b}|\leq r_b}|\sum_{t=0}^{T-1}\xi_t b|\right]$$

$$\leq \frac{2\beta_1 r_a}{T}\left(r_{\tilde{w}}\mathbb{E}\left[\|\sum_{t=0}^{T-1}\xi_t\tilde{\boldsymbol{x}}^{(t)}\|_2\right] + r_b\sqrt{T}\right)$$

$$\leq \frac{2\beta_1(r_{\tilde{w}}\|\boldsymbol{U}\|_{\mathrm{F}}+r_b)r_a}{\sqrt{T}},$$

where (a) holds by Hölder's inequality and the fact that $\|\boldsymbol{a}\|_1 \leq \sqrt{m}\|\boldsymbol{a}\|_2 \leq r_a$, and (b) follows from the fact that $\sigma$ is $\beta_1$ Lipschitz, thus another application of Talagrand's contraction principle. $\qquad\square$

**Proof. [Proof of Theorem 5]** Let $\mathcal{E}_1$ denote the event of Lemma 17. We begin with the following decomposition for generalization error which holds on $\mathcal{E}_1$,

$$R_\tau(\boldsymbol{W}^T) - \hat{R}_\tau(\boldsymbol{W}^T) = R_\tau(\boldsymbol{W}^T) - R_\tau(\boldsymbol{W}_\parallel^T) + R_\tau(\boldsymbol{W}_\parallel^T) - \hat{R}_\tau(\boldsymbol{W}_\parallel^T) + \hat{R}_\tau(\boldsymbol{W}_\parallel^T) - \hat{R}_\tau(\boldsymbol{W}^T)$$

$$\leq C\tau\beta_1\|\boldsymbol{a}\|_2\|\boldsymbol{W}_\perp^T\|_{\mathrm{F}} + R_\tau(\boldsymbol{W}_\parallel^T) - \hat{R}_\tau(\boldsymbol{W}_\parallel^T).$$

where the upper bound follows from Lemma 17. Consequently,

$$\sup_{\boldsymbol{a},\boldsymbol{b}} R_\tau(\boldsymbol{W}^T,\boldsymbol{a},\boldsymbol{b}) - \hat{R}_\tau(\boldsymbol{W}^T,\boldsymbol{a},\boldsymbol{b}) \leq \tfrac{C\tau\beta_1 r_a}{\sqrt{m}}\|\boldsymbol{W}_\perp^T\|_{\mathrm{F}} + \sup_{\boldsymbol{a},\boldsymbol{b}} R_\tau(\boldsymbol{W}_\parallel^T,\boldsymbol{a},\boldsymbol{b}) - \hat{R}_\tau(\boldsymbol{W}_\parallel^T,\boldsymbol{a},\boldsymbol{b}).$$
(C.5)

We begin by upper bounding the first term. From Theorem 3, on an event $\mathcal{E}_2$ we have with probability at least $1 - \mathcal{O}(\delta)$

$$\frac{\|\boldsymbol{W}_\perp^T\|_{\mathrm{F}}}{\sqrt{m}} \lesssim \kappa\sqrt{\frac{d+\log(1/\delta)}{\gamma^2 T}}.$$

Next, we bound the second term in (C.5). For each $\boldsymbol{W}$, define $\tilde{\boldsymbol{W}} := \boldsymbol{U}^\dagger\boldsymbol{W}_\parallel$, where $\boldsymbol{U}^\dagger$ is the Moore–Penrose pseudo-inverse of $\boldsymbol{U}$. Then, since we have the representation $\boldsymbol{W}_\parallel = \boldsymbol{M}\boldsymbol{U}$ for some $\boldsymbol{M} \in \mathbb{R}^{m\times k}$,

$$\tilde{\boldsymbol{W}}\boldsymbol{U} = \boldsymbol{W}_\parallel\boldsymbol{U}^\dagger\boldsymbol{U} = \boldsymbol{M}\boldsymbol{U}\boldsymbol{U}^\dagger\boldsymbol{U} = \boldsymbol{M}\boldsymbol{U} = \boldsymbol{W}_\parallel.$$

Thus, $\boldsymbol{W}\boldsymbol{x} = \tilde{\boldsymbol{W}}\tilde{\boldsymbol{x}}$ and $\ell(\hat{y}(\boldsymbol{x};\boldsymbol{W},\boldsymbol{a},\boldsymbol{b}),y) = \ell(\hat{y}(\tilde{\boldsymbol{x}};\tilde{\boldsymbol{W}},\boldsymbol{a},\boldsymbol{b}),y)$ for $\tilde{\boldsymbol{x}} = \boldsymbol{U}\boldsymbol{x}$, when $\boldsymbol{W}$ is in the principal subspace, i.e. $\boldsymbol{W} = \boldsymbol{W}_\parallel$. Let $\mathcal{E}_3$ denote the event of Lemma 16, on which

$$\|\boldsymbol{w}_j^T\|_2 \leq \prod_{i=0}^{T-1}(1-\tfrac{\eta_i\gamma}{m})\|\boldsymbol{w}_j^0\|_2 + \frac{3\kappa_\infty\sqrt{d}}{\lambda}$$

and consequently

$$\|\tilde{\boldsymbol{w}}_j^T\|_2 \leq \|\boldsymbol{U}^\dagger\|_2\left(\prod_{i=0}^{T-1}(1-\tfrac{\eta_i\gamma}{m})\|\boldsymbol{w}_j^0\|_2 + \frac{3\kappa_\infty\sqrt{d}}{\lambda}\right)$$

for any $1 \leq j \leq m$. Define $r_{\tilde{\boldsymbol{w}}^T}$ as the RHS bound above. Then on $\mathcal{E}_3$

$$\sup_{\boldsymbol{a},\boldsymbol{b}} R_\tau(\boldsymbol{W}_\|^T) - \hat{R}_\tau(\boldsymbol{W}_\|^T) \leq \sup_{(\tilde{\boldsymbol{W}},\boldsymbol{a},\boldsymbol{b})\in\tilde{S}} R_\tau(\tilde{\boldsymbol{W}},\boldsymbol{a},\boldsymbol{b}) - \hat{R}_\tau(\tilde{\boldsymbol{W}},\boldsymbol{a},\boldsymbol{b}),$$

where we recall

$$\tilde{S} := \left\{\tilde{\boldsymbol{W}} \in \mathbb{R}^{m\times k}, \boldsymbol{a},\boldsymbol{b}\in\mathbb{R}^m \; : \; \|\boldsymbol{a}\|_2 \leq \tfrac{r_a}{\sqrt{m}}\,, \|\boldsymbol{b}\|_\infty \leq r_b\,, \quad \|\tilde{\boldsymbol{w}}_j\|_2 \leq r_{\tilde{\boldsymbol{w}}^T}\,, \forall\, 1\leq j\leq m\right\}.$$

Additionally define

$$\mathcal{G} = \{(\tilde{\boldsymbol{x}},y)\mapsto \ell(\hat{y}(\tilde{\boldsymbol{x}};\tilde{\boldsymbol{W}},\boldsymbol{a},\boldsymbol{b}),y)\wedge\tau \; : \; (\tilde{\boldsymbol{W}},\boldsymbol{a},\boldsymbol{b})\in\tilde{S}\}.$$

Then Lemma 31 and Lemma 18 yield

$$\mathbb{E}\left[\sup_{(\tilde{\boldsymbol{W}},\boldsymbol{a},\boldsymbol{b})\in\tilde{S}} R_\tau(\tilde{\boldsymbol{W}}) - \hat{R}_\tau(\tilde{\boldsymbol{W}})\right] \leq 2\mathfrak{R}(\mathcal{G}) \lesssim \tau\beta_1(r_{\tilde{\boldsymbol{w}}^T}+r_b)r_a\|\boldsymbol{U}\|_{\mathrm{F}}\sqrt{\frac{1}{T}}.$$

Besides, as the loss is bounded by $\tau$, by McDiarmid's inequality, on an event $\mathcal{E}_4$ which happens with probability at least $1-\mathcal{O}(\delta)$ we have

$$\sup_{(\tilde{\boldsymbol{W}},\boldsymbol{a},\boldsymbol{b})\in\tilde{S}} R_\tau(\tilde{\boldsymbol{W}}) - \hat{R}_\tau(\tilde{\boldsymbol{W}}) \leq \mathbb{E}\left[\sup_{(\tilde{\boldsymbol{W}},\boldsymbol{a},\boldsymbol{b})\in\tilde{S}} R(\tilde{\boldsymbol{W}})\right] + C\tau\sqrt{\frac{\log(1/\delta)}{T}}.$$

and consequently on $\cap_{i=1}^4 \mathcal{E}_i$

$$\sup_{\boldsymbol{a},\boldsymbol{b}} R_\tau(\boldsymbol{W}_\|^T,\boldsymbol{a},\boldsymbol{b}) - \hat{R}_\tau(\boldsymbol{W}_\|^T,\boldsymbol{a},\boldsymbol{b}) \lesssim \tau\beta_1(r_{\tilde{\boldsymbol{w}}^T}+r_b)r_a\|\boldsymbol{U}\|_{\mathrm{F}}\sqrt{\frac{1}{T}} + \tau\sqrt{\frac{\log(1/\delta)}{T}}.$$

Finally, observe that $\|\boldsymbol{a}\|_1 \leq \sqrt{m}\|\boldsymbol{a}\|_2 \leq r_a$, and without loss of generality assume $\boldsymbol{U}$ is orthonormal, hence $\|\boldsymbol{U}^\dagger\|_2 = 1$ and $\|\boldsymbol{U}\|_{\mathrm{F}} = \sqrt{k}$, thus with probability at least $1 - o(\delta)$,

$$\begin{aligned}
\sup_{\boldsymbol{a},\boldsymbol{b}} R_\tau(\boldsymbol{W}^T,\boldsymbol{a},\boldsymbol{b}) - \hat{R}_\tau(\boldsymbol{W}^T,\boldsymbol{a},\boldsymbol{b}) \lesssim\, & \tau\beta_1 r_a\kappa\sqrt{\frac{d+\log(1/\delta)}{\gamma^2 T}} \\
& + \tau\beta_1 r_a\left\{\left(\frac{t^*}{t^*+T}\right)^2 r_w + \frac{\kappa_\infty}{\lambda} + r_b\right\}\sqrt{\frac{dk}{T}} \\
& + \tau\sqrt{\frac{\log(1/\delta)}{T}}. \tag{C.6}
\end{aligned}$$

We remark that in the setting of Theorem 3 which is adapted in Theorem 5, $\|\boldsymbol{a}\|_\infty \lesssim m^{-1}$, thus $\kappa_\infty \lesssim m^{-1}$. Finally, we observe that $r_w \leq \sqrt{2m}$ with probability at least $1-\mathcal{O}(\delta)$ over initialization, which completes the proof. $\qquad\square$

## C.2 PROOF OF THEOREM 4

Note that due to the special symmetry in the initialization of Algorithm 1, while training the first layer, all neurons have an identical value, i.e. $\boldsymbol{w}_j^t = \boldsymbol{w}^t$ for all $j$, and that the stochastic gradient with respect to any neuron can be denote by $\nabla\ell = a\partial_1\ell(\hat{y},y)\sigma'(\langle\boldsymbol{w},\boldsymbol{x}\rangle + b)\boldsymbol{x}$. Furthermore, $\nabla_{\boldsymbol{w}_j}\mathcal{R}_\lambda(\boldsymbol{W})$ will also be identical for all $j$, which due to the population gradient formula (2.6), we denote by

$$\nabla\mathcal{R}_\lambda(\boldsymbol{w}) = (h(\boldsymbol{w})+\lambda)\boldsymbol{w} + \mathfrak{d}(\boldsymbol{w})\boldsymbol{u},$$

where $h(\boldsymbol{w}) = \sum_{j=1}^m \boldsymbol{\mathcal{H}}_{ij}(W)$ and $\mathfrak{d}(\boldsymbol{w}) = a\,\mathbb{E}\big[\partial_{12}^2\ell(\hat{y},y)\sigma'(\langle\boldsymbol{w},\boldsymbol{x}\rangle + b)f'(\langle\boldsymbol{u},\boldsymbol{x}\rangle)\big]$. Additionally, via the arguments in the proof of Lemma 1, it is not difficult to observe $\gamma/m \leq h(\boldsymbol{w}) + \lambda \lesssim m^{-1}$. Furthermore, similar to the arguments of Lemma 11, $\langle\nabla\ell,\boldsymbol{v}\rangle$ is $Ca\|\boldsymbol{v}\|_2$-sub-Gaussian for any $\boldsymbol{v}\in\mathbb{R}^d$. Next, we will derive a lower bound for $\langle\boldsymbol{w}^t,\boldsymbol{u}\rangle$ to argue that useful features are learned, which first requires obtaining a sharper upper bound on $\|\boldsymbol{w}^t\|_2$ than that of Lemma 16. This improvement is possible due to considering the special case of $\boldsymbol{w}_j^t = \boldsymbol{w}^t$ here.

**Lemma 19.** *Suppose $t \geq d$. Then,*

$$\|\boldsymbol{w}^t\|_2 \leq \left(\frac{t^*}{t^* + t}\right)\|\boldsymbol{w}^0\|_2 + \frac{Cma}{\gamma}$$

*with probability at least $1 - \exp(-C(t^* + t))$. In particular, using the union bound, we have*

$$\sup_{t \geq t_0}\|\boldsymbol{w}^t\|_2 \leq \|\boldsymbol{w}^0\|_2 + \frac{Cma}{\gamma} \lesssim 1$$

*with probability at least $1 - \exp(-C(t^* + t_0)) - \exp(-Cd)$.*

**Proof.** Let $\mathfrak{e}^t := \nabla_w\ell - \nabla_w R$. Then we have

$$\boldsymbol{w}^{t+1} = \boldsymbol{w}^t - \eta_t\nabla_{\boldsymbol{w}}\mathcal{R}_\lambda - \eta_t\mathfrak{e}^t.$$

Recall that $\langle\mathfrak{e}^t, \boldsymbol{v}\rangle$ is $Ca\|\boldsymbol{v}\|_2$-sub-Gaussian, and $\mathcal{F}_t$ is the sigma algebra generated by $\{w^j\}_{0 \leq j \leq t}$. Let $\boldsymbol{\omega}^t := \boldsymbol{w}^t - \eta_t\nabla_{\boldsymbol{w}}\mathcal{R}_\lambda$. Then, for any $0 \leq s \lesssim \frac{\gamma}{m\eta_t a^2}$,

$$\mathbb{E}\big[\exp\big(s\|\boldsymbol{w}^{t+1}\|_2^2\big) \,|\, \mathcal{F}_0\big] = \mathbb{E}\big[\exp\big(s\|\boldsymbol{\omega}^t\|_2^2 - 2s\eta_t\langle\boldsymbol{\omega}^t, \mathfrak{e}^t\rangle + s\eta_t^2\|\mathfrak{e}^t\|_2^2\big) \,|\, \mathcal{F}_0\big]$$
$$\leq \mathbb{E}\Big[\exp\big(s\|\boldsymbol{\omega}^t\|_2^2\big)\,\mathbb{E}\big[\exp\big(-4s\eta_t\langle\boldsymbol{\omega}^t, \mathfrak{e}^t\rangle\big) \,|\, \mathcal{F}_t\big]^{\frac{1}{2}}\,\mathbb{E}\big[\exp\big(2s\eta_t^2\|\mathfrak{e}^t\|_2^2\big) \,|\, \mathcal{F}_t\big]^{\frac{1}{2}} \,|\, \mathcal{F}_0\Big].$$

By sub-Gaussianity of $\langle\boldsymbol{\omega}^t, \mathfrak{e}^t\rangle$ we have $\mathbb{E}[\exp(-4s\eta_t\langle\boldsymbol{\omega}^t, \mathfrak{e}^t\rangle) \,|\, \mathcal{F}_t] \leq \exp(Cs^2\eta_t^2 a^2\|\boldsymbol{\omega}^t\|_2^2)$. Moreover, as $\|\nabla\ell\|_2 \leq |a|\|\boldsymbol{x}\|_2$, by Jensen's inequality

$$\|\mathfrak{e}^t\|_2^2 \leq 2\|\nabla\ell\|_2^2 + 2\,\mathbb{E}\big[\|\nabla\ell\|_2^2\big] \leq 2a^2(\|\boldsymbol{x}\|_2^2 + d).$$

Thus $\mathbb{E}\big[\exp(2s\eta_t^2\|\mathfrak{e}^t\|_2^2) \,|\, \mathcal{F}_t\big] \leq \exp(Cs\eta_t^2 a^2 d)$ for $s \lesssim \frac{1}{\eta_t^2 a^2}$ (which holds by $s \lesssim \frac{\gamma}{m\eta_t a^2}$, see the proof of Lemma 12 for more details), i.e. we have

$$\mathbb{E}\big[\exp\big(s\|\boldsymbol{w}^{t+1}\|_2^2\big) \,|\, \mathcal{F}_0\big] \leq \mathbb{E}\big[\exp\big(s(1 + Cs\eta_t^2 a^2)\|\boldsymbol{\omega}^t\|_2^2 + Cs\eta_t^2 a^2 d\big) \,|\, \mathcal{F}_0\big].$$

Recall that by our choice of $\eta_t$, $0 \leq (1 - \eta_t(h(\boldsymbol{w}^t) + \lambda)) \leq 1 - \frac{\eta_t\gamma}{m}$ (cf. proof of Lemma 12), and $\boldsymbol{\omega}^t = (1 - \eta_t(h(\boldsymbol{w}^t) + \lambda))\boldsymbol{w}^t - \eta_t\mathfrak{d}(\boldsymbol{w}^t)\boldsymbol{u}$. As $\|\boldsymbol{u}\|_2 = 1$ and $|\mathfrak{d}(\boldsymbol{w}^t)| \lesssim |a|$, we have

$$\|\boldsymbol{\omega}^t\|_2^2 \leq (1 - \tfrac{\eta_t\gamma}{m})^2\|\boldsymbol{w}^t\|_2^2 + Ca^2\eta_t^2 + 2\eta_t Ca(1 - \tfrac{\eta_t\gamma}{m})\|\boldsymbol{w}^t\|_2$$
$$\overset{(a)}{\leq} (1 - \tfrac{\eta_t\gamma}{m})^2\|\boldsymbol{w}^t\|_2^2 + \eta_t\left(\frac{4Cma^2}{\gamma} + \frac{\gamma}{4m}(1 - \tfrac{\eta_t\gamma}{m})^2\|\boldsymbol{w}^t\|_2^2\right) + Ca^2\eta_t^2$$
$$\overset{(b)}{\leq} (1 - \tfrac{3\eta_t\gamma}{2m})\|\boldsymbol{w}^t\|_2^2 + \frac{Cm\eta_t a^2}{\gamma} + Ca^2\eta_t^2.$$

where (a) holds by Young's inequality and (b) holds for $\eta_t\gamma/m \lesssim 1$ with a sufficiently small absolute constant. Therefore, for $s \lesssim \frac{\gamma}{m\eta_t a^2}$,

$$\mathbb{E}\big[\exp\big(s\|\boldsymbol{w}^{t+1}\|_2^2\big) \,|\, \mathcal{F}_0\big] \leq \mathbb{E}\left[\exp\left(s(1 - \tfrac{\eta_t\gamma}{m})\|\boldsymbol{w}^t\|_2^2 + \frac{Csm\eta_t a^2}{\gamma} + Cs\eta_t^2 a^2 d\right) \,|\, \mathcal{F}_0\right].$$

Notice that we can expand the recursion since $s(1 - \frac{\eta_t\gamma}{m}) \lesssim \frac{\gamma}{m\eta_{t-1}a^2}$ (cf. proof of Lemma 12, Eq. (B.5)). Expanding the recursion yields,

$$\mathbb{E}\big[\exp\big(s\|\boldsymbol{w}^t\|_2^2\big) \,|\, \mathcal{F}_0\big] \leq \exp\left(s\left(\frac{t^*}{t^* + t}\right)^2\|\boldsymbol{w}^0\|_F^2 + \frac{Csm^2 a^2(t + d)}{\gamma^2(t^* + t)}\right).$$

Finally, we apply a Chernoff bound with the maximum choice of $s \lesssim \frac{\gamma}{m\eta_t a^2}$, and combine it with the fact that $\|\boldsymbol{w}^0\|_2 \lesssim 1$ with probability at least $1 - \exp(-Cd)$. $\qquad\square$

**Lemma 20.** *Suppose $mab < 1 - |f(0)|$. Then, we have $|\langle\boldsymbol{w}^t, \boldsymbol{u}\rangle| \gtrsim 1$ with probability at least $1 - 2\exp(-Ct) - \exp(-Cd)$.*

**Proof.** We will only prove for the case where $f$ is increasing as the case for decreasing $f$ is similar. We begin by proving an upper bound for $\mathfrak{d}(\boldsymbol{w})$ when $\|\boldsymbol{w}\|_2 \lesssim 1$. By the triangle inequality,

$$|\hat{y} - y| \le |\hat{y}| + |f(0)| + |f(\langle \boldsymbol{u}, \boldsymbol{x} \rangle) - f(0)| + |\epsilon|.$$

Furthermore, $|\hat{y}| \le ma(|\langle \boldsymbol{w}, \boldsymbol{x} \rangle| + b)$. Thus, for

$$|\langle \boldsymbol{w}, \boldsymbol{x} \rangle| \le \left( \frac{1 - |f(0)| - mab}{2ma} \right) \wedge b,$$

$|\langle \boldsymbol{u}, \boldsymbol{x} \rangle| \lesssim 1$ and $|\epsilon| \lesssim 1$ for sufficiently small absolute constants, we have $|\hat{y} - y| \le 1$ hence $\partial_{12}^2 \ell(\hat{y}, y) = -1$. Then we have,

$$
\begin{aligned}
\mathfrak{d}(\boldsymbol{w}) &= a\, \mathbb{E}\big[\partial_{12}^2 \ell(\hat{y}, y) \sigma'(\langle \boldsymbol{w}, \boldsymbol{x} \rangle + b) f'(\langle \boldsymbol{u}, \boldsymbol{x} \rangle)\big] \\
&\lesssim -a\, \mathbb{E}[\mathbf{1}(|\epsilon| \lesssim 1)\mathbf{1}(|\langle \boldsymbol{w}, \boldsymbol{x} \rangle| \lesssim 1)\mathbf{1}(|\langle \boldsymbol{u}, \boldsymbol{x} \rangle| \lesssim 1) f'(\langle \boldsymbol{u}, \boldsymbol{x} \rangle)] \\
&= -a\, \mathbb{E}[\mathbf{1}(|\epsilon| \lesssim 1)]\, \mathbb{E}[\mathbf{1}(|\langle \boldsymbol{w}, \boldsymbol{x} \rangle| \lesssim 1)\mathbf{1}(|\langle \boldsymbol{u}, \boldsymbol{x} \rangle| \lesssim 1) f'(\langle \boldsymbol{u}, \boldsymbol{x} \rangle)] \\
&\lesssim -a.
\end{aligned}
$$

where the last line is obtained by considering supremum over $\|\boldsymbol{w}\|_2 \lesssim 1$.

Let $A_t = \{\sup_{t_0 \le t' \le t} \|\boldsymbol{w}^{t'}\|_2 \lesssim 1\}$. Then,

$$
\begin{aligned}
\mathbb{E}\big[\exp\big(-s\langle \boldsymbol{w}^{t+1}, \boldsymbol{u} \rangle\big)\mathbf{1}_{A_{t+1}}\big] &\le \mathbb{E}\big[\exp\big(-s\langle \boldsymbol{w}^{t+1}, \boldsymbol{u} \rangle\big)\mathbf{1}_{A_t}\big] \\
&= \mathbb{E}\big[\exp\big(-s\langle \boldsymbol{w}^t, \boldsymbol{u} \rangle + s\eta_t \langle \nabla \ell + \lambda \boldsymbol{w}^t, \boldsymbol{u} \rangle\big)\mathbf{1}_{A_t}\big] \\
&\overset{(a)}{\le} \mathbb{E}\big[\exp\big(-s\langle \boldsymbol{w}^t, \boldsymbol{u} \rangle + s\eta_t \langle \nabla \mathcal{R}_\lambda, \boldsymbol{u} \rangle + Cs^2 \eta_t^2 a^2\big)\mathbf{1}_{A_t}\big] \\
&\overset{(b)}{=} \mathbb{E}\big[\exp\big(-s(1 - \eta_t(h(\boldsymbol{w}^t) + \lambda))\langle \boldsymbol{w}^t, \boldsymbol{u} \rangle + s\eta_t(\mathfrak{d}(\boldsymbol{w}^t) + Cs\eta_t a^2)\big)\mathbf{1}_{A_t}\big] \\
&\overset{(c)}{\le} \exp(-Cs\eta_t a)\, \mathbb{E}\big[\exp\big(-s(1 - \eta_t(h(\boldsymbol{w}^t) + \lambda))\langle \boldsymbol{w}^t, \boldsymbol{u} \rangle\big)\mathbf{1}_{A_t}\big],
\end{aligned}
$$

where (a) follows from the sub-Gausianity of the stochastic noise in the gradient, (b) follows since $\langle \nabla \mathcal{R}_\lambda(\boldsymbol{w}^t), \boldsymbol{u} \rangle = \mathfrak{d}(\boldsymbol{w}^t)$ by definition, and (c) holds for $s \lesssim (\eta_t a)^{-1}$ with a sufficiently small absolute constant. Notice that by the condition on $t^*$ inherited from Theorem 3, $1 - \eta_t(h(\boldsymbol{w}^t) + \lambda)) > 0$, and since $s(1 - \eta_t(h(\boldsymbol{w}^t) + \lambda)) \le s(1 - \frac{\eta_t \gamma}{m})$, we can expand the recursion,

$$
\begin{aligned}
\mathbb{E}\big[\exp\big(-s\langle \boldsymbol{w}^t, \boldsymbol{u} \rangle\big)\mathbf{1}_{A_t}\big] &\le \mathbb{E}\left[\exp\left(-Csa \sum_{i=t_0}^{t-1} \eta_i \prod_{j=i+1}^{t-1} (1 - \tfrac{\eta_j \gamma}{m}) + s \prod_{i=t_0}^{t-1} (1 - \tfrac{\eta_i \gamma}{m})|\langle \boldsymbol{w}^{t_0}, \boldsymbol{u} \rangle|\right)\mathbf{1}_{A_{t_0}}\right] \\
&\le \mathbb{E}\left[\exp\left(-Cs\left(1 - \left(\frac{t^* + t_0}{t^* + t}\right)^2\right) + Cs\left(\frac{t^* + t_0}{t^* + t}\right)^2\right)\right].
\end{aligned}
$$

where in the second inequality we used $a \asymp m^{-1}$ and $\gamma \asymp 1$. Applying the Chernoff bound implies that $\langle \boldsymbol{w}^t, \boldsymbol{u} \rangle \gtrsim 1$ with probability at least $1 - \mathbb{P}(A_t^C) - \exp(-Ct) \ge 1 - \exp(-C(t^* + t_0)) - \exp(-Cd) - \exp(-Ct)$. Finally the result follows by letting $t_0 = Ct$ for a sufficiently small absolute constant $C$. □

We have proven that $|\langle \boldsymbol{w}^t, \boldsymbol{u} \rangle| \gtrsim 1$ while $\|\boldsymbol{w}_\perp^t\|_2 \to 0$. This fact shows that the features learned in the first layer are useful. What remains to be shown is an approximation result, such that for a carefully constructed second layer, the network can approximate polynomials of the desired type. This type of approximation using random biases has been adopted from Damian et al. (2022). We first present an approximation result using infinite neurons.

**Lemma 21.** *Let $0 < |\alpha| \le r$ and $b \sim \mathrm{Unif}(-2r\Delta, 2r\Delta)$. For any smooth $f : \mathbb{R} \to \mathbb{R}$, let $\tilde{f}_\alpha : \mathbb{R} \to \mathbb{R}$ be a smooth function such that $\tilde{f}_\alpha(z) = f(z)$ for $|z| \le \frac{r\Delta}{|\alpha|}$ and $\tilde{f}_\alpha(-\frac{2r\Delta}{\alpha}) = \tilde{f}_\alpha'(-\frac{2r\Delta}{\alpha}) = 0$. Then, for $|z| \le \Delta$ we have*

$$\mathbb{E}_b\left[\frac{4r\Delta}{\alpha^2} \tilde{f}_\alpha''\left(-\frac{b}{\alpha}\right)\sigma(\alpha z + b)\right] = f(z).$$

**Proof.** Using integration by parts, we have

$$
\begin{aligned}
\mathbb{E}_b\left[\frac{4r\Delta}{\alpha^2}\tilde{f}''_\alpha\left(-\frac{b}{\alpha}\right)\sigma(\alpha z+b)\right] &= \int_{-\alpha z}^{2r\Delta}\tilde{f}''_\alpha(-\frac{b}{\alpha})(z+\frac{b}{\alpha})\frac{\mathrm{d}b}{\alpha} \\
&= -\tilde{f}'_\alpha(-\frac{2r\Delta}{\alpha})(z+\frac{2r\Delta}{\alpha}) + \int_{-\frac{2r\Delta}{\alpha}}^{z}\tilde{f}'_\alpha(b)\,\mathrm{d}b \\
&= \tilde{f}_\alpha(z) = f(z).
\end{aligned}
$$

$\square$

Now, by a concentration argument, we state an approximation result with finitely many neurons.

**Lemma 22.** *Let $r^* \le |\alpha_j| \le r$ and $b_j \sim \mathrm{Unif}(-2r\Delta, 2r\Delta)$. Let*

$$
\Delta_* := \Delta \sup_{j, |z| \le \frac{2r\Delta}{r^*}} |\tilde{f}''_{\alpha_j}(z)|, \tag{C.7}
$$

*where $\tilde{f}_{\alpha_j}$ is the extension of $f_{\alpha_j}$ introduced in Lemma 21. Then there exists $a(\alpha_j, b_j)$ such that for any fixed $z \in [-\Delta, \Delta]$, with probability at least $1 - \delta$ over the choice of $(b_j)$, we have*

$$
\left|\sum_{j=1}^m a(\alpha_j, b_j)\sigma(\alpha_j z + b_j) - f(z)\right| \lesssim \frac{r^2\Delta\Delta_*}{r^{*2}}\sqrt{\frac{\log(1/\delta)}{m}}.
$$

*Moreover, $\|a\|_2 \lesssim \frac{r\Delta_*}{r^{*2}\sqrt{m}}$.*

**Proof.** Let $\tilde{f}_\alpha(z)$ be a candidate in Lemma 21, which can be obtained by e.g. extending $f$ with suitable polynomials (notice that $\tilde{f}_\alpha$ only needs to be twice differentiable on its domain). Now choose $a_j = 4\frac{r\Delta}{\alpha_j^2 m}\tilde{f}''_{\alpha_j}(-\frac{b_j}{\alpha_j})$. Then Lemma 21 ensures that

$$
\mathbb{E}_{b_j}[a(\alpha_j, b_j)\sigma(\alpha_j z + b_j)] = f(z).
$$

It immediately follows that $\|a\|_2 \le \frac{Cr\Delta_*}{r^{*2}\sqrt{m}}$ and $|a_j\sigma(\alpha z + b_j)| \le \frac{Cr^2\Delta\Delta_*}{r^{*2}m}$. Applying the Hoeffding's inequality finishes the proof. $\square$

In the following lemma, we will briefly record useful properties of $\boldsymbol{W}^T$ which will be of help for invoking the above approximation results and providing guarantees when the second layer is optimized by SGD. Through the rest of the proof, we will add the mild assumption that $d \gtrsim \log(1/\delta)$. Otherwise, we need to add $e^{-Cd}$ to the probability of failure in Theorem 4.

**Lemma 23.** *Suppose $T \gtrsim d + \log(1/\delta)$. Then with probability at least $1 - \delta$ over the choice of $(b_j)_{1\le j\le m}$ and $\{(\boldsymbol{x}^{(t)}, y^{(t)})\}_{t=0}^{T-1}$, the following statements hold:*

1. $\|\boldsymbol{w}_j^T\|_2 \asymp |\langle \boldsymbol{w}_j^T, \boldsymbol{u}\rangle| \asymp 1$ *for all $1 \le j \le m$.*

2. $\|\frac{1}{T}\sum_{t=0}^{T-1}\boldsymbol{x}^{(t)}(\boldsymbol{x}^{(t)})^\top\|_2 \lesssim 1$.

3. $\|\boldsymbol{W}_\perp^T\|_F \lesssim \sqrt{\frac{m(d+\log(1/\delta))}{T}}$.

4. $|\langle \boldsymbol{u}, \boldsymbol{x}^{(t)}\rangle| \lesssim \Delta$ *for all $0 \le t \le T - 1$.*

5. $\|\boldsymbol{W}^T x^{(t)}\|_2 \lesssim \sqrt{m}(\sqrt{d} + \Delta)$ *for all $0 \le t \le T - 1$.*

**Proof.** We will show that each of the events holds with probability (w.p.) at least $1 - \mathcal{O}(\delta)$. Recall from Lemma 19 that $\|\boldsymbol{w}_j^T\|_2 \lesssim 1$ for all $j$ w.p. $\ge 1 - \mathcal{O}(\delta)$, which implies the same for $\langle \boldsymbol{w}_j^T, \boldsymbol{u}\rangle$. On the other hand, from Lemma 20, $|\langle \boldsymbol{w}_j^T, \boldsymbol{u}\rangle| \gtrsim 1$ for all $j$ w.p. $\ge 1 - \mathcal{O}(\delta)$. Combining these events implies that $|\langle \boldsymbol{w}_j^T, \boldsymbol{u}\rangle| \asymp 1$. The fact that $\|\frac{1}{T}\sum_{t=0}^{T-1}\boldsymbol{x}^{(t)}(\boldsymbol{x}^{(t)})^\top\|_2 \lesssim 1$ w.p. $\ge 1 - \mathcal{O}(\delta)$ for $T \gtrsim$

$d + \log(1/\delta)$ follows from the statement of Lemma 30. Furthermore, $\|\boldsymbol{W}_\perp^T\|_{\mathrm{F}} \lesssim \sqrt{\frac{m(d+\log(1/\delta))}{T}}$ w.p. $1 - \mathcal{O}(\delta)$ follows from Theorem 3. Note that as $\langle \boldsymbol{u}, \boldsymbol{x}^{(t)} \rangle \sim \mathcal{N}(0,1)$, by the choice of $\Delta$, we have $\langle \boldsymbol{u}, \boldsymbol{x}^{(t)} \rangle \gtrsim \Delta$ w.p. $\leq \mathcal{O}(\delta/T)$, thus $|\langle \boldsymbol{u}, \boldsymbol{x}^{(t)} \rangle| \lesssim \Delta$ for any $0 \leq t \leq T - 1$ w.p. $\geq 1 - \mathcal{O}(\delta)$ by a union bound. Finally, we have

$$\|\boldsymbol{W}^T \boldsymbol{x}^{(t)}\|_2 \leq \|\boldsymbol{W}_\|^T \boldsymbol{x}^{(t)}\|_2 + \|\boldsymbol{W}_\perp^T \boldsymbol{x}^{(t)}\|_2 \lesssim \sqrt{m}|\langle \boldsymbol{u}, \boldsymbol{x}^{(t)} \rangle| + \sqrt{\frac{m(d + \log(1/\delta))}{T}}\|\boldsymbol{x}^{(t)}\|_2$$
$$\lesssim \sqrt{m}|\langle \boldsymbol{u}, \boldsymbol{x}^{(t)} \rangle| + \sqrt{m}\|\boldsymbol{x}^{(t)}\|_2$$

The first term is already bounded by $\sqrt{m}\Delta$ with probability at least $1 - \mathcal{O}(\delta)$. Moreover, recall that $\|\boldsymbol{x}^{(t)}\|_2 - \mathbb{E}[\|\boldsymbol{x}^{(t)}\|_2]$ is 1-sub-Gaussian, thus by the union bound $\|\boldsymbol{x}^{(t)}\|_2 - \sqrt{d} \lesssim \sqrt{\log(T/\delta)} \lesssim \Delta$ for all $0 \leq t \leq T - 1$. Thus w.p. $\geq 1 - \mathcal{O}(\delta)$ we have $\|\boldsymbol{W}^T \boldsymbol{x}^{(t)}\|_2 \lesssim \sqrt{m}(\sqrt{d} + \Delta)$ which completes the proof. $\qquad\square$

From this point onwards, we will denote the Huber loss with $\ell_{\mathrm{H}}(\hat{y}, y) = \ell(\hat{y} - y)$. Notice that $\ell_{\mathrm{H}}$ is 1-Lischitz.

**Lemma 24.** *Recall*

$$\hat{R}(\boldsymbol{W}^T, \boldsymbol{a}, \boldsymbol{b}) = \frac{1}{T}\sum_{t=0}^{T-1} \ell_{\mathrm{H}}\left(\sum_{j=1}^m a_j \sigma(\langle \boldsymbol{w}_j^T, \boldsymbol{x}^{(t)} \rangle + b_j) - f(\langle \boldsymbol{u}, \boldsymbol{x}^{(t)} \rangle) - \epsilon^{(t)}\right),$$

*the empirical risk of $\boldsymbol{W}^T$ given by Algorithm 1. Let $\Delta \asymp \sqrt{\log(\frac{T}{\delta})}$, $\Delta_*$ as defined in (C.7), and $b_j \overset{i.i.d.}{\sim} \mathrm{Unif}(-\Delta, \Delta)$. Then, with probability at least $1 - \delta$ (over the randomness of $(b_j)_{1 \leq j \leq m}$ and $\{\boldsymbol{x}^{(t)}, y^{(t)}\}_{t=0}^{T-1}$ hence $\boldsymbol{W}^T$), for $T \gtrsim d + \log(1/\delta)$, there exists $\boldsymbol{a}^*$ with $\|\boldsymbol{a}^*\|_2 \lesssim \frac{\Delta_*}{\sqrt{m}}$ such that*

$$\hat{R}(\boldsymbol{W}^T, \boldsymbol{a}^*, \boldsymbol{b}) - \mathbb{E}[\ell_H(\epsilon)] \lesssim \Delta_*\left(\Delta\sqrt{\frac{\log(T/\delta)}{m}} + \Delta_*\sqrt{\frac{d + \log(1/\delta)}{T}}\right) + \nu\sqrt{\frac{\log(1/\delta)}{T}}.$$

**Proof.** We will condition the following discussion on the event of Lemma 23. Let $\alpha_j = \langle \boldsymbol{w}_j^T, \boldsymbol{u} \rangle$, and let $\boldsymbol{a}^*$ be constructed according to Lemma 22. By the Lipschitzness of the Huber loss, for an inividual sample $(\boldsymbol{x}, y)$ we have

$$\ell_{\mathrm{H}}(\hat{y}(\boldsymbol{x}; \boldsymbol{W}^T, \boldsymbol{a}^*, \boldsymbol{b}) - f(\langle \boldsymbol{u}, \boldsymbol{x} \rangle) - \epsilon \leq \ell_{\mathrm{H}}(\epsilon) + |\hat{y}(\boldsymbol{x}; \boldsymbol{W}^T, \boldsymbol{a}^*, \boldsymbol{b}) - f(\langle \boldsymbol{u}, \boldsymbol{x} \rangle)|$$
$$\leq \ell_{\mathrm{H}}(\epsilon) + |\hat{y}(\boldsymbol{x}; \boldsymbol{W}^T, \boldsymbol{a}^*, \boldsymbol{b}) - \hat{y}(\boldsymbol{x}; \boldsymbol{W}_\|^T, \boldsymbol{a}^*, \boldsymbol{b})|$$
$$+ |\hat{y}(x; \boldsymbol{W}_\|^T, \boldsymbol{a}^*, \boldsymbol{b}) - f(\langle \boldsymbol{u}, \boldsymbol{x} \rangle)|.$$

Moreover, by the Cauchy-Schwartz inequality

$$|\hat{y}(\boldsymbol{x}; \boldsymbol{W}^T, \boldsymbol{a}^*, \boldsymbol{b}) - \hat{y}(\boldsymbol{x}; \boldsymbol{W}_\|^T, \boldsymbol{a}^*, \boldsymbol{b})| \leq \|\boldsymbol{a}^*\|_2\sqrt{\sum_{j=1}^m \left(\sigma(\langle \boldsymbol{w}_j^T, \boldsymbol{x} \rangle + b_j) - \sigma(\langle (\boldsymbol{w}_j^T)_\|, \boldsymbol{x} \rangle + b_j)\right)^2}$$
$$\leq \|\boldsymbol{a}^*\|_2\sqrt{\sum_{j=1}^m \langle (\boldsymbol{w}_j^T)_\perp, \boldsymbol{x} \rangle^2}.$$

Additionally, since $\|\frac{1}{T}\sum_{t=0}^{T-1} \boldsymbol{x}^{(t)}(\boldsymbol{x}^{(t)})^\top\|_2 \lesssim 1$, by Jensen's inequality,

$$\sum_{t=0}^{T-1}\frac{1}{T}|\hat{y}(\boldsymbol{x}; \boldsymbol{W}^T, \boldsymbol{a}^*, \boldsymbol{b}) - \hat{y}(\boldsymbol{x}; \boldsymbol{W}_\|^T, \boldsymbol{a}^*, \boldsymbol{b})| \leq \|\boldsymbol{a}^*\|_2\sqrt{\frac{1}{T}\sum_{t=0}^{T-1}\|\boldsymbol{W}_\perp^T x^{(t)}\|_{\mathrm{F}}^2}$$
$$\lesssim \|\boldsymbol{a}^*\|_2\|\boldsymbol{W}_\perp^T\|_{\mathrm{F}}$$
$$\lesssim \Delta_*\sqrt{\frac{d + \log(1/\delta)}{T}}$$

On the other hand, let $z^{(t)} := \langle \boldsymbol{u}, \boldsymbol{x}^{(t)} \rangle \lesssim \Delta$. Then, we can apply Lemma 22 along with a union bound, which states that with probability $1 - \mathcal{O}(\delta)$ over the choice of $(b_j)_{1 \leq j \leq m}$,

$$
\frac{1}{T} \sum_{t=0}^{T-1} |\hat{y}(\boldsymbol{x}^{(t)}; \boldsymbol{W}_\parallel^T, \boldsymbol{a}^*, \boldsymbol{b}) - f(\langle \boldsymbol{u}, \boldsymbol{x}^{(t)} \rangle)| \leq \frac{1}{T} \sum_{t=0}^{T-1} |\sum_{j=1}^m a_j^* \sigma(\alpha_j z^{(t)} + b_j) - f(z^{(t)})|
$$

$$
\lesssim \Delta \Delta_* \sqrt{\frac{\log(T/\delta)}{m}}.
$$

Combining the events above, we have with probability at least $1 - \delta$,

$$
\hat{R}(\boldsymbol{W}^T, \boldsymbol{a}^*, \boldsymbol{b}) - \frac{1}{T} \sum_{t=0}^{T-1} \ell_{\mathrm{H}}(\epsilon^{(t)}) \lesssim \Delta_* \left( \Delta \sqrt{\frac{\log(T/\delta)}{m}} + \sqrt{\frac{d + \log(1/\delta)}{T}} \right).
$$

The final step is to apply a concentration bound for $\sum_{t=0}^{T-1} \ell_{\mathrm{H}}(\epsilon^{(t)})$. Note that as $\ell_{\mathrm{H}}(\epsilon) \leq |\epsilon|$, if $|\epsilon|$ is $\nu$-sub-Gaussian, then $\ell_{\mathrm{H}}(\epsilon) - \mathbb{E}[\ell_{\mathrm{H}}(\epsilon)]$ is also $C\nu$-sub-Gaussian (can be verified e.g. by Lemma 28). Then a sub-Gaussian concentration bound implies that $\mathbb{E}[\ell_{\mathrm{H}}(\epsilon)] - \frac{1}{T} \sum_{t=0}^{T-1} \ell_{\mathrm{H}}(\epsilon^{(t)}) \lesssim \nu \sqrt{\frac{\log(1/\delta)}{T}}$, which finishes the proof. $\qquad \square$

Let $\mathbb{E}_S[\cdot]$ denote expectation w.r.t. the random sampling of SGD used to train $a$, hence conditioned on $\{\boldsymbol{x}^{(t)}, y^{(t)}\}_{t=0}^{T-1}$. Also, define the stochastic noise in the gradient w.r.t. $a$ as

$$
\mathfrak{e}_a^t = \nabla_{\boldsymbol{a}} \ell(\hat{y}(\boldsymbol{x}^{(i_t)}; \boldsymbol{W}^T, \boldsymbol{a}^t, \boldsymbol{b}) - y^{(i_t)}) - \nabla_{\boldsymbol{a}} \hat{R}(\boldsymbol{W}^T, \boldsymbol{a}^t, \boldsymbol{b}).
$$

Notice that $\mathbb{E}_S[\mathfrak{e}_a^t] = 0$.

**Lemma 25.** *On the event of Lemma 23 and with $(b_j) \overset{i.i.d.}{\sim} \mathrm{Unif}(-\Delta, \Delta)$, consider the mapping $\boldsymbol{a} \mapsto \hat{\mathcal{R}}_{\lambda'}(\boldsymbol{a})$. Then, $\nabla_{\boldsymbol{a}}^2 \hat{\mathcal{R}}_{\lambda'}(\boldsymbol{a}) \precsim m\Delta^2 + \lambda'$, and $\|\mathfrak{e}_a^t\|_2 \lesssim \sqrt{m}(\sqrt{d} + \Delta)$.*

**Proof.** For $\nabla_{\boldsymbol{a}}^2 \hat{R}(\boldsymbol{a})$, and any $\boldsymbol{v} \in \mathbb{R}^m$ with $\|\boldsymbol{v}\|_2 = 1$, we have the following computation:

$$
\left\langle \boldsymbol{v}, \nabla_{\boldsymbol{a}}^2 \hat{\mathcal{R}}_{\lambda'}(\boldsymbol{a}) \boldsymbol{v} \right\rangle = \frac{1}{T} \sum_{t=0}^{T-1} \partial_1^2 \ell(\hat{y}, y) \boldsymbol{v}^\top \sigma(\boldsymbol{W}^T \boldsymbol{x}^{(t)} + \boldsymbol{b}) \sigma(\boldsymbol{W}^T \boldsymbol{x}^{(t)} + \boldsymbol{b})^\top \boldsymbol{v} + \lambda'
$$

$$
\leq \frac{1}{T} \sum_{t=0}^{T-1} \|\sigma(\boldsymbol{W}^T \boldsymbol{x}^{(t)} + \boldsymbol{b})\|^2 + \lambda'
$$

$$
\overset{(a)}{\lesssim} \|\boldsymbol{W}^T\|_{\mathrm{F}} + \|\boldsymbol{b}\|_2^2 + \lambda'
$$

$$
\lesssim m\Delta^2 + \lambda'
$$

where (a) holds since $\|\frac{1}{T} \sum_{t=0}^{T-1} \boldsymbol{x}^{(t)} \boldsymbol{x}^{(t)\top}\|_2 \lesssim 1$. Thus $\nabla_{\boldsymbol{a}}^2 \hat{\mathcal{R}}_{\lambda'}(\boldsymbol{a}) \precsim m\Delta^2 + \lambda'$. On the other hand, as $\|\boldsymbol{W}^T \boldsymbol{x}^{(t)}\|_2 \lesssim \sqrt{m}(\sqrt{d} + \Delta)$ for all $0 \leq t \leq T-1$, we have

$$
\|\mathfrak{e}_a^t\|_2 \leq 2\|\nabla_{\boldsymbol{a}} \ell\|_2 \leq 2\|\boldsymbol{W}^T \boldsymbol{x}^{(t)} + \boldsymbol{b}\|_2 \lesssim \sqrt{m}(\sqrt{d} + \Delta).
$$

$\qquad \square$

Now we can analyze the SGD run on the second layer $\boldsymbol{a}$ to give a high probability statement for $\hat{\mathcal{R}}_{\lambda'}(\boldsymbol{a}^T)$. As $\hat{\mathcal{R}}_{\lambda'}(\boldsymbol{a})$ is a smooth and strongly convex function of $a$, we will state the following well-known elementary convergence result of SGD for smooth and strongly convex functions with bounded noise, which we present in a high-probability framework suitable for our analysis.

**Lemma 26.** *Let $\mathcal{R} : \mathbb{R}^m \to \mathbb{R}$ be a $\mu$-strongly convex function satisfying $\mu \mathbf{I}_m \preceq \nabla_{\boldsymbol{a}}^2 \mathcal{R}(\boldsymbol{a}) \preceq L\mathbf{I}_m$. Suppose we run the SGD iterates $\boldsymbol{a}^{t+1} = \boldsymbol{a}^t - \eta_t \mathfrak{g}^t$ with $\mathbb{E}[\mathfrak{g}^t \mid \boldsymbol{a}^t] = \nabla_{\boldsymbol{a}} \mathcal{R}(\boldsymbol{a}^t)$ and $\|\mathfrak{g}^t\|_2 \leq G$. Choose $\eta_t = \frac{2t+1}{\mu(t+1)^2}$. Then with probability at least $1 - \delta$*

$$
\mathcal{R}(\boldsymbol{a}^T) - \mathcal{R}^* \leq \frac{\mathcal{R}^0}{T^2} + \frac{CLG^2}{\mu^2 T} + \frac{CG^2 \log(1/\delta)}{\mu T},
$$

*where $\mathcal{R}^* = \arg\min_{\boldsymbol{a}} \mathcal{R}(\boldsymbol{a})$.*

**Proof.** Let $\mathfrak{e}^t = \mathfrak{g}^t - \nabla_{\boldsymbol{a}}\mathcal{R}(\boldsymbol{a}^t)$ denote the stochastic noise. By the smoothness property of $\mathcal{R}$, we have

$$\mathcal{R}(\boldsymbol{a}^{t+1}) - \mathcal{R}^* \leq \mathcal{R}(\boldsymbol{a}^t) - \mathcal{R}^* - \eta_t\langle\nabla_{\boldsymbol{a}}\mathcal{R}(\boldsymbol{a}^t), \nabla_{\boldsymbol{a}}\mathcal{R}(\boldsymbol{a}^t) + \mathfrak{e}^t\rangle + \frac{L\eta_t^2}{2}\|\mathfrak{g}^t\|_2^2$$

$$\leq \mathcal{R}(\boldsymbol{a}^t) - \mathcal{R}^* - \eta_t\|\nabla_{\boldsymbol{a}}\mathcal{R}(\boldsymbol{a}^t)\|_2^2 - \eta_t\langle\nabla_{\boldsymbol{a}}\mathcal{R}(\boldsymbol{a}^t), \mathfrak{e}^t\rangle + \frac{L\eta_t^2 G^2}{2}.$$

Notice that by Jensen's inequality, $\|\nabla_{\boldsymbol{a}}\mathcal{R}(\boldsymbol{a}^t)\|_2 \leq G$, thus $\|\mathfrak{e}^t\|_2 \leq 2G$ and the zero-mean random variable $\langle\nabla_{\boldsymbol{a}}\mathcal{R}(\boldsymbol{a}^t), \mathfrak{e}^t\rangle$ is $2G\|\nabla_{\boldsymbol{a}}\mathcal{R}(\boldsymbol{a}^t)\|_2$-sub-Gaussian conditioned on $\boldsymbol{a}^t$. Now, we can establish the following recursive bound on the MGF of $\mathcal{R}^t := \mathcal{R}(\boldsymbol{a}^t) - \mathcal{R}^*$. For $0 \leq s \leq \frac{1}{4\eta_t G^2}$ we have

$$\mathbb{E}\left[e^{s\mathcal{R}^{t+1}}\right] \leq \mathbb{E}\left[\exp\left(s\mathcal{R}^t - s\eta_t\|\nabla_{\boldsymbol{a}}\mathcal{R}(\boldsymbol{a}^t)\|_2^2 - s\eta_t\langle\nabla_{\boldsymbol{a}}\mathcal{R}(\boldsymbol{a}^t), \mathfrak{e}^t\rangle + \frac{\eta_t^2 LG^2}{2}\right)\right]$$

$$\leq \mathbb{E}\left[\exp\left(s\mathcal{R}^t - s\eta_t(1 - 2s\eta_t G^2)\|\nabla_{\boldsymbol{a}}\mathcal{R}(\boldsymbol{a}^t)\|_2^2 + \frac{LG^2\eta_t^2}{2}\right)\right]$$

$$\overset{(a)}{\leq} \mathbb{E}\left[\exp\left(s(1 - \eta_t\mu)\mathcal{R}^t + \frac{LG^2\eta_t^2}{2}\right)\right]$$

where (a) follows since $\mathcal{R}(\boldsymbol{a})$ is strongly convex thus satisfies the Polyak-Łojasiewicz inequality $2\mu(\mathcal{R}(\boldsymbol{a}) - \mathcal{R}^*) \leq \|\nabla_{\boldsymbol{a}}\mathcal{R}(\boldsymbol{a})\|_2^2$. As $s(1 - \eta_t\mu) \leq \frac{1}{4\eta_{t-1}G^2}$ (cf. (B.5)), we can expand the recursion and have

$$\mathbb{E}\left[\exp(s\mathcal{R}^t)\right] \leq \exp\left(s\left(\frac{t^*}{t^* + t}\right)^2\mathcal{R}^0 + \frac{16LG^2}{\mu^2(t^* + t)}\right).$$

Finally, applying a Chernoff bound using $s = (4\eta_{t-1}G^2)^{-1}$ concludes the proof. $\qquad\square$

We are finally in a position to complete the proof of Theorem 4.

**Proof. [Proof of Theorem 4]** We will consider the event of Lemma 23 on which from Lemma 24 we know with probabililty at least $1 - \delta$ over the dataset and $(b_j)_{1 \leq j \leq m}$ we have

$$\min_{\boldsymbol{a}:\|\boldsymbol{a}\|_2 \lesssim \frac{\Delta_*}{\sqrt{m}}} \hat{R}(\boldsymbol{W}^T, a, b) - \mathbb{E}[\ell_H(\epsilon)] \lesssim \Delta_*^2\left(\sqrt{\frac{\log(T/\delta)}{m}} + \sqrt{\frac{d + \log(1/\delta)}{T}}\right) + \nu\sqrt{\frac{\log(1/\delta)}{T}}.$$

Notice that $\boldsymbol{a} \mapsto \hat{R}(\boldsymbol{W}, \boldsymbol{a}, \boldsymbol{b})$ is a convex function. Thus by strong duality, there exists $\lambda' > 0$ such that the value of the above constrained minimization problem is equal to the value of the following regularized minimization problem,

$$\min_{\boldsymbol{a}} \hat{\mathcal{R}}_{\lambda'}(\boldsymbol{W}^T, \boldsymbol{a}, \boldsymbol{b}) - \mathbb{E}[\ell_H(\epsilon)] \lesssim \Delta_*^2\left(\sqrt{\frac{\log(T/\delta)}{m}} + \sqrt{\frac{d + \log(1/\delta)}{T}}\right) + \nu\sqrt{\frac{\log(1/\delta)}{T}}.$$

Explicitly, this $\lambda'$ can be chosen such that the unique solution to

$$\nabla_a\hat{R}(\boldsymbol{W}^T, \boldsymbol{a}^*, \boldsymbol{b}) + \lambda'\boldsymbol{a}^* = 0 \tag{C.8}$$

has $\|\boldsymbol{a}^*\|_2 \lesssim \frac{\Delta_*}{\sqrt{m}}$. Notice that this $\boldsymbol{a}^*$ is the unique solution to $\arg\min_{\boldsymbol{a}} \hat{\mathcal{R}}_{\lambda'}(\boldsymbol{W}^T, \boldsymbol{a}, \boldsymbol{b})$.

Moreover, from Lemma 26 we have

$$\hat{\mathcal{R}}_{\lambda'}(\boldsymbol{W}^T, \boldsymbol{a}^{T'}, \boldsymbol{b}) - \hat{\mathcal{R}}_{\lambda'}(\boldsymbol{W}^T, \boldsymbol{a}^*, \boldsymbol{b}) \lesssim \frac{\hat{R}(\boldsymbol{W}^T, \boldsymbol{a}^0, \boldsymbol{b})}{T'^2} + \frac{(d + \Delta^2)(\Delta^2 + \lambda'/m + \log(1/\delta))}{(\lambda'/m)^2 T'},$$

and by strong convexity

$$\|\boldsymbol{a}^{T'} - \boldsymbol{a}^*\|_2^2 \leq \frac{2}{m}\left(\hat{\mathcal{R}}_{\lambda'}(\boldsymbol{W}^T, \boldsymbol{a}^{T'}, \boldsymbol{b}) - \hat{\mathcal{R}}_{\lambda'}(\boldsymbol{W}^T, \boldsymbol{a}^*, \boldsymbol{b})\right).$$

Thus, with sufficiently large $T'$ such that

$$\frac{\hat{R}(\boldsymbol{W}^T, \boldsymbol{a}^0, \boldsymbol{b})}{T'^2} + \frac{(d + \Delta^2)(\Delta^2 + \lambda'/m + \log(1/\delta))}{(\lambda'/m)^2 T'} \lesssim \Delta_*^2\sqrt{\frac{d + \log(1/\delta)}{T}} \wedge \frac{\lambda'\Delta_*}{\sqrt{m}}, \tag{C.9}$$

we have $\|\boldsymbol{a}^{T'}\|_2 \lesssim \frac{\Delta_*}{\sqrt{m}}$ and

$$\hat{\mathcal{R}}_{\lambda'}(\boldsymbol{a}^{T'}) - \mathbb{E}[\ell_{\mathrm{H}}(\epsilon)] \lesssim \Delta_*^2 \left( \sqrt{\frac{\log(T/\delta)}{m}} + \sqrt{\frac{d + \log(1/\delta)}{T}} \right).$$

Finally, we invoke Theorem 5, to close the generalization gap and get

$$R_\tau(\boldsymbol{W}^T, \boldsymbol{a}^{T'}, \boldsymbol{b}) - \mathbb{E}[\ell_{\mathrm{H}}(\epsilon)] \lesssim \Delta_*^2 \left( \sqrt{\frac{\log(T/\delta)}{m}} + \sqrt{\frac{d + \log(1/\delta)}{T}} \right) + \nu \sqrt{\frac{\log(1/\delta)}{T}}.$$

$\square$

## D   EXAMPLE OF NON-CONVEX $\mathcal{R}_\lambda(\boldsymbol{W})$

Here, we outline examples for which $\mathcal{R}_\lambda(\boldsymbol{W})$ is non-convex on a neighborhood around $\boldsymbol{W} = \boldsymbol{0}$ while $\lambda = \frac{\tilde{\lambda}}{m}$ satisfies the condition in Proposition 2 or Theorem 3. For simplicity of exposition, in both examples we fix $\boldsymbol{a} = \frac{\boldsymbol{1}_m}{m}$ where $\boldsymbol{1}_m$ is the vector of all ones. It is easy to observe that the results hold with high probability when $\boldsymbol{a}$ follows the initialization of Assumption 3 as well. Furthermore, we work on the event where $\|\boldsymbol{W}^0\|_{\mathrm{F}} \leq 2\sqrt{m}$, which happens with probability at least $1 - \exp(-md/2)$.

We begin by constructing a non-convex example for Proposition 2. For this example, we choose $\sigma$ such that $\beta_1 \leq 1$, $\sigma(1) = \sigma(-1) = 0$, $\sigma'(1) = \sigma'(-1) = 0$, and $\sigma''(-1) = \sigma''(1) = \beta_2 = 1$. An example of such a function is $\sigma(z) = \frac{\cos(\pi z)+1}{\pi^2}$. Then, using the computations of Lemma 8 we have

$$\nabla_{\boldsymbol{W}}^2 \mathcal{R}_\lambda(\boldsymbol{W}) = \mathbb{E}\left[ (\sigma'_{\boldsymbol{a},\boldsymbol{b}}(\boldsymbol{W}\boldsymbol{x})\sigma'_{\boldsymbol{a},\boldsymbol{b}}(\boldsymbol{W}\boldsymbol{x})^\top + (\hat{y}(\boldsymbol{x}; \boldsymbol{W}, \boldsymbol{a}, \boldsymbol{b}) - y)\operatorname{diag}(\sigma''_{\boldsymbol{a},\boldsymbol{b}}(\boldsymbol{W}\boldsymbol{x}))) \otimes \boldsymbol{x}\boldsymbol{x}^\top \right] + \frac{\tilde{\lambda}}{m}\mathbf{I}_{md}.$$

Which simplifies to

$$\nabla_{\boldsymbol{W}}^2 \mathcal{R}_\lambda(\boldsymbol{0}) = \frac{-\mathbf{I}_m}{m} \otimes \mathbb{E}[y\boldsymbol{x}\boldsymbol{x}^\top] + \lambda\mathbf{I}_{md} = \mathbf{I}_m \otimes \left( \frac{-\mathbb{E}[y\boldsymbol{x}\boldsymbol{x}^\top]}{m} + \frac{\tilde{\lambda}}{m}\mathbf{I}_d \right).$$

Therefore, $\nabla^2\mathcal{R}_\lambda(\boldsymbol{0})$ is not positive semi-definite (PSD) if and only if $\frac{-1}{m}\mathbb{E}[y\boldsymbol{x}\boldsymbol{x}^\top] + \frac{\tilde{\lambda}}{m}\mathbf{I}_d$ is not PSD. Moreover, by Jensen's inequality

$$\hat{y}^2(\boldsymbol{x}; \boldsymbol{W}^0, \boldsymbol{a}, \boldsymbol{b}) \leq \frac{1}{m}\sum_{i=1}^m (\sigma(\langle \boldsymbol{w}_j^0, \boldsymbol{x}\rangle + b_j) - \sigma(b_j))^2 \leq \frac{\|\boldsymbol{W}^0\boldsymbol{x}\|_{\mathrm{F}}^2}{m},$$

hence

$$2R(\boldsymbol{W}^0) \leq 2\mathbb{E}[\hat{y}^2] + 2\mathbb{E}[y^2] \leq 8 + 2\mathbb{E}[y^2].$$

Now, let $y = \mathcal{K}\langle \boldsymbol{w}, \boldsymbol{x}\rangle^2$ for some $\boldsymbol{w}$ with $\|\boldsymbol{w}\|_2 = 1$. Then $\mathbb{E}[y^2] = 3\mathcal{K}^2$, and choosing $\tilde{\lambda} = 1 + \sqrt{9 + 6\mathcal{K}^2} + \vartheta$ for arbitrarily small $\vartheta$ suffices to satisfy the condition of Proposition 2. Then we have

$$\boldsymbol{w}^\top \nabla_{\boldsymbol{W}}^2 \mathcal{R}_\lambda(0)\boldsymbol{w} = \boldsymbol{w}^\top \left( \frac{-\mathbb{E}[y\boldsymbol{x}\boldsymbol{x}^\top] + \tilde{\lambda}}{m} \right)\boldsymbol{w} = \frac{-\mathbb{E}[\mathcal{K}\langle \boldsymbol{w}, \boldsymbol{x}\rangle^4] + \tilde{\lambda}}{m}$$

$$= \frac{-3\mathcal{K} + 1 + \sqrt{9 + 6\mathcal{K}^2} + \vartheta}{m} < 0$$

where the above inequality holds for sufficiently large $\mathcal{K}$, hence $\mathcal{R}_\lambda(\boldsymbol{W})$ is non-convex at least on a neighborhood around zero.

Next, we construct a non-convex example for the smooth and decaying step size case of Theorem 3. This time, we require $\sigma(\pm 1) = -\beta_0 = -1$ (which automatically implies $\sigma'(\pm 1) = 0$ as $\sigma$ attains its minimum) and $\sigma''(\pm 1) = \beta_2 = 1$. For instance, we can choose $\sigma(z) = \frac{\cos(\pi z)-\pi^2+1}{\pi^2}$. Then simplifying $\nabla^2\mathcal{R}_\lambda(\boldsymbol{0})$ yields

$$\nabla_{\boldsymbol{W}}^2 \mathcal{R}_\lambda(\boldsymbol{0}) = \frac{\mathbf{I}_m \otimes (-\mathbf{I}_d - \mathbb{E}[y\boldsymbol{x}\boldsymbol{x}^\top])}{m} + \frac{\tilde{\lambda}}{m}\mathbf{I}_{md} = \mathbf{I}_m \otimes \left( \frac{\tilde{\lambda}-1}{m}\mathbf{I}_d - \frac{\mathbb{E}[y\boldsymbol{x}\boldsymbol{x}^\top]}{m} \right).$$

Thus, we need to show that $\frac{\tilde{\lambda}-1}{m}\mathbf{I}_d - \frac{\mathbb{E}[y\boldsymbol{x}\boldsymbol{x}^\top]}{m}$ is not PSD. Let $y = \frac{1}{2}(1 + \tanh(\langle\boldsymbol{w},\boldsymbol{x}\rangle^2 - \|\boldsymbol{w}\|_2^2))$ and $\tilde{\lambda} = 1 + \mathbb{E}[y] + \gamma$ (notice that $y \geq 0$ thus $\tilde{\lambda}$ indeed satisfies the assumption in Theorem 3) with

$$\gamma = \frac{1}{4}\mathbb{E}\Big[(\langle\boldsymbol{w},\boldsymbol{x}\rangle^2 - \|\boldsymbol{w}\|_2^2)\tanh(\langle\boldsymbol{w},\boldsymbol{x}\rangle^2 - \|\boldsymbol{w}\|_2^2)\Big] > 0.$$

Then we have

$$\boldsymbol{w}^\top\left(\frac{\tilde{\lambda}-1}{m}\mathbf{I}_d - \frac{\mathbb{E}[y\boldsymbol{x}\boldsymbol{x}^\top]}{m}\right)\boldsymbol{w} = \frac{\gamma\|\boldsymbol{w}\|_2^2 + \mathbb{E}\Big[y(\|\boldsymbol{w}\|_2^2 - \langle\boldsymbol{w},\boldsymbol{x}\rangle^2)\Big]}{m}$$

$$= \frac{\gamma\|\boldsymbol{w}\|_2^2 - \frac{1}{2}\mathbb{E}\Big[(\langle\boldsymbol{w},\boldsymbol{x}\rangle^2 - \|\boldsymbol{w}\|_2^2)\tanh(\langle\boldsymbol{w},\boldsymbol{x}\rangle^2 - \|\boldsymbol{w}\|_2^2)\Big]}{m} < 0.$$

Therefore, once again we have shown that $\mathcal{R}_\lambda(\boldsymbol{W})$ is not convex on a neighborhood around zero.

# E   AUXILIARY LEMMAS

In order to be explicit, we state the following definitions and lemmas that will be used in the proof of Theorem 5. We only state the next definitions and lemmas and refer the reader to Wainwright (2019) and Vershynin (2018) for proof and more details.

**Definition 27.** (Wainwright, 2019, Definitions 2.2 and 2.7) *A real-valued random variable $z$ is said to be $\nu$-sub-Gaussian if for all $s \in \mathbb{R}$ we have $\mathbb{E}[\exp(sz)] \leq \exp(s\,\mathbb{E}[z] + \frac{s^2\nu^2}{2})$, and is said to be $u$-sub-exponential if for all $|s| \leq \frac{1}{\nu}$ we have $\mathbb{E}[\exp(sz)] \leq \mathbb{E}\Big[\exp(s\,\mathbb{E}[z] + \frac{s^2\nu^2}{2})\Big]$.*

**Lemma 28.** (Vershynin, 2018, Propositions 2.5.2 and 2.7.1) *Suppose $z$ is a zero-mean random variable and $\mathbb{E}[|z|^p]^{\frac{1}{p}} \leq L\sqrt{p}$ for all $p \geq 1$. Then $z$ is $cL$-sub-Gaussian for an absolute constant $c > 0$, i.e. $\mathbb{E}[\exp(sz)] \leq \exp(\frac{s^2 c^2 L^2}{2})$ for all $s \in \mathbb{R}$. Similarly, suppose $\mathbb{E}[|z|^p]^{\frac{1}{p}} \leq Lp$. Then $z$ is $cL$-sub-exponential for an absolute constant $c > 0$.*

**Lemma 29.** (Wainwright, 2019, Theorem 2.26) *Let $\boldsymbol{x} \sim \mathcal{N}(0, \mathbf{I}_d)$. If $f : \mathbb{R}^d \to \mathbb{R}$ is $L$-Lipschitz, then $f(x)$ is sub-Gaussian with parameter $L$.*

**Lemma 30.** (Wainwright, 2019, Example 6.3) *Let $\{\boldsymbol{x}^{(i)}\}_{1\leq i\leq n}$ be a sequence of i.i.d. standard Gaussian random vectors $x_i \sim \mathcal{N}(0, \mathbf{I}_d)$. It holds with probability at least $1 - \delta$ that*

$$\|\frac{1}{n}\sum_{i=1}^n \boldsymbol{x}^{(i)}\boldsymbol{x}^{(i)\top} - \mathbf{I}_d\|_2 \leq C\left(\sqrt{\frac{d}{n}} + \sqrt{\frac{\log(1/\delta)}{n}} + \frac{d + \log(1/\delta)}{n}\right),$$

*where $C$ is an absolute constant.*

The next lemma is the well-known symmetrization argument that upper bounds the expected value of an empirical process with Rademacher complexity.

**Lemma 31.** (Mohri et al., 2018, Theorem 3.3) *Let $\mathcal{F}$ be a class functions $f : \mathbb{R}^p \to \mathbb{R}$ for some $p > 0$. For a number of samples $T$ and a probability distribution $\mathcal{P}$ on $\mathbb{R}^p$, define the Rademacher complexity of $\mathcal{F}$ as*

$$\mathfrak{R}(\mathcal{F}) = \mathbb{E}\left[\sup_{f\in\mathcal{F}} \frac{1}{T}\sum_{t=0}^{T-1} \xi_t f(\boldsymbol{x}^{(t)})\right], \tag{E.1}$$

*where $\{\boldsymbol{x}^{(t)}\}_{t=0}^{T-1} \overset{i.i.d.}{\sim} \mathcal{P}$ and $\{\xi_t\}_{t=0}^{T-1}$ are independent Rademacher random variables (i.e. $\pm 1$ equiprobably). Then the following holds,*

$$\mathbb{E}\left[\sup_{f\in\mathcal{F}}|\frac{1}{T}\sum_{t=0}^{T-1} f(\boldsymbol{x}^{(t)}) - \mathbb{E}[f(\boldsymbol{x})]|\right] \leq 2\mathfrak{R}(\mathcal{F}).$$

Furthermore, we have the following fact for standard normal random vectors.

**Lemma 32.** *Let $x \sim \mathcal{N}(0, \mathbf{I}_d)$. There exists an absolute constant $C > 0$ such that for any $V \in \mathbb{R}^{m \times k}$ and $p \geq 1$ we have*

$$\mathbb{E}[\|\boldsymbol{V}\boldsymbol{x}\|_2^p]^{\frac{1}{p}} \leq \|\boldsymbol{V}\|_F + C\|\boldsymbol{V}\|_2\sqrt{p}.$$

**Proof.** First of all, $\|\boldsymbol{V}\boldsymbol{x}\|_2$ is a $\|\boldsymbol{V}\|_2$-Lipschitz function of $x$, thus Lemma 29 applies and $\|\boldsymbol{V}\boldsymbol{x}\|_2$ is sub-Gaussian. Furthermore, by applying Lemma 28 to $\|\boldsymbol{V}\boldsymbol{x}\|_2 - \mathbb{E}[\|\boldsymbol{V}\boldsymbol{x}\|_2]$ and Minkowski's inequality, we have

$$C\|\boldsymbol{V}\|_2\sqrt{p} \geq \mathbb{E}[\|\boldsymbol{V}\boldsymbol{x}\|_2^p]^{\frac{1}{p}} - \mathbb{E}[\|\boldsymbol{V}\boldsymbol{x}\|_2]$$
$$\geq \mathbb{E}[\|\boldsymbol{V}\boldsymbol{x}\|_2^p]^{\frac{1}{p}} - \|\boldsymbol{V}\|_{\mathrm{F}},$$

where the last inequality follows from Jensen's inequality. $\qquad\square$

**Lemma 33.** *Let $\boldsymbol{x} \sim \mathcal{N}(0, \mathbf{I}_d)$. Then $\mathbb{E}\big[\exp(c\|\boldsymbol{x}\|_2^2)\big] \leq \exp(2cd)$ for $c \leq 1/4$.*

**Proof.** Gaussian integration yields $\mathbb{E}\big[\exp(cx_i^2)\big] = \frac{1}{\sqrt{1-2c}}$. Furthermore, for $c \leq \frac{1}{4}$ we have $\frac{1}{\sqrt{1-2c}} \leq \exp(2c)$. $\qquad\square$

