# OpenReview forum: "Neural Networks Efficiently Learn Low-Dimensional Representations with SGD"
_ICLR.cc/2023/Conference — ICLR 2023 notable top 25%_

### Official Review · Reviewer_qmRw · 2022-10-24

**Confidence:** 3
**Correctness:** 4
**Technical Novelty And Significance:** 3
**Empirical Novelty And Significance:** Not applicable
**Recommendation:** 8

**Clarity, Quality, Novelty And Reproducibility:**

This paper is well-organized and easy to read, and there is a certain improvement over the existing results.

Minor comment:

- The definitions of notations $R(W), R_\lambda(W)$ omitting $a, b$ are missing, although we can guess them.

Question:

- The right-hand side of Eq. (3.2), (3.3) can be arbitrarily small as $\gamma \to \infty$. Is there any condition on $\gamma$ missed in this statement?

**Strength And Weaknesses:**

**Strengths**:

The feature learning property of neural networks trained with SGD has become a challenging research interest. Recently, many studies have been devoted to this line of research, but there are still several limitations. In this sense, the paper certainly contributes to the context by studying the SGD with multiple steps.

- This paper provides an efficient sample complexity linear in the input dimension for learning the single-index model, which improves the existing results in related studies which treat one-step gradient descent. This improvement would be nice.

- Moreover, the connection with an efficient generalization bound (Theorem 5) and the compressibility guarantee (Theorem 6) are also interesting, although these results are straightforward applications of Theorem 3.

**Weaknesses**:

- For the case $k>1$, the theory (Theorem 3) does not guarantee the convergence to the true parameter. In other words, the optimization accuracy is unclear in general, and thus the theory cannot estimate how small the expected risk can be.
- A learnable class (i.e., $k=1$ and $f$ is monotone) is still rather limited compared to [Abbe et al. (2022)] which shows the learnability of staircase functions.

However, I also acknowledge the difficulty of overcoming these limitations.

**Summary Of The Paper:**

This paper studies layer-wise training by SGD for two-layer neural networks under teacher-student setups. Specifically, for the multiple-index models as teachers, the paper shows the convergence to the principal subspace spanned by the true parameters of the teacher. Moreover, several applications of this result are provided: (i) the learnability of the single-index model, (ii) an efficient generalization error bound, and (iii) compressibility guarantees.

**Summary Of The Review:**

Although there are still several limitations in the theory, this paper certainly contributes to the context by providing an improvement over the existing studies.

---

> ### Author Response · Authors · 2022-11-16
> **Reply to reviewer qmRw**
>
> We thank the reviewer for their positive evaluation and thoughtful feedback. We address the reviewer’s questions and concerns below:
> ***
> **For the case $k>1$, the optimization accuracy is unclear, and thus the theory cannot estimate how small the expected risk can be. The learnable class (i.e.,  $k=1$ and $f$ is monotone) is rather limited.**
>
> We agree with the reviewer regarding the limitations of our work. As the reviewer pointed out, convergence to the principal subspace does not guarantee recovering the true parameter (or the principal directions) in the case $k>1$; thus our optimization result provides a somewhat limited characterization of the SGD dynamics within the principal subspace. However, we note that without resorting to the NTK or mean-field settings, a complete characterization of SGD trained two-layer networks is notoriously difficult and we see our work as an important step towards this direction, particularly when compared to recent works [Damien et al., Ba et al., and Barak et al] that only consider one gradient descent step.
>
> ***
>
>
> **The right-hand side of Eq. (3.2), (3.3) can be arbitrarily small as $\gamma \to \infty$. Is there any condition on $\gamma$ missed in this statement?**
>
> Eq. (3.2) and (3.3) hold when $T \gtrsim \tilde{\lambda}^2 / (d + \log(1/\delta))$ (recall that $\tilde{\lambda} \asymp \gamma$), as specified at the end of the Theorem statement. Choosing larger $\gamma$ implies choosing larger $\tilde{\lambda}$, which in turn implies a larger $T$. Thus in general one can not achieve immediate convergence to the principal subspace by letting $\gamma \to \infty$.
>
> We have made the following clarification in the revised version:
>
> Our results are most insightful when $\gamma \asymp 1$ with respective rates of $\widetilde{\mathcal{O}}(\sqrt{d/T})$ and $\mathcal{O}(\sqrt{d/T})$ in the constant and decreasing step size settings; this scaling allows efficient learning of certain targets (see Theorem 4). Indeed, choosing a large $\gamma$ may significantly restrict the learnability properties and will result in underfitting. However, if we ignore the underfitting issue, one can get the fastest convergence rate by choosing $\gamma \asymp \sqrt{T(d+\log(1/\delta))}$, from which we obtain ${\lVert W^T_\perp\rVert_F}/{\sqrt{m}} \lesssim {1}/{T}$.
>
> ***
>
>
> We would be happy to clarify any concerns or answer any questions that may come up during the discussion period.

---

> > ### Comment · Reviewer_qmRw · 2022-12-01
> > **Thanks**
> >
> > Thanks for your explanations. The authors clarified the condition on $\gamma$ in the revision. I think the quality of the paper is very good. I would like to keep the score.

---

### Official Review · Reviewer_vyfX · 2022-10-25

**Confidence:** 3
**Clarity, Quality, Novelty And Reproducibility:** The paper is well orgnized.
**Correctness:** 3
**Technical Novelty And Significance:** 3
**Empirical Novelty And Significance:** 3
**Recommendation:** 6

**Strength And Weaknesses:**


This paper study the convergence of SGD (applied  to two layer networks)  to the principle subspaces of the teacher model.
Strength:
Studying the convergence of SGD sequence to the estimation target (not only to the minimizer at the optimization level) is an important perspective to understand SGD in deep learning.
Since the the target model is a multi-index model which is easier to due to the low dimensional intrinsics structure. So it is not surprise the final   results   reducing  the curse of dimensionality.

Weakness:  In the perspective of nonparametric regression, we also need bound the estimation error, i.e., the estimated network function and
target function g.  But the authors never mentioned  about this.   See for example
1:  I. Kuzborskij and Cs. Szepesvári. Nonparametric regression with shallow overparameterized neural networks trained by gd with early stopping. COLT, 2021.
2:  T. Hu, W. Wang, C. Lin, and G. Cheng. Regularization matters: A nonparametric perspective
on overparametrized neural network. AISTATS, 2021.
3: S Frei, NS Chatterji, PL Bartlett， Benign Overfitting without Linearity: Neural Network Classifiers Trained by Gradient Descent for Noisy Linear Data， COLT,  2022.

**Summary Of The Paper:**

This paper  proves  that SGD on neural networks can learn low-dimensional features in certain settings, and uses this to derive novel generalization and excess risk bounds.

**Summary Of The Review:**

A nice work. And it can be better if  the author will study the nonparametric estimation error.

---

> ### Author Response · Authors · 2022-11-16
> **Reply to reviewer vyfX**
>
> We thank the reviewer for their positive evaluation and their valuable feedback.
> ***
> **In the perspective of nonparametric regression, we also need to bound the estimation error, i.e., the estimated network function and target function g.**
>
> For the single-index case, Thm 4 provides an upper bound on the excess risk (the difference between the risk of the trained neural network and the best achievable risk) under the Huber loss. If we use the squared error loss instead, this would exactly be the same as the estimation error between the prediction of the network and the true target $g$. Unfortunately, our proof technique relies heavily on the linear tail-growth of the Huber loss and relaxing this condition is not straightforward within the current framework. As such, converting the excess risk bound we have, which is under the Huber loss, to a bound in estimation error is difficult. We agree with the reviewer that extending the analysis to provide vanishing estimation error bounds is an interesting direction for future research.
>
> We finally thank the reviewer for mentioning the works on nonparametric regression with neural networks in the NTK regime. We have cited and discussed these works after Thm 4 in the revised version of our paper.
>
> ***
>
> We would be happy to clarify any concerns or answer any questions that may come up during the discussion period.

---

### Official Review · Reviewer_k8J3 · 2022-11-05

**Confidence:** 3
**Correctness:** 4
**Technical Novelty And Significance:** 3
**Empirical Novelty And Significance:** Not applicable
**Recommendation:** 8

**Clarity, Quality, Novelty And Reproducibility:**

The paper is excellently written. The paper has a similar motivation, and
applies similar technical tools, to other concurrent works in this area --
there are slight differences in the setting considered here that the authors
may unpack in more detail in the rebuttal (the differences mentioned in the
paper, that of SGD vs one-step gradient, seem like they might be superficial).



**Strength And Weaknesses:**

## Strengths

- The paper is very well written. The coverage of background material and
  related work seems essentially exhaustive. The notation and technical
  discussions are precise and easy to parse at multiple levels of detail. The
  authors provide interesting intuitions and connections to classical work in
  statistics (e.g.  model misspecification) that enhances the level of insight
  of the presentation.

- The results concern the important problem of representation learning in
  nonlinear neural networks. The authors seem to have taken some care to
  present their results in some generality (different losses and activations,
  step size regimes for SGD, etc.). The conclusions for learning the first
  layer weights are qualitatively interesting.

## Weaknesses

- The algorithmic results for neural network training + generalization are
  relatively weaker than the results for recovery of the first layer weights --
  just concerning single index models or the generalization gap.

- It would be helpful for the sake of comparison if the theorems did not hide
  properties of the link $g$ -- especially to compare with the results of
  Damian et al., which expose specific rates when the target is a polynomial
  with a certain neural-net-like structure.

## Minor / Questions / Etc.

- The role of the noise $\epsilon$ in the link function does not seem clear in
  the theorems in the main body, prior to section 4 -- what dependence does it
  play in the results?  Can it be zero? In section 4, why is this noise
  necessary? (It would be helpful in understanding this if Theorem 4 exposed
  the dependence on the subgaussian rate of the noise $\sigma^2$, say.)

- Some discussion of why the risk needs to be truncated at level $\tau$ in
  section 4 would be helpful.

- With regards to the similarity to previous works, some discussion in the
  rebuttal might be helpful to understand the distinct insights in this work
  (especially, on the technical side, the differences that the authors allude
  to in the related work section between analysis of the SGD trajectory, as
  here, with the one-step gradient trajectory, given that both are unnatural
  two-stage ("layerwise training") algorithms).

- The claim at the start of section 4.1, suggesting that a key aspect of the
  neural network training/generalization problem revolves around coping with
  model misspecification (positing the existence of such an underlying model),
  is interesting and thought-provoking. Another line of theoretical work on
  neural nets focuses on the ability of neural networks to learn to adapt to
  specific low-dimensional structures in data itself -- e.g. in terms of
  representation capacity [1], or algorithmic aspects [2-5]. Adapting to this
  type of structure, and the analysis it entails, seems somewhat different from
  what the authors study here, and I would be curious to hear the thoughts of
  the authors on whether there are similarities, whether they are in tension,
  etc.

[1] http://dx.doi.org/10.1093/imaiai/iaac001

[2] https://proceedings.neurips.cc/paper/2020/file/a9df2255ad642b923d95503b9a7958d8-Paper.pdf

[3] https://openreview.net/forum?id=O-6Pm_d_Q-

[4] https://papers.nips.cc/paper/2021/hash/f26df67e8110ee2b44923db775e3e47f-Abstract.html

[5] http://arxiv.org/abs/2206.12314



**Summary Of The Paper:**

The authors consider the representation learning ability of two layer neural
networks trained with gradient descent to approximate the outputs of a function
in a nonparametric class, applied to a few projections of a high-dimensional
input vector -- in this setting, kernel methods and other rotation-invariant
predictors generally suffer from a curse of dimensionality, but methods such as
neural networks can potentially avoid the curse of dimensionality by learning
the relevant directions and then fitting the target in this lower-dimensional
space. They consider the setting of gaussian inputs, standard (mean-field-type)
random initialization, and training with variants of stochastic gradient
descent on the square loss and smooth activations, and on convex Lipschitz losses
and the ReLU, with large weight decay regularization. Their main technical result
consists of showing that online SGD (samples from the population loss) on the
first layer weights leads to fairly rapid convergence to weights that are
suitably close to the span of the relevant $k$ directions of the teacher model
-- the rates for the norm of the irrelevant components of the trained weights
are on the order $\sqrt{dm/T}$, where $d$ is the ambient dimension, $m$ is the
number of neurons, and $T$ is the number of iterations of SGD, which is
sufficient to bound the risk given that the losses are locally Lipschitz. The
authors apply this result to obtain a guarantee for learning noisy single-index
teacher models (single neuron) up to the noise level with a two-stage procedure
(after SGD on the first-layer weights to learn the right representation for the
input, fit the readout layer weights with essentially ridge regression), and
generalization gap control (no empirical risk control) in more general settings
due to the low-dimensional representation learned.



**Summary Of The Review:**

Similar to other quasi-concurrent works, this work presents an interesting
mathematical study of a single level of representation learning in nonlinear
neural networks trained in function approximation-type tasks with imperfectly
specified models (beyond the typical teacher-student). The results here are
similar to those in concurrent works, but the excellent presentation makes this
work a valuable entry point to this evolving topic in the literature.

---

> ### Author Response · Authors · 2022-11-16
> **Reply to reviewer k8J3**
>
> We highly appreciate the reviewer’s positive evaluation and detailed feedback. We address the reviewer’s concerns below.
> ***
> **The algorithmic results for neural network training + generalization are relatively weaker than the results for recovery of the first layer weights.**
>
> We agree with the reviewer that the optimization guarantees we provided in Thm 3 cover a more general setting than their implications on the learnability of single-index models as well as the generalization gap. However, we still highlight that the excess risk estimate of $O(\sqrt{d/T})$ for an SGD-trained neural network has not been previously established, even for targets satisfying a single-index model. We also agree that establishing guarantees for SGD-trained neural networks learning multiple index models is an important open problem, and we see our work as an important step towards this direction.
>
> ***
> **It would be helpful for the sake of comparison if the theorems did not hide properties of the link g.**
>
> Remarkably, the only property of the link function $g$ that influences the convergence behavior is that it relies on the input only along $k$ principal directions. In other words, the nonlinearity in $g$ has no impact on the convergence to the principal subspace (Thm 3) as long as Assmp 1 is satisfied.
>
> The results that depend on the properties of $g$ are the learnability (Thm 4) and the warm-up case of population gradient descent (Prop 2). More specifically in Thm 4, the second derivative of $g$ enters the rate through $\Delta_*$. All other results hold irrespective of the other properties of $g$.
> ***
> **The role of the noise $\epsilon$ in the link function does not seem clear in the theorems in the main body.**
>
> The effect of noise is subsumed into other regularity conditions, on either the loss or the response. Indeed, in the ReLU activation, Lipschitzness of the loss implies bounded stochastic gradients (conditioned on $x$), and the same is implied by the bounded response variable due to Assmp 2.A for smooth activations. In either case, any noise distribution that is independent of the input is admissible, including the noiseless case where $\epsilon = 0$.
>
> In Section 4, we require sub-Gaussian noise for the term $1/T\sum_{t=0}^{T-1}\ell_H(\epsilon^{(t)})$ to concentrate around its mean. We revised our Thm 4 as per the reviewer’s suggestion to incorporate any $\nu$-sub-Gaussian noise, and the current rate shows the dependence on $\nu$ explicitly. In particular, $\nu=0$ is covered by this framework – please see Thm 4 in the revised version.
> ***
> **Some discussion of why the risk needs to be truncated at level $\tau$ in section 4 would be helpful.**
>
> We truncate the loss to obtain a sharp dependence on the inverse probability of failure $\delta$ by ensuring sub-Gaussian concentration of the empirical risk, thus yielding $O(\sqrt{\log(1/\delta)})$ dependence. We have clarified this point in the revised version.
> ***
> **With regards to the similarity to previous works, some discussion in the rebuttal might be helpful to understand the distinct insights in this work.**
>
> Compared to prior works on representation learning (e.g. Damian et al., Ba et al., Barak et al.) which mostly focused on one full-batch (specialized) gradient step, our work provides guarantees for SGD with **any number of iterations**. Indeed, one can write a closed-form expression for the updated weights after a single gradient step, which is heavily exploited in the prior work. However, in the setting of SGD, one needs to study the entire SGD trajectory on a non-convex loss landscape, which is only made possible via the Stein-type lemma (Lemma 1). Further, to provide high-probability statements, we establish bounds on the moment generating function of the distance to the principal subspace, to be iterated to derive explicit convergence guarantees, which is not previously employed in the literature on feature learning.
> ***
> **Comparison with the literature on neural network adaptation to low-dimensional structures of data.**
>
> We thank the reviewer for bringing these works to our attention. We highlight that the convergence to the principal subspace aligns with neural networks’ ability to adapt to the underlying low-dimensional structure of the data, and thus is in agreement with [1-4] (in particular [2] also suggests that neural networks beat kernel methods when the input is isotropic). However, [5] proposes an interesting counterpoint where in certain settings with inputs having a fixed low dimension, lazy training (with NTK) can generalize better than feature learning. That said, our work focuses on the high-dimensional setting, in which learning polynomials of the input has provably worse generalization in the lazy training regime (Donhauser et al.). We have included a detailed discussion of these works in the revised version.
>
> ***
>
> We would be happy to clarify any concerns or answer any questions that may come up during the discussion period.

---

> > ### Comment · Reviewer_k8J3 · 2022-12-06
> > **thanks**
> >
> > Thank you for the detailed response and for fixing various small issues in the revision. I am happy with the work's quality and contribution and have updated my score accordingly.

---

### Decision · Program_Chairs · 2023-01-20

**Decision:**

Accept: notable-top-25%

**Justification For Why Not Higher Score:**

Overall, this is a strong technical paper. There are some minor limitations to the results -- the strongest generalization guarantees are for the k = 1 case, and, while the theory shows convergence to the principle subspace, it does not control what is learned within the principle subspace.

**Justification For Why Not Lower Score:**

This is a high quality technical paper, with an interesting message (weights converge to the principal subspace) and technical contributions (control of entire SGD trajectory, generalization results for single index model and generalization gap for multi-index model that leverage low-dimensional structure). It deserves to be accepted in some form.

**Metareview: Summary, Strengths And Weaknesses:**

The paper studies neural network learning problems in which the target y is a function of a few linear projections <u_1,x>, …, <u_k,x> of the input x (the “multi-index model”). This problem is motivated by the general goal of understanding how deep networks adapt to low-complexity structure in the learning task. The paper studies the convergence and generalization of SGD for learning two-layer networks from this data, and shows that with weight decay, the first layer weights converge to the subspace spanned by u_1, …, u_k. The paper gives generalization bounds for SGD which are dictated by the dimension of this subspace, and proves that the weights of the learned network are compressible.

Reviewers expressed a uniformly positive evaluation of the paper’s contributions: conceptually, it illustrates how neural networks adapt to low-dimensional task structure, while at a technical level it improves over related works by controlling the entire trajectory of SGD, rather than just a single gradient step (which has a closed-form expression). One technical limitation is that although this approach shows convergence to the principal subspace, it does to control the behavior within the principal subspace, and hence for k > 1 controls the generalization gap, and not the excess risk. However, these results still show the benefit of low-dimensionality, and the paper’s learning results for single index models are novel. The paper’s results for single index models contrast favorably with lower bounds for learning polynomials in the NTK regime. Reviewers also found the paper to be very clearly presented.

**Note From Pc:**

if the above contains the word "oral" or "spotlight" please see: "oral" presentation means -> notable-top-5% and "spotlight" means -> notable-top-25%. As stated in our emails, we are disassociating presentation type from AC recommendations